# FROM BRICKS TO BRIDGES: PRODUCT OF INVARIANCES TO ENHANCE LATENT SPACE COMMUNICATION

**Irene Cannistraci**[1]     **Luca Moschella\*[1]**     **Marco Fumero\*[1]**     **Valentino Maiorca**[1]

**Emanuele Rodolà**[1]

[1]Sapienza University of Rome, \*Equal Contribution

## ABSTRACT

It has been observed that representations learned by distinct neural networks conceal structural similarities when the models are trained under similar inductive biases. From a geometric perspective, identifying the classes of transformations and the related invariances that connect these representations is fundamental to unlocking applications, such as merging, stitching, and reusing different neural modules. However, estimating task-specific transformations a priori can be challenging and expensive due to several factors (e.g., weights initialization, training hyperparameters, or data modality). To this end, we introduce a versatile method to directly incorporate a set of invariances into the representations, constructing a product space of invariant components on top of the latent representations without requiring prior knowledge about the optimal invariance to infuse. We validate our solution on classification and reconstruction tasks, observing consistent latent similarity and downstream performance improvements in a zero-shot stitching setting. The experimental analysis comprises three modalities (vision, text, and graphs), twelve pretrained foundational models, nine benchmarks, and several architectures trained from scratch.

## 1 INTRODUCTION

Discovering symmetries and conserved quantities is a core step for extracting meaningful representations from raw data in biological and artificial systems (Higgins et al., 2022; Benton et al., 2020; Lyle et al., 2020; Marchetti et al., 2023). Achieving invariance to specific groups of transformations within neural models holds significant utility in a wide range of real-world applications, such as comparing similar latent spaces across multiple training instances, facilitating communication, and enabling model reuse (Cohen & Welling, 2016; Fawzi et al., 2016; Salamon & Bello, 2017; Klabunde et al., 2023; Lähner & Moeller, 2023; Maiorca et al., 2023). These desired invariances can be defined with respect to transformations in the input space (Benton et al., 2020; Immer et al., 2022; Cohen & Welling, 2016; Cohen et al., 2019), or in relation to the latent space, as explored by Moschella et al. (2023). Such properties can arise implicitly from architectural choices (Cohen & Welling, 2016; Cohen et al., 2019; Worrall et al., 2017; Zhou et al., 2017) or be explicitly enforced using methods like loss penalties (Arjovsky et al., 2019). A recent study introduced the concept of Relative Representation (RR) (Moschella et al., 2023). In its original formulation, this framework enforces invariance to angle-preserving transformations of the latent space. This approach enhances

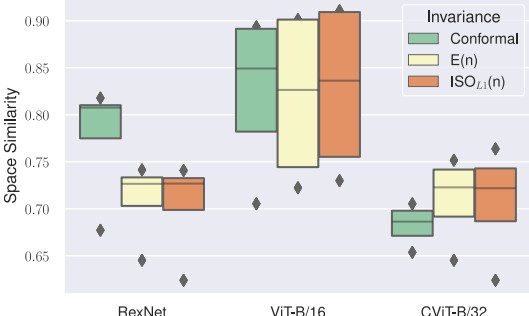

Figure 1: CKA similarity between pretrained models on `F-MNIST` measured infusing invariances to specific classes of transformations (Conformal[2], Euclidean, Orthogonal). Each bar reports the distribution of similarity to the other models. The score diversity highlights the absence of a universal transformation connecting all latent spaces.

---

[2]Global orthogonal transformations composed with local rescalings

communication between latent spaces by projecting them into a shared relative space determined by distances between data points. However, as shown in Figure 1, the transformations relating different neural representations are not always consistent with a single class of transformations, such as the one considered in Moschella et al. (2023). Determining a priori which class of transformations relates distinct latent spaces is challenging due to complex interactions in the data, and multiple nuisance factors that are typically irrelevant but can nevertheless affect the representation (e.g. random initialization, neural architecture, and data modality). To address this challenge, we expand upon the method of RR, presenting a framework to *efficiently incorporate a set of invariances into the learned latent space*. This is achieved by constructing a *product space of invariant components* on top of the latent representations of, possibly pretrained, neural models. Each component of this product space is obtained by projecting samples as a function of fixed data points, denoted as *anchors*. Using different similarity functions for each subspace, we can infuse invariances to specific transformations into each component of the product space. Our main contributions can be summarized as follows:

- We show that the class of transformation that relates representations learned by distinct Neural Networks (NNs)–trained on semantically similar data–may vary and depends on multiple factors;

- We introduce a framework for infusing multiple invariances into a single latent representation, constructing a *product space of invariant components* to enhance latent communication;

- We validate our findings on stitching tasks across various data modalities, including images, text, and graphs: product of invariances can capture complex transformations of the latent space in a single representation, achieving the best performance without any prior knowledge of the transformation or the factors that may affect it.

## 2   RELATED WORK

**Representation Similarity.** Several metrics have been proposed to compare latent spaces generated by independent NNs, capturing their inherent similarity up to transformations that correlate the spaces. A classical statistical method is Canonical Correlation Analysis (CCA) (Hotelling, 1992), which is invariant to linear transformations; variations of CCA seek to improve robustness through techniques like Singular Value Decomposition (SVD) and Singular Value CCA (SVCCA) (Raghu et al., 2017) or to reduce sensitivity to perturbations using methods such as Projection Weighted CCA (PWCCA) (Morcos et al., 2018). Closely related to these metrics, the Centered Kernel Alignment (CKA) (Kornblith et al., 2019) measures the similarity between latent spaces while disregarding orthogonal transformations. However, recent research (Davari et al., 2022) demonstrates its sensitivity to shifts in the latent space. Finally, Limbeck et al. (2023) proposes a novel family of magnitude-based measures to quantify the intrinsic diversity of latent spaces, while Wayland et al. (2024) employs persistent homology to measure the (dis)similarity of different representations.

**Learning and Incorporating Invariance and Equivariance into Representations.** Invariances in NNs can be enforced through various techniques operating at different levels, including adjustments to model architecture, training constraints, or input manipulation (Lyle et al., 2020). Benton et al. (2020) proposes a method to learn invariances and equivariances, Immer et al. (2022) introduces a gradient-based approach that effectively captures inherent invariances in the data. Meanwhile, van der Ouderaa & van der Wilk (2022) enables training of NNs with invariance to specific transformations by learning weight-space equivalents instead of modifying the input data. Other works directly incorporate invariances into the model through specific constraints. Rath & Condurache (2023) enforces a multi-stream architecture to exhibit invariance to various symmetry transformations without relying on data-driven learning; Kandi et al. (2019) propose an improved Convolutional Neural Network (CNN) architecture for better rotation invariance; Gandikota et al. (2021) introduces a method for designing architectures that are invariant or equivariant to structured transformations; Sanborn & Miolane (2023) propose a general method to achieve robust group-invariance in Group-Equivariant CNNs; and Fumero et al. (2021) proposes to recover the factors of variation underlying the data distribution by modeling them as a product manifold. Finally, Moschella et al. (2023) proposes an alternative representation of the latent space that guarantees invariance to angle-preserving transformation without requiring additional training but only a set of anchors, possibly very small (Cannistraci et al., 2023), or very large (Norelli et al., 2023). This approach has shown efficiency in different settings (Crisostomi et al., 2023; Ricciardi et al., 2023; Chen et al., 2023).

Our work leverages the RR framework to *directly incorporate a set of invariances into the learned latent space, creating a product space of invariant components which, combined, can capture complex transformations of the latent space*.

# 3 INFUSING INVARIANCES

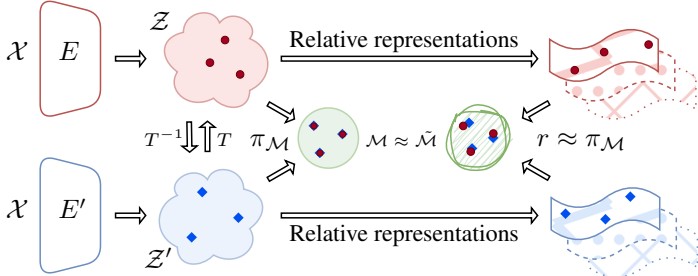

Figure 2: **Framework description**. Given two latent spaces $\mathcal{Z}, \mathcal{Z}'$ related by an unknown transformation $T$ (resp. $T^{-1}$), we assume that there exist a manifold $\mathcal{M}$ where samples in $\mathcal{Z}, \mathcal{Z}'$ coincides when projected into $\mathcal{M}$, via $\pi_{\mathcal{M}}$. We approximate $\mathcal{M}$ building a product space $\tilde{\mathcal{M}}$, where each space is a RR computed using a similarity function $d_i$ *invariant* to a specific, known class of transformations. Combining the resulting spaces, we recover a representation $r$ which should approximate $\pi_{\mathcal{M}}$.

**Setting.** We consider NNs as parametric functions $F_\theta$ compositions of *encoding* and *decoding* maps $F_\theta = D_{\theta_2} \circ E_{\theta_1}$, where the encoder $E_{\theta_1}$ is responsible for computing a latent representation $z = E_{\theta_1}(x), \quad x \in \mathcal{X}$ for some domain $\mathcal{X}$, with $dim(\mathcal{Z}) << dim(\mathcal{X})$; and the decoder $D_{\theta_2}$ is responsible for solving the task at hand (e.g., reconstruction, classification). In the following, we will drop the dependence on parameters $\theta$ for notational convenience. For a single module $E$ (equivalently for $D$), we indicate with $E_{\mathcal{X}}$ if the module $E$ was trained on the domain $\mathcal{X}$. In the upcoming, we will summarize the necessary background to introduce our method.

**Background.** The RR framework (Moschella et al., 2023) provides a straightforward approach to represent each sample in the latent space according to its similarity to a set of fixed training samples, denoted as *anchors*. Representing samples in the latent space as a function of the anchors corresponds to transitioning from an absolute coordinate frame into a *relative* one defined by the anchors and the similarity function. Given a domain $\mathcal{X}$, an encoding function $E_{\mathcal{X}} : \mathcal{X} \to \mathcal{Z}$, a set of anchors $\mathcal{A}_{\mathcal{X}} \subset \mathcal{X}$, and a similarity or distance function $d : \mathcal{Z} \times \mathcal{Z} \to \mathbb{R}$. The RR for each sample $x \in \mathcal{X}$ is:

$$RR(z; \mathcal{A}_{\mathcal{X}}, d) = \bigoplus_{a_i \in \mathcal{A}_{\mathcal{X}}} d(z, E_{\mathcal{X}}(a_i)) \tag{1}$$

where $z = E_{\mathcal{X}}(x)$, and $\bigoplus$ denotes row-wise concatenation. In Moschella et al. (2023), $d$ was set as Cosine similarity. This choice induces a representation invariant to *angle-preserving transformations*. In this work, our focus is to *leverage different choices of the similarity function to induce a set of invariances into the representations to capture complex transformations between latent spaces*.

**Overview.** When considering different NNs $F, F'$, we are interested in modeling the class of transformations $\mathcal{T}$ that relates their latent spaces $\mathcal{Z}, \mathcal{Z}'$. $\mathcal{T}$ could be something known, e.g., rotations, or a nontrivial, complex class of transformations. The two networks could differ by their initialization seeds (i.e., training dynamics), by architectural changes, or even domain changes, i.e., $\mathcal{X} \neq \mathcal{X}'$, which could affect the latent space in a different way (as observed in Figure 1). The fundamental assumption of this work is that these variations induce changes in the latent representations of the models, but there exists an underlying manifold $\mathcal{M}$ where the representations are the same (see Figure 2). Formally:

**Assumption.** *Given multiple models $\mathcal{F}_1..\mathcal{F}_n$ we assume that there exists a manifold $\mathcal{M}$ which identifies an equivalence class of encoders $\mathcal{E}_{\mathcal{T}}$ induced by the class of transformation $\mathcal{T}$ (e.g. rotations), defined as $\mathcal{E}_{\mathcal{T}} := \{E \mid \pi_{\mathcal{M}}TE = \pi_{\mathcal{M}}E, \quad \forall T \in \mathcal{T}\}$, where $\pi_{\mathcal{M}}$ represent the projection on $\mathcal{M}$. $\mathcal{M}$ is equipped with a metric $d_{\mathcal{M}}$ which is preserved under the action of elements of $\mathcal{T}$, i.e. $d_{\mathcal{M}}(\pi_{\mathcal{M}}z, \pi_{\mathcal{M}}z') = d_{\mathcal{M}}(\pi_{\mathcal{M}}T(z), \pi_{\mathcal{M}}T(z')), \forall T \in \mathcal{T}$.*

What we look for is a function $r$ which independently projects the latent spaces $\mathcal{Z}_1..\mathcal{Z}_n$ into $\mathcal{M}$ and is *invariant* to $\mathcal{T}$, i.e. $r(z) = r(Tz)$, for each $T \in \mathcal{T}$, and for each $z \in \mathcal{Z}_1..\mathcal{Z}_n$. Generalizing the framework of Moschella et al. (2023) to arbitrary similarity functions, or distance metrics, gives us a straightforward way to define representations $r$ invariant to specific classes of transformations.

However, $d_{\mathcal{M}}$ is typically unknown a priori, and in general, it is challenging to capture $\mathcal{T}$ with a single class of transformations (as observed in Figure 1 and empirical demonstrated in Section 4.1). To overcome this challenge, in this work, we approximate $\mathcal{M}$ with a product space $\tilde{\mathcal{M}} := \prod_{i=1}^{N} \mathcal{M}_i$, where each component is obtained by projecting samples of $\mathcal{Z}$ in a RR space equipped with a different similarity function $d_i$. Each $\mathcal{M}_i$ will have properties induced by a similarity function $d_i$ invariant to a specific, known, class of transformations $\tilde{\mathcal{T}}_i$ (e.g. dilations). By combining this set of invariances, we want to recover the representation $r$ such that it approximates well $\pi_{\mathcal{M}}$ (see Figure 2). We define $r$ formally as the *projection* from $\mathcal{Z}$ to $\tilde{\mathcal{M}}$:

**Definition** (Product projection). *Given a set of latent spaces $\mathcal{Z}_1..\mathcal{Z}_n$, related to one another by an unknown class of transformation $\mathcal{T}$, a set of similarity functions $\mathcal{D}$ each one invariant to a specific known class of transformations $\tilde{\mathcal{T}}_i$ (e.g. rotations), i.e. $RR(z, d_i) = RR(Tz, d_i)$, $\forall T \in \tilde{\mathcal{T}}_i$. We define the product projection $r : \mathcal{Z} \mapsto \tilde{\mathcal{M}}$ as:*

$$r(z) = \phi \circ RR(z; \mathcal{A}_{\mathcal{X}}, d_i), \quad \forall d_i \in \mathcal{D}$$

*where $\phi$ is an aggregation function (e.g. concatenation) responsible for merging the relative spaces induced by each $d_i \in \mathcal{D}$.*

We give more details on different strategies on how to implement $\phi$ in section 3.

**Distance-induced invariances.** We leverage the RR framework considering the following similarity functions $d$: Cosine, Euclidean, Manhattan ($L_1$), Chebyshev ($L_{\infty}$), each one inducing invariances to a specific, known class of transformations. For formal definitions, synthetic examples, and visualizations, please refer to the Appendix A.2.

**Aggregation functions.** This section summarizes different strategies to construct the product space $\tilde{\mathcal{M}}$, directly integrating a set of invariances into the representations. Consider a latent space $\mathcal{Z}$ image of an encoder $E : \mathcal{X} \mapsto \mathcal{Z}$, and a set of similarity functions $\mathcal{D}$. For each $d \in \mathcal{D}$, we produce $n = |\mathcal{D}|$ relative latent spaces. Every space is produced via a similarity function (i.e., Cosine, Euclidean, $L_1$, or $L_{\infty}$), enforcing invariance to a specific class of transformations.

These spaces can be aggregated using diverse strategies, corresponding to different choices of $\phi$:

- *Concatenation* (Concat): the subspaces are independently normalized and concatenated, giving to $\tilde{\mathcal{M}}$ the structure of a cartesian product space.
- *Aggregation by sum* (MLP+Sum): similar to DeepSet (Zhang et al., 2019), the spaces are independently normalized and non-linearly projected. The resulting components are summed.
- *Self-Attention* (SelfAttention): the spaces are independently normalized and aggregated via a self-attention layer.

When not specified, all the results are obtained using the *Aggregation by sum* strategy. For the implementation details of each strategy, please refer to the Appendix A.3. The product space $\mathcal{M}$ yields a *robust* latent representation, made of *invariant components* which are combined to capture *nontrivial, complex* transformations, boosting the performance on downstream tasks.

## 4 EXPERIMENTS

In this section, we perform qualitative and quantitative experiments to analyze the effectiveness of our framework in constructing representations that can capture complex transformations of the latent space. In Section 4.1, we empirically motivate our study by analyzing the similarity between latent spaces generated by trained from scratch AutoEncoder (AE) and Variational AutoEncoder (VAE) architectures, and pretrained foundation models. We analyze the emerging class of transformations between different latent spaces by enforcing invariances into the representations. Additionally, in

Section 4.2, we assess the zero-shot stitching performance of our framework across text, vision, and graph modalities. Finally, in Section 4.3 we conduct an ablation study on different aggregation functions, and in Section 4.4 we investigate attention weights and their significance in selecting the optimal relative space.

## 4.1 LATENT SPACE ANALYSIS

### TRAINING FROM SCRATCH

**Experimental setting.** We perform image reconstruction on `CIFAR-10`, `CIFAR-100` (Krizhevsky et al., 2009), `MNIST` (Deng, 2012), and `Fashion MNIST` (Xiao et al., 2017) datasets (refer to Table 7 for additional information). Using comparable convolutional architectures, we focus on AEs and VAEs. We consider models with an unflattened latent image bottleneck (referred to as AE and VAE) that preserve the image spatial structure, and models with linear projections into and out of a flattened latent space (referred to as Linearized AutoEncoder (LinAE) and Linearized Variational AutoEncoder (LinVAE)). Using different random seeds, we train five instances of each model until convergence. Then we project each latent space into its relative counterpart using distinct projection functions to infuse different invariances in each representation. Finally, for each combination, we measure the cross-seed latent space similarity.

**Result Analysis.** In Figures 3, 12 and 13, we report the Pearson and Spearman cross-seed correlations for various architectures on diverse datasets. These outcomes illustrate that it is impossible to unify the latent spaces obtained with different initializations by means of a single invariance. For instance, in Figure 3, we observe that the highest cross-seed similarity is achieved using different projection types when considering the VAE or LinVAE architecture, even when keeping fixed all the other parameters. Additionally, dataset variations significantly alter the trends in the behavior of all the architectures. This discovery challenges the assumption in Moschella et al. (2023) that angle-preserving transformations are the primary drivers of correlation among the latent spaces of models trained with different seeds. Please refer to Figures 12 and 13 for additional results.

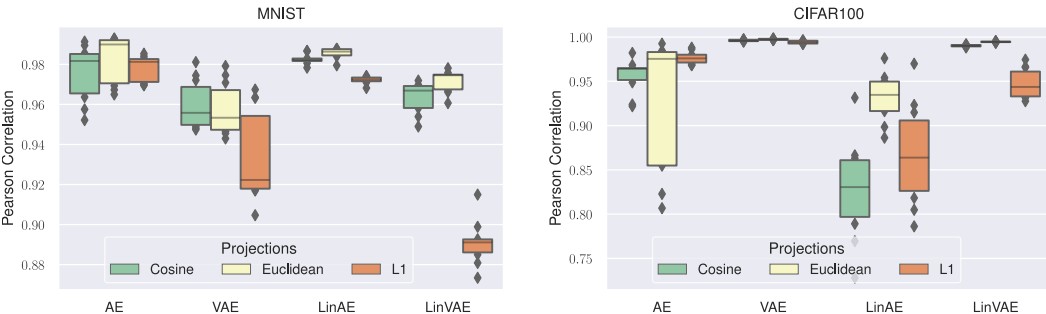

Figure 3: **Latent Spaces Cross-Seed Similarity.** Cross-seed pearson correlation of latent spaces for AEs trained on `MNIST` (*left*) and `CIFAR-100` ((*right*)) until convergence. Notably, no single projection consistently outperforms others across all settings. The $L_\infty$ projection is not displayed to improve visualization.

### PRETRAINED MODELS

**Experimental setting.** We analyze the similarity of latent spaces produced by pretrained foundational models in both the vision and text domains. For the vision domain, we evaluate five distinct foundational models (either convolutional or transformer-based) using the `CIFAR-10`, `CIFAR-100`, `MNIST`, and `F-MNIST` datasets. Meanwhile, in the text domain, we assess seven different foundational models using the `DBpedia` (Zhang et al., 2015), `TREC(Coarse)` (Hovy et al., 2001), and `N24news(Text)` (Wang et al., 2022) datasets. Refer to Table 6 for details on the pretrained models and Table 7 for the datasets.

**Result Analysis.** In Figure 4, we report the Linear CKA correlations for various pretrained models for vision and text modalities. This analysis highlights the absence of a universally shared transformation class that connects latent spaces of foundation models across distinct conditions. For example, on `CIFAR-10`, the highest similarity is achieved with different projection functions when using different

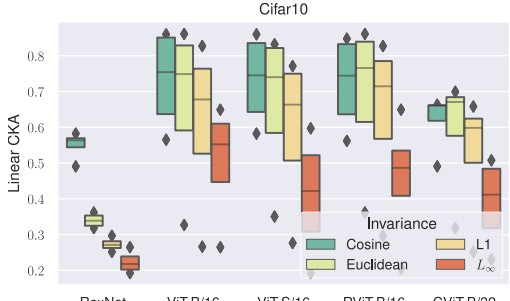 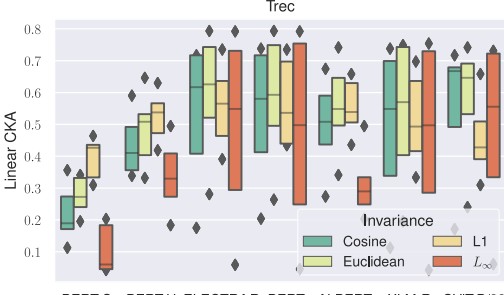

Figure 4: **Latent Spaces Cross-Architecture Similarity.** Linear CKA similarity of latent spaces across several pretrained models on `CIFAR-10` (*left*) and `TREC` (*right*). In each bar, we report the space similarities distribution to the other models while infusing a specific invariance. There is no singular projection that consistently outperforms others across all configurations.

architectures. Moreover, it is possible to see that similar architectures (i.e., ViT-based models) exhibit similar trends when tested on the same dataset. Refer to Figure 14 for further results.

**Takeaway.** The transformation class that correlates different latent spaces produced by both pretrained and trained-from-scratch models depends on the dataset, architecture, task, and possibly other factors.

## 4.2 DOWNSTREAM TASK: ZERO-SHOT STITCHING

**Experimental setting.** We perform zero-shot stitching *classification* using text, vision, and graph modalities with various models and datasets. For the vision and text domains, we used the same datasets and pretrained models employed in Section 4.1. For the *Graph* domain, we employed the `CORA` dataset (Sen et al., 2008) and a Graph Convolutional Network (GCN) trained from scratch (refer to Tables 6 and 7 for additional details of models and datasets). Additionally, we perform zero-shot stitching *reconstruction* using AEs on `CIFAR-100`. Relative decoders are trained with three different seed values, and the resulting representations are transformed into relative representations by projecting the encodings onto 1280 randomly selected but fixed anchors. It is important to highlight, as demonstrated in Table 9 and Figure 10, that varying the number of anchors results in different emerging transformations, indicating that a single projection function cannot capture the desired invariance. Nevertheless, our method achieves the highest score, *regardless of the number of anchors*.

**Zero-Shot Model Stitching.** The stitching methodology allows combining components of different NNs to obtain a new model. In this paper, we adopted the same setting proposed by Moschella et al. (2023), where each element of the stitched model functions as an autonomous frozen module: the encoder handles data embedding, while the dedicated relative decoder manages the downstream task. Refer to Figure 11 for a visual depiction of the procedure. The approach is termed *zero-shot* because the stitching procedure is executed without any further training or fine-tuning.

Table 1: **Graph and Text Classification Stitching Performance.** Zero-shot accuracy scores across various decoders, seeds, and datasets. For the text domain, we use pretrained models, while for the graph domain, we train `GCN` models from scratch and evaluate the stitching across seeds. Additionally, we calculate the stitching index for the graph modality, showing that composing different projections using the *Aggregation by sum* enables zero-shot stitching *without* any performance drop in this setting, ensuring competitive end-to-end performance. Refer to Tables 11 and 14 for complete results.

| | Text | | Graph | |
| | ALBERT | | GCN | |
| Projection | DBpedia↑ | TREC↑ | CORA ↑ | Stitching Index ↑ |
|---|---|---|---|---|
| Cosine | $0.50 \pm 0.02$ | $0.54 \pm 0.03$ | $\mathbf{0.53} \pm 0.06$ | 0.71 |
| Euclidean | $0.50 \pm 0.00$ | $0.60 \pm 0.03$ | $0.27 \pm 0.06$ | 0.58 |
| $L_1$ | $\mathbf{0.52} \pm 0.01$ | $\mathbf{0.65} \pm 0.02$ | $0.26 \pm 0.06$ | 0.58 |
| $L_\infty$ | $0.18 \pm 0.02$ | $0.29 \pm 0.06$ | $0.12 \pm 0.03$ | $\mathbf{1.00}$ |
| Cosine, Euclidean, $L_1$, $L_\infty$ | $\mathbf{0.53} \pm 0.01$ | $\mathbf{0.65} \pm 0.02$ | $\mathbf{0.77} \pm 0.01$ | $\mathbf{1.00}$ |

Table 2: **Image Classification Stitching Performance Cross-Architecture and Cross-Seed.** Zero-shot accuracy score across different pretrained models, seeds, and datasets. The proposed method using the *Aggregation by sum* consistently achieves the highest accuracy score or comparable results, without prior knowledge of the optimal projection to employ. See Table 10 for complete results.

| Encoder | Projection | Accuracy ↑ | | | |
|---|---|---|---|---|---|
| | | CIFAR-100 | CIFAR-10 | MNIST | F-MNIST |
| CViT-B/32 | Cosine | $0.52 \pm 0.03$ | $\mathbf{0.87} \pm 0.02$ | $0.61 \pm 0.06$ | $0.68 \pm 0.02$ |
| | Euclidean | $0.53 \pm 0.02$ | $\mathbf{0.87} \pm 0.02$ | $\mathbf{0.66} \pm 0.05$ | $\mathbf{0.70} \pm 0.03$ |
| | $L_1$ | $\mathbf{0.53} \pm 0.04$ | $\mathbf{0.87} \pm 0.02$ | $\mathbf{0.66} \pm 0.05$ | $\mathbf{0.70} \pm 0.03$ |
| | $L_\infty$ | $0.27 \pm 0.04$ | $0.52 \pm 0.04$ | $0.57 \pm 0.03$ | $0.55 \pm 0.01$ |
| | Cosine, Euclidean, $L_1$, $L_\infty$ | $\mathbf{0.58} \pm 0.03$ | $\mathbf{0.88} \pm 0.02$ | $\mathbf{0.68} \pm 0.05$ | $\mathbf{0.70} \pm 0.01$ |
| RViT-B/16 | Cosine | $\mathbf{0.79} \pm 0.03$ | $0.94 \pm 0.01$ | $0.69 \pm 0.04$ | $0.76 \pm 0.03$ |
| | Euclidean | $\mathbf{0.79} \pm 0.03$ | $0.94 \pm 0.01$ | $\mathbf{0.71} \pm 0.04$ | $0.77 \pm 0.03$ |
| | $L_1$ | $0.77 \pm 0.04$ | $\mathbf{0.95} \pm 0.01$ | $\mathbf{0.71} \pm 0.04$ | $\mathbf{0.79} \pm 0.03$ |
| | $L_\infty$ | $0.31 \pm 0.03$ | $0.75 \pm 0.04$ | $0.61 \pm 0.05$ | $0.60 \pm 0.03$ |
| | Cosine, Euclidean, $L_1$, $L_\infty$ | $\mathbf{0.81} \pm 0.04$ | $\mathbf{0.95} \pm 0.01$ | $\mathbf{0.72} \pm 0.04$ | $0.76 \pm 0.04$ |

**Results Analysis.** Tables 1 and 2 present the performance for the classification downstream task of various projection functions for different modalities. While Figure 5 shows the qualitative results for the reconstruction task (see Table 23 for quantitative results). As previously observed in Section 4.1 and Figure 3, the experiments reveal the absence of a single optimal projection function across architectures, modalities, and even within individual datasets. Our proposed framework, which leverages a product space to harness multiple invariances, followed by a trainable aggregation mechanism, consistently achieves superior accuracy across most scenarios. It is important to emphasize that the dimensionality of each independent projection and the aggregated product space remains constant, ensuring fair comparison. Additional stitching results on graphs and text in Appendix Table 14 and 11; moreover, in Tables 15 to 21 we show the performance for each pair of encoder and decoder without averaging over the architectures. The reference end-to-end performance are reported in Tables 24 to 31 to better interpret the performance of the stitched models on the downstream tasks. To this end, we propose an additional evaluation metric named the *Stitching Index* computed as the ratio between the stitching score and the end-to-end score. It measures how closely the stitching accuracy aligns with the original score, i.e., a stitching score of one indicates there is no drop in performance when stitching modules. Results in Table 1 highlight that our method enables zero-shot stitching *without* any performance drop in this setting while still ensuring competitive end-to-end performance.

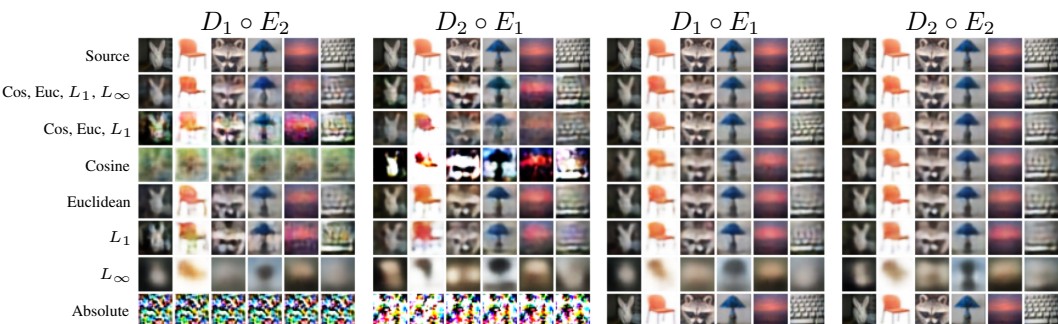

Figure 5: **Reconstruction Qualitative Results on `CIFAR-100` using AEs.** The last two sets of columns ($D_1 \circ E_1$ and $D_2 \circ E_2$) are the end-to-end AEs with unique initialization seeds, while the first two ($D_1 \circ E_2$ and $D_2 \circ E_1$) illustrate the outputs of the zero-shot stitching of these independently trained models. The first row displays the source images, the two subsequent rows show the results from distinct combinations of projection aggregated through the *Aggregation by sum*, while the last one shows the outputs from a baseline model that does not incorporate our methodology. It is interesting to see that when using the $L_\infty$ projection (*second to last row*), the reconstructed images are blurred, but when removing the $L_\infty$ space from the aggregation (*third row*) the reconstruction drops in performance, meaning that it captures information not captured by the others.

**Takeaway.** A product space of invariant components can improve the zero-shot stitching performance without any prior knowledge of the class of transformation that relates different spaces.

### 4.3 AGGREGATION FUNCTIONS: ABLATION

**Experimental setup.** We perform an ablation study on the merging strategies presented in Section 3. In these experiments, we conduct zero-shot stitching classification on the three modalities using the same datasets and models described in previous sections (refer to Tables 6 and 7 for additional details of models and datasets). The relative decoders are trained using three different seeds, and the accuracy score is assessed on each assembled model.

Table 3: **Ablation Study on the Aggregation Functions.** Zero-shot classification accuracy score across different architectures, seeds, and datasets. The *Aggregation by sum* (*third row*) obtains consistently the best accuracy score. See Appendix Tables 12 to 14 for the complete results.

|  | Vision | Text | Graph |
|---|---|---|---|
| Aggregation | RViT-B/16 | BERT-C | GCN |
| Concat* | $0.81 \pm 0.00$ | $0.54 \pm 0.03$ | $0.75 \pm 0.02$ |
| MLP+SelfAttention | $0.84 \pm 0.01$ | $0.51 \pm 0.03$ | $0.63 \pm 0.13$ |
| MLP+Sum | $\mathbf{0.85} \pm 0.00$ | $\mathbf{0.55} \pm 0.04$ | $\mathbf{0.77} \pm 0.01$ |
| SelfAttention | $0.76 \pm 0.03$ | $0.36 \pm 0.22$ | $0.76 \pm 0.02$ |

**Result Analysis.** Table 3 presents the results of the ablation study conducted using various aggregation methodologies (complete results can be found in Tables 12 to 14). Among the different methodologies, MLP+Sum outperforms the others consistently. This method preprocesses each subspace with an independent MultiLayer Perceptron (MLP), which includes a normalization layer, a linear layer, and tanh activation, and then sums the resulting representations. It is essential to highlight that the Concat aggregation method is not directly comparable to the others since it increases the dimensionality of the space linearly with the number of subspaces (refer to Appendix A.4 for more details).

**Takeaway.** The aggregation by sum methodology consistently guarantees the highest performance between merging strategies without increasing the dimensionality of the space.

### 4.4 SPACE SELECTION

In the preceding sections, we discussed integrating individual and multiple invariances into the representation through various projection functions and appropriate aggregation strategies. In this section, we aim to analyze and understand if tuning only the aggregation strategy at stitching time is a reasonable cost for selecting the optimal space. We focus on the *Self-attention* aggregation, which is a single self-attention layer as described in Section 3, and finetune only the Query, Key, Value (QKV) parameters (i.e., the ones responsible for space blending). Each space is generated by its own projection function. We remark that stitching-time fine-tuning is exclusive to this experiment.

**Experimental setup.** We identify two crucial components within the stitched model: (1) the linear projections associated with QKV in the attention mechanism, which is responsible for selecting and blending the spaces, and (2) the MLP in the classification head following the attention mechanism, which classifies the aggregated embeddings. We examine two distinct approaches: the first approach fine-tunes only the first component (`QKV opt`), while the second one fine-tunes the second one (`MLP opt`). All the experiments in this section are conducted on the `CIFAR-100` dataset using the `RexNet` as encoder and the `ViT-B/16` as decoder.

**Result Analysis.** Table 4 summarizes downstream classification accuracy for the stitched model using various projection functions and aggregation strategies. Incorporating multiple invariances and aggregating them via *Self-attention* (*fifth row*) does not perform well; meanwhile, using the *MLP+Sum* or the Cosine projection alone is more effective. This is expected, considering the attention mechanism is trained to improve end-to-end performance rather than to maximize compatibility between different spaces. Incorporating the adaptation strategies at stitching time significantly boosts performance, either focusing on the optimal space selection and blending (`QKV opt`) or the classification head (`MLP opt`). We find that an informed fine-tuning of the parameters responsible for the space blending (i.e., only the QKV projections) significantly impacts performances, even more than tuning the whole classifier. Figure 6 illustrates the attention weights averaged over the test dataset, the attention weights of the zero-shot stitched model (*left*) remain unchanged when

fine-tuning only the classifier (*right*). Meanwhile, fine-tuning the QKV projections (*center*) shifts the attention weights to allocate less importance to worse-performing projections (i.e., $L_\infty$).

Table 4: **Optimal Space Selection**. Classification accuracy for the stitched model with `RexNet` as encoder and `ViT-B/16` as decoder on `CIFAR-100`, using different projection functions and aggregation strategies. Fine-tuning the space selection and blending module (`QKV opt`) has a more significant effect on performance improvement than fine-tuning the MLP head (`MLP opt`).

| Projection | Aggregation | Accuracy ↑ |
|---|---|---|
| Cosine | - | **0.50** |
| Euclidean | - | 0.38 |
| $L_1$ | - | 0.24 |
| $L_\infty$ | - | 0.21 |
| Cosine, Euclidean, $L_1$, $L_\infty$ | SelfAttention | 0.17 |
| Cosine, Euclidean, $L_1$, $L_\infty$ | MLP+Sum | **0.45** |
| Cosine, Euclidean, $L_1$, $L_\infty$ | SelfAttention + `QKV opt` | **0.75** |
| Cosine, Euclidean, $L_1$, $L_\infty$ | SelfAttention + `MLP opt` | 0.52 |

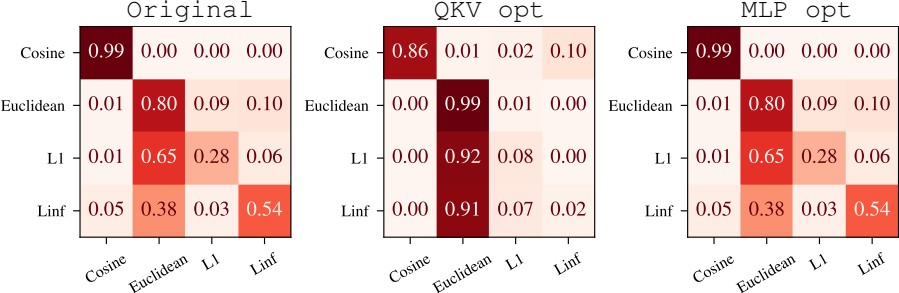

Figure 6: **Attention Weights Visualization.** Comparison of attention weights for the stitched model with `RexNet` as encoder and `ViT-B/16` as decoder on `CIFAR-100`, before and after fine-tuning. *(left)* the attention weights of the initial zero-shot stitched model, which remain unchanged when fine-tuning the classifier (`MLP opt`) *(right)*. Conversely, fine-tuning the QKV projections (`QKV opt`) *(center)* leads to a notable shift in attention weights, assigning lower importance to the space that performs worst individually.

**Takeaway.** Appropriate selection and aggregation of the optimal invariant spaces are crucial in enhancing latent communication between neural models.

## 5 CONCLUSION

In this paper, we introduced a framework to incorporate invariances into neural representations to enhance latent space communication without prior knowledge of the optimal invariance to be enforced. Constructing a product space with invariant components, we showed that it is possible to capture a large class of complex transformations between latent spaces within a single representation that is robust to multiple changing factors such as dataset, architecture, and training hyperparameter variations.

**Limitations and Future Works.** Our framework allows to incorporate multiple invariances in a single latent representation, induced by specific similarity functions. However, this can become limiting when a similarity function cannot be modeled analytically, or expressed in closed form. In such cases, an interesting direction would be to *learn* the similarity functions, paving the way to an even more extensive set of possible invariances to be enforced in the representation. Furthermore, while our method achieves good results on the tested benchmarks, we observed that selecting the optimal invariant spaces can yield equivalent performance using fewer components. Therefore, integrating a better space selection strategy holds promise for future research.

ACKNOWLEDGMENTS

The authors gratefully acknowledge the anonymous reviewers and the AC for the thoughtful remarks, and Luigi Gresele and Emanuele Marconato for the insightful discussions. This work is supported by the ERC grant no.802554 (SPECGEO), PRIN 2020 project no.2020TA3K9N (LEGO.AI), and PNRR MUR project PE0000013-FAIR.

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

# A  APPENDIX

## A.1  REPRODUCIBILITY STATEMENT

In Section Section 4, we provide a detailed description of the proposed framework and the experimental settings for the various scenarios. In the following sections, we present all implementation details that are not described in the main manuscript. Additionally, we are releasing a modular PyTorch implementation [3].

## A.2  DISTANCE-INDUCED INVARIANCES DETAILS

### A.2.1  DISTANCES DEFINITION

This section provides additional details about the metrics described in section Section 3.

**Cosine**. Given two vectors *u, v*, the cosine similarity is defined as:

$$\cos(u, v) = \frac{u \cdot v}{\|u\| \, \|v\|} \tag{2}$$

**Euclidean**. Given two vectors *u, v*, the Euclidean distance is defined as:

$$d(u, v) = \sqrt{\sum_{i=1}^{n} (u_i - v_i)^2} \tag{3}$$

**Manhattan** ($L_1$). Given two vectors *u, v*, the $L_1$ distance is defined as:

$$d(u, v) = \sum_{i=1}^{n} |(u_i - v_i)| \tag{4}$$

**Chebyshev** ($L_\infty$). Given two vectors *u, v*, the $L_\infty$ distance is defined as:

$$d(u, v) = \max_i(|u_i - y_i|) = \lim_{p \to \infty} \left( \sum_{i=1}^{n} |x_i - y_i|^p \right)^{\frac{1}{p}}, \tag{5}$$

and can be approximated in a differentiable way employing high values for $p$.

**Geodesic distance**. Given a manifold $\mathcal{M}$ and its parametrization $g : \mathcal{Z} \mapsto \mathcal{X}$ we can represent the Riemannian metric as symmetric, positive definite matrix $G(z)$ defined at each point in $Z$. $G(z)$ can be obtained as $G(z) = J_g(z)^T J_g(z)$, where $J_g(z)$ indicates the Jacobian of $g$ evaluated at $z$. This metric enables us to define an inner product on tangent spaces on $\mathcal{M}$. Considering a smooth curve $\gamma : [a, b] \mapsto \mathcal{Z}$, this corresponds to a curve on $\mathcal{M}$ via $g \circ \gamma(t)$. Its arc length is defined as:

$$L(\gamma) = \int_a^b \sqrt{\dot{\gamma}(t)^T G_{\gamma(t)} \dot{\gamma}(t) dt} \tag{6}$$

A *geodesic* curve is a curve that locally minimizes the arc length, corresponding to minimizing the following energy functional:

$$E(\gamma) = \frac{1}{2} \int_a^b \dot{\gamma}(t)^T G_{\gamma(t)} \dot{\gamma}(t) dt \tag{7}$$

In Figure 7, we show how geodesic distance is preserved under several classes of transformations, including manifold isometries, i.e., possibly nonlinear transformations that preserve the metric on $\mathcal{M}$. In the synthetic experiment, geodesic distances are computed using the heat method of Crane et al. (2017), and the manifold isometry is calculated using Isomap (Tenenbaum et al., 2000). Possible approaches to extend geodesic computation to real cases when $dim(\mathcal{Z}) > 3$ include Shao et al. (2017). We leave this promising direction for future work.

---

[3] https://github.com/icannistraci/latent-invariances

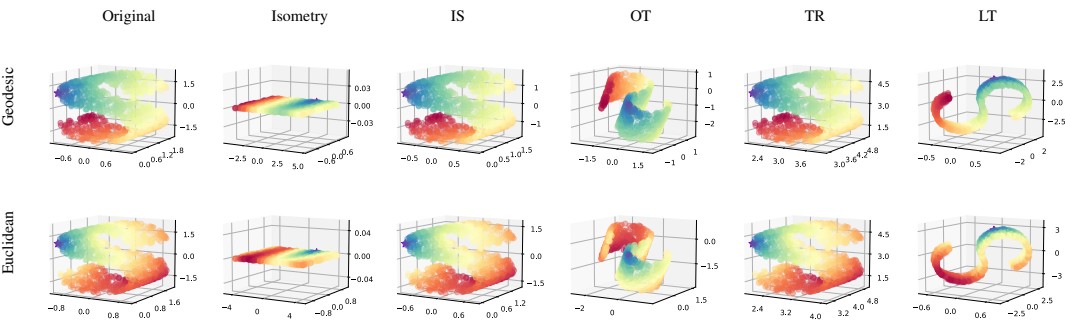

Figure 7: **Qualitative Synthetic Results**. We show invariances induced using a geodesic distance-based representation. We plot geodesic distances (*top row*) from the violet star point with values going from blue (closer) to red (farther). On the *bottom row*, we compare with Euclidean distances, showing that the latter does not estimate nor preserve well the metric information under transformations of the manifold.

### A.2.2 INFUSED INVARIANCES

In Table 5, we summarize the invariances guaranteed by different distance metrics concerning the following standard classes of transformations: Isotropic Scaling (IS), Orthogonal Transformation (OT), Translation (TR), Permutation (PT), Affine Transformation (AT), Linear Transformation (LT), and Manifold Isometry (MIS). Where MIS is an isometric deformation of the manifold that preserves the geodesic distances between points, see Figure 7 for a synthetic example. In general, capturing the set of invariances induced by a similarity function is not straightforward. For example, the $L_\infty$ distance does not enforce isometry invariance in the representation but, simultaneously, induces an invariance to perturbations in dimensions other than the maximum one. Formalizing and analyzing such types of invariances presents challenges since these transformations cannot be neatly classified into a specific simple class of transformations.

Table 5: **Invariances.** Overview of the different distance-induced invariances.

| Similarity Function | Isotropic Scaling | Orthogonal Transf. | Translation | Permutation | Affine Transf. | Linear Transf. | Manifold Isometry |
|---|---|---|---|---|---|---|---|
| Absolute | ✗ | ✗ | ✗ | ✗ | ✗ | ✗ | ✗ |
| Cosine | ✓ | ✓ | ✗ | ✓ | ✗ | ✗ | ✗ |
| Euclidean | ✗ | ✓ | ✓ | ✓ | ✗ | ✗ | ✗ |
| Manhattan | ✗ | ✗ | ✓ | ✓ | ✗ | ✗ | ✗ |
| Chebyshev | ✗ | ✗ | ✓ | ✓ | ✗ | ✗ | ✗ |
| Geodesic | ✓ | ✓ | ✓ | ✓ | ✗ | ✗ | ✓ |

### A.3 AGGREGATION FUNCTIONS

In this section, we report the implementation details of the aggregation functions $\phi$ described in Section 3. There are two possible preprocessing strategies applied to each subspace independently:

- *Normalization layer*: an independent LayerNorm for each subspace.

- *MLP*: a compact, independent, fully connected network defined for each subspace, comprised of LayerNorm, a Linear layer, and a Tanh activation function.

These preprocessing modules can be applied before either the (Sum) or (Self-attention) aggregation strategies.

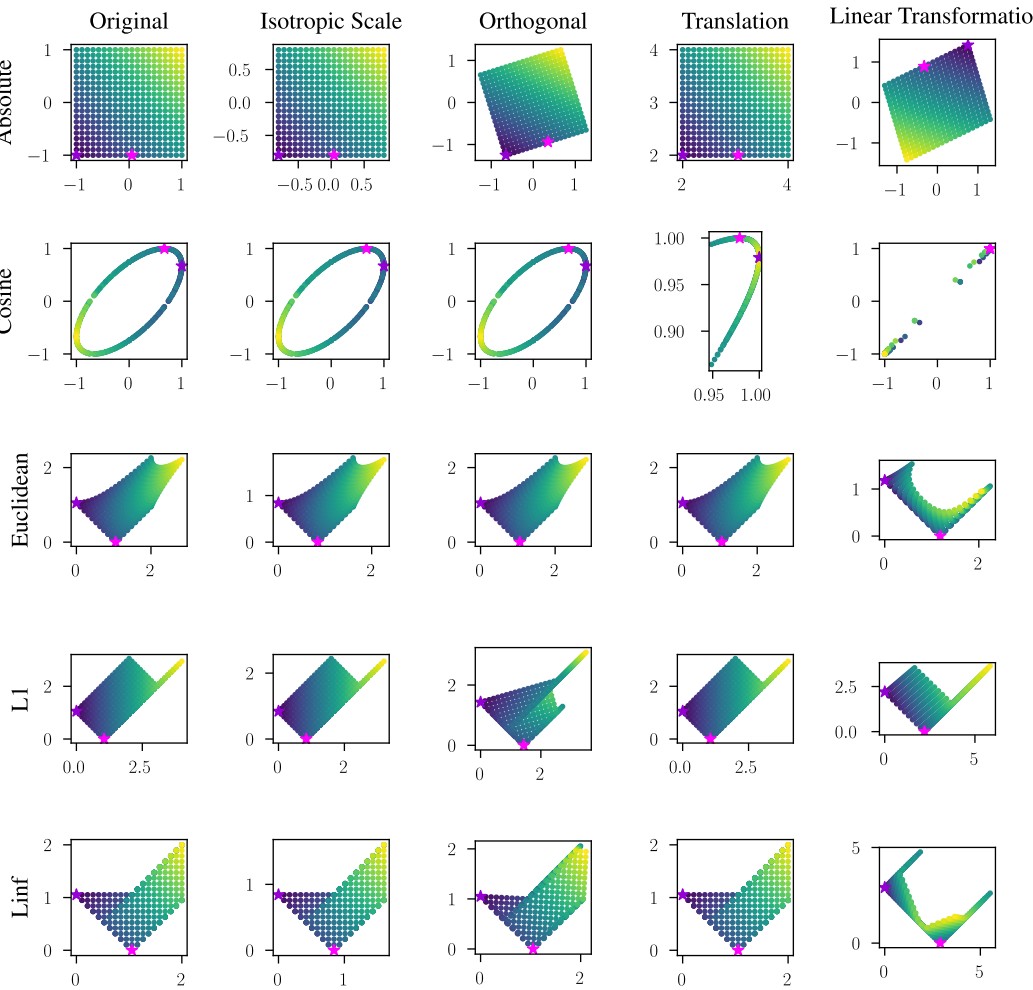

Figure 8: **Qualitative synthetic experiments using a grid initialization.** We consider a synthetic absolute latent space by initializing points in a grid shape *(top left)*. We then apply various transformations to the absolute space, converting it into different transformed spaces *(first row)*. We convert the entire first row into the corresponding relative space for all the different similarity functions considered (i.e. Cosine, Euclidean, $L_1$, and $L_\infty$). Observing which transformation does not change the original relative space *(left column)*, shows which projections induce an invariance to each considered transformation.

## A.4 IMPLEMENTATION DETAILS

This section details the experiments conducted in Section 4. Table 6 contains the full list of the pretrained models, while Table 7 contains dataset information.

### A.4.1 TOOLS & TECHNOLOGIES

All the experiments presented in this work employ the following tools:

- *PyTorch Lightning*, to ensure reproducible results while also getting a clean and modular codebase;
- *NN-Template GrokAI (2021)*, to easily bootstrap the project and enforce best practices;
- *Transformers by HuggingFace*, to get ready-to-use transformers for both text and images;
- *Datasets by HuggingFace*, to access most of the datasets;
- *DVC* (Kuprieiev et al., 2022), for data versioning;

Table 6: **Pretrained models details.** Details of the pretrained feature extractors with their Hugging-Face key, their alias, and their latent space dimensionality.

| Modality | HuggingFace model name | Alias | Enc. Dim |
|---|---|---|---|
| Language | bert-base-cased | `BERT-C` (Devlin et al., 2019) | 768 |
| | bert-base-uncased | `BERT-U` (Devlin et al., 2019) | 768 |
| | google/electra-base-discriminator | `ELECTRA` (Clark et al., 2020) | 768 |
| | roberta-base | `RoBERTa` (Liu et al., 2019) | 768 |
| | albert-base-v2 | `ALBERT` (Lan et al., 2019) | 768 |
| | xlm-roberta-base | `XLM-R` (Conneau et al., 2019) | 768 |
| | openai/clip-vit-base-patch32 | `CViT-B/32` (Radford et al., 2021) | 768 |
| Vision | rexnet_100 | `RexNet` (Han et al., 2020) | 1280 |
| | vit_small_patch16_224 | `ViT-S/16` (Dosovitskiy et al., 2021) | 384 |
| | vit_base_patch16_384 | `ViT-B/16` (Dosovitskiy et al., 2021) | 768 |
| | vit_base_resnet50_384 | `RViT-B/16` (Dosovitskiy et al., 2021) | 768 |
| | openai/clip-vit-base-patch32 | `CViT-B/32` (Radford et al., 2021) | 768 |
| Graph | `GCN` | `GCN` | 300 |

Table 7: **Dataset details.** Details of the HuggingFace datasets used in the classification and reconstruction experiments, with the associated number of classes.

| Modality | Name | Number of Classes |
|---|---|---|
| Image | `MNIST` | 10 |
| | `Fashion MNIST` | 10 |
| | `CIFAR-10` | 10 |
| | `CIFAR-100` | 20 (coarse) — 100 (fine) |
| | `ImageNet1k` | 1000 |
| Text | `TREC` | 6 (coarse) — 50 (fine) |
| | `DBpedia` | 14 |
| | `N24news(Text)` | 24 |
| Graph | `CORA` | 7 |

## A.5 LATENT SPACE ANALYSIS

### RECONSTRUCTION

This experiment adopts convolution-based AE and VAE. The design variations encompass 2D-bottleneck architectures with a dimensionality of $16 \times 7 \times 7$ and 1D-bottleneck architectures with a dimensionality of $784 = 16 \times 7 \times 7$ for fair comparison. The models with 2D bottlenecks are endowed with approximately 50k parameters (AE) and 60k parameters (VAE), while their linearized counterparts possess 1.3 million parameters (LinAE) and 1.9 million parameters (LinVAE). Intriguingly, the variants preserving spatial structure in the latent space demonstrate superior performance despite having significantly fewer parameters. All models undergo training using the Adam optimizer (Kingma & Ba, 2015) with a learning rate set to $1e{-3}$. Early stopping is employed based on validation reconstruction error, quantified by mean squared error.

## A.6 SPACE SELECTION

In Table 8 and Figure 9, we present similar results to the ones reported in Section 4.4, but using a more expressive classifier. This choice establishes a less favorable scenario since we are fine-tuning more parameters in the *SelfAttention+MLP opt* setting. It illustrates that is still better to fine-tune the QKV rather than the whole classifier.

Table 8: **Optimal Space Selection.** Classification accuracy for the stitched model with `RexNet` as encoder and `ViT-B/16` as decoder on `CIFAR-100`, using different projection functions and aggregation strategies. Fine-tuning the subspace selection and blending part (`QKV opt`) has a more significant effect on performance improvement than fine-tuning only the larger MLP (`MLP opt`).

| Projection | Aggregation | Accuracy ↑ |
|---|---|---|
| Cosine | - | **0.52** |
| Euclidean | - | 0.42 |
| $L_1$ | - | 0.34 |
| $L_\infty$ | - | 0.22 |
| Cosine, Euclidean, $L_1$, $L_\infty$ | SelfAttention | 0.25 |
| Cosine, Euclidean, $L_1$, $L_\infty$ | MLP+Sum | **0.43** |
| Cosine, Euclidean, $L_1$, $L_\infty$ | SelfAttention + `QKV opt` | **0.75** |
| Cosine, Euclidean, $L_1$, $L_\infty$ | SelfAttention + `MLP opt` | 0.65 |

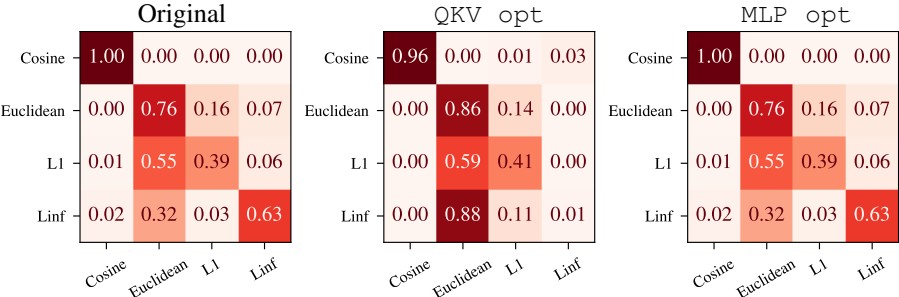

Figure 9: **Attention Weights Visualization.** Comparison of attention weights for the stitched model between `RexNet` and `ViT-B/16` on `CIFAR-100`, before and after fine-tuning. *(left)* the attention weights of the initial zero-shot stitched model, which remain unchanged when fine-tuning the classifier (`MLP opt`) *(right)*. Conversely, fine-tuning the QKV projections (`WKV opt`) *(center)* leads to a notable shift in attention weights, assigning greater importance to the subspace that performs better individually.

## A.7 ANCHOR SELECTION ANALYSIS

In this section, we present the analysis performed on the anchor choice. In Section 4.2 when referring to random but fixed anchors, we mean that the anchors are uniformly randomly sampled from the training set, but with a fixed seed value. Thus, we conducted an analysis on the number of randomly selected anchors for the stitching task using `CIFAR-100` across three different anchor selection seeds. The results in Table 9 and Figure 10 reveal that varying the number of anchors leads to different transformations, indicating that a single projection function cannot incorporate the desired invariance. In contrast, our method enables the attainment of the highest score regardless of the number of anchors or the seed value employed for the uniform random sampling of anchors.

## A.8 STITCHING PROCEDURE

In Figure 11, we illustrate in detail the stitching procedure introduced in Section 4.2.

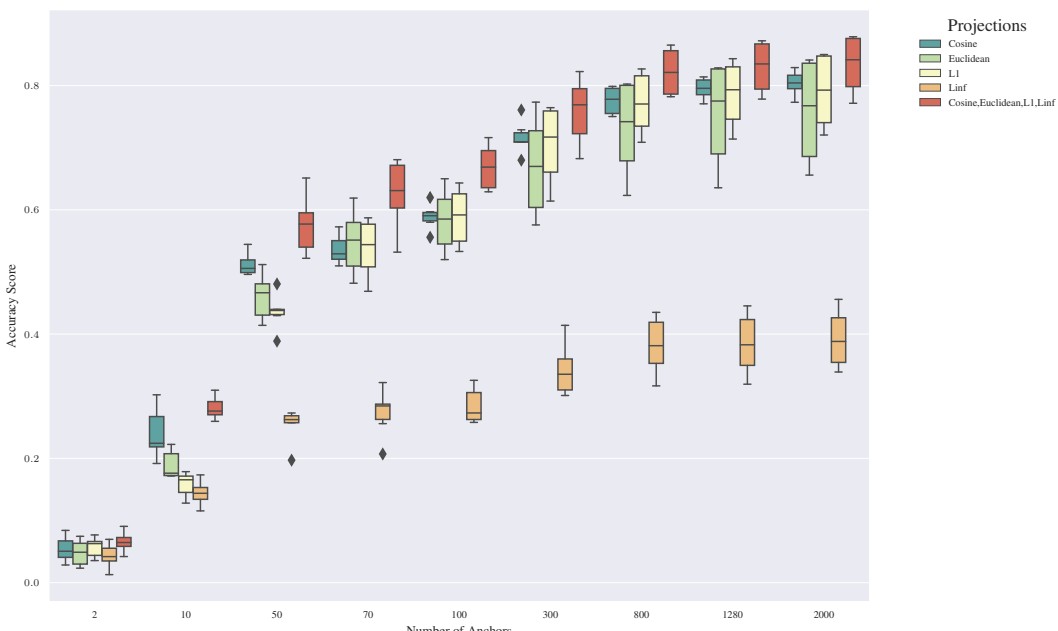

Figure 10: **Accuracy vs. Number of Anchors.** Each box represents a stitched model using `CViT-B/32` and `ViT-B/16` on `CIFAR-100`. The proposed methodology using the *Aggregation by sum* consistently outperforms other results regardless of the number of anchors.

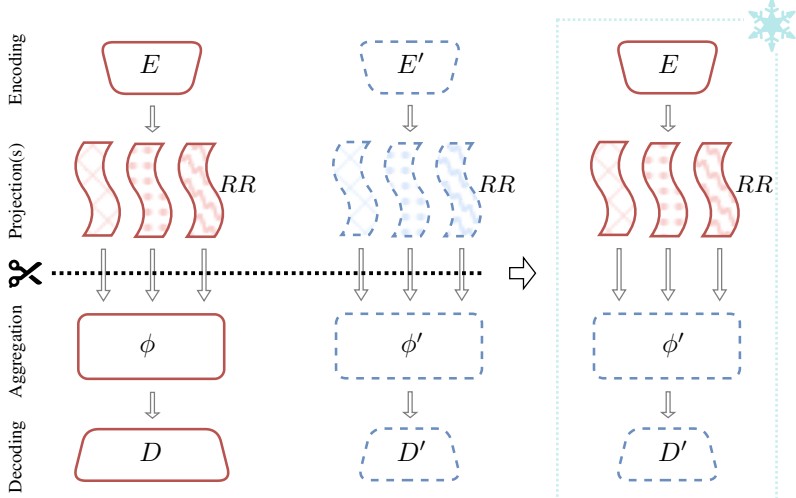

Figure 11: **Stitching Procedure Description**. Given two trained models (*left* and *center* columns), the zero-shot stitched model (*right* column) is formed by combining the encoder of the first model with the decoder of the second one. The zero-shot stitched model *does not require additional training or fine-tuning*. Although certain components of our module have learnable parameters (e.g., $\phi$), these parameters are exclusively trained during their respective network training phase and are subsequently utilized in their trained, frozen state in the stitched model.

Table 9: **Number of Anchors Ablation.** Zero-shot stitching classification performance results using `CViT-B/32` and `ViT-B/16` on `CIFAR-100`. The proposed methodologies utilizing the *Aggregation by sum* consistently outperform the other results regardless of the number of anchors.

| Number of Anchors ↓ | Projection | Accuracy ↑ |
|---|---|---|
| 2 | Cosine | $0.05 \pm 0.02$ |
| | Euclidean | $0.05 \pm 0.02$ |
| | $L_1$ | $\mathbf{0.06} \pm 0.02$ |
| | $L_\infty$ | $0.04 \pm 0.02$ |
| | Cosine, Euclidean, $L_1$, $L_\infty$ | $\mathbf{0.07} \pm 0.02$ |
| 10 | Cosine | $\mathbf{0.24} \pm 0.04$ |
| | Euclidean | $0.19 \pm 0.02$ |
| | $L_1$ | $0.16 \pm 0.02$ |
| | $L_\infty$ | $0.14 \pm 0.02$ |
| | Cosine, Euclidean, $L_1$, $L_\infty$ | $\mathbf{0.28} \pm 0.02$ |
| 50 | Cosine | $\mathbf{0.51} \pm 0.02$ |
| | Euclidean | $0.46 \pm 0.04$ |
| | $L_1$ | $0.44 \pm 0.03$ |
| | $L_\infty$ | $0.25 \pm 0.03$ |
| | Cosine, Euclidean, $L_1$, $L_\infty$ | $\mathbf{0.58} \pm 0.05$ |
| 70 | Cosine | $0.54 \pm 0.02$ |
| | Euclidean | $\mathbf{0.55} \pm 0.05$ |
| | $L_1$ | $0.54 \pm 0.05$ |
| | $L_\infty$ | $0.27 \pm 0.04$ |
| | Cosine, Euclidean, $L_1$, $L_\infty$ | $\mathbf{0.63} \pm 0.06$ |
| 100 | Cosine | $\mathbf{0.59} \pm 0.02$ |
| | Euclidean | $0.58 \pm 0.05$ |
| | $L_1$ | $\mathbf{0.59} \pm 0.05$ |
| | $L_\infty$ | $0.28 \pm 0.03$ |
| | Cosine, Euclidean, $L_1$, $L_\infty$ | $\mathbf{0.67} \pm 0.04$ |
| 300 | Cosine | $\mathbf{0.72} \pm 0.03$ |
| | Euclidean | $0.67 \pm 0.08$ |
| | $L_1$ | $0.70 \pm 0.06$ |
| | $L_\infty$ | $0.34 \pm 0.04$ |
| | Cosine, Euclidean, $L_1$, $L_\infty$ | $\mathbf{0.76} \pm 0.05$ |
| 800 | Cosine | $\mathbf{0.78} \pm 0.02$ |
| | Euclidean | $0.73 \pm 0.08$ |
| | $L_1$ | $0.77 \pm 0.05$ |
| | $L_\infty$ | $0.38 \pm 0.05$ |
| | Cosine, Euclidean, $L_1$, $L_\infty$ | $\mathbf{0.82} \pm 0.04$ |
| 1280 | Cosine | $\mathbf{0.80} \pm 0.02$ |
| | Euclidean | $0.75 \pm 0.09$ |
| | $L_1$ | $0.79 \pm 0.05$ |
| | $L_\infty$ | $0.38 \pm 0.05$ |
| | Cosine, Euclidean, $L_1$, $L_\infty$ | $\mathbf{0.83} \pm 0.04$ |
| 2000 | Cosine | $\mathbf{0.80} \pm 0.02$ |
| | Euclidean | $0.76 \pm 0.09$ |
| | $L_1$ | $0.79 \pm 0.06$ |
| | $L_\infty$ | $0.39 \pm 0.05$ |
| | Cosine, Euclidean, $L_1$, $L_\infty$ | $\mathbf{0.83} \pm 0.05$ |

## A.9    ADDITIONAL RESULTS

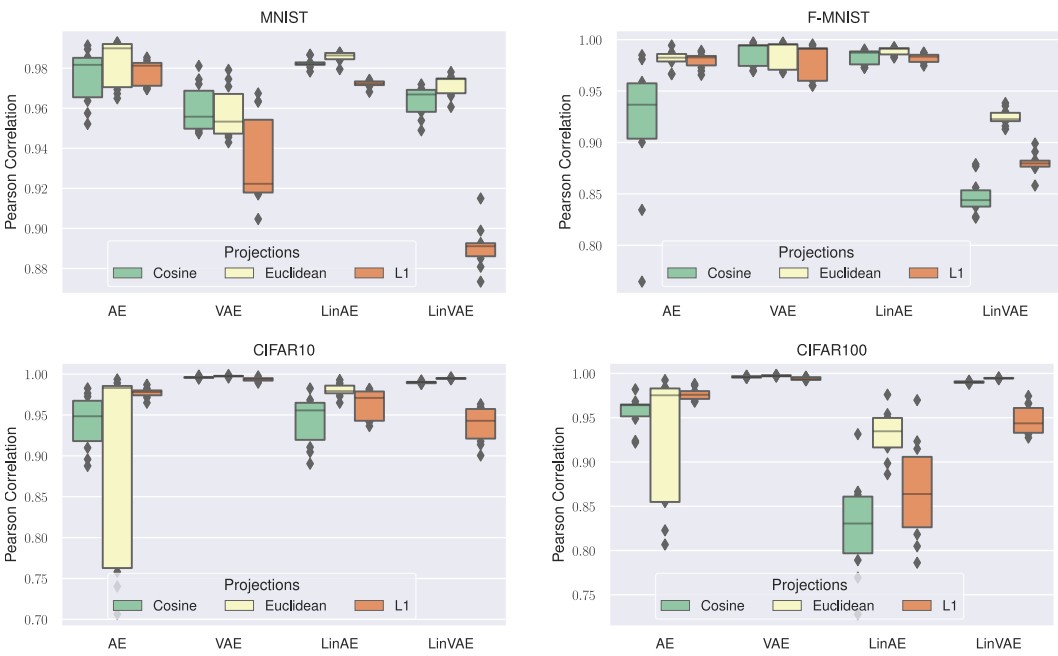

Figure 12: **Latent Spaces Cross-Seed Similarity.** Pearson correlation, measuring the similarity of latent spaces across different seeds for various architectures and datasets.

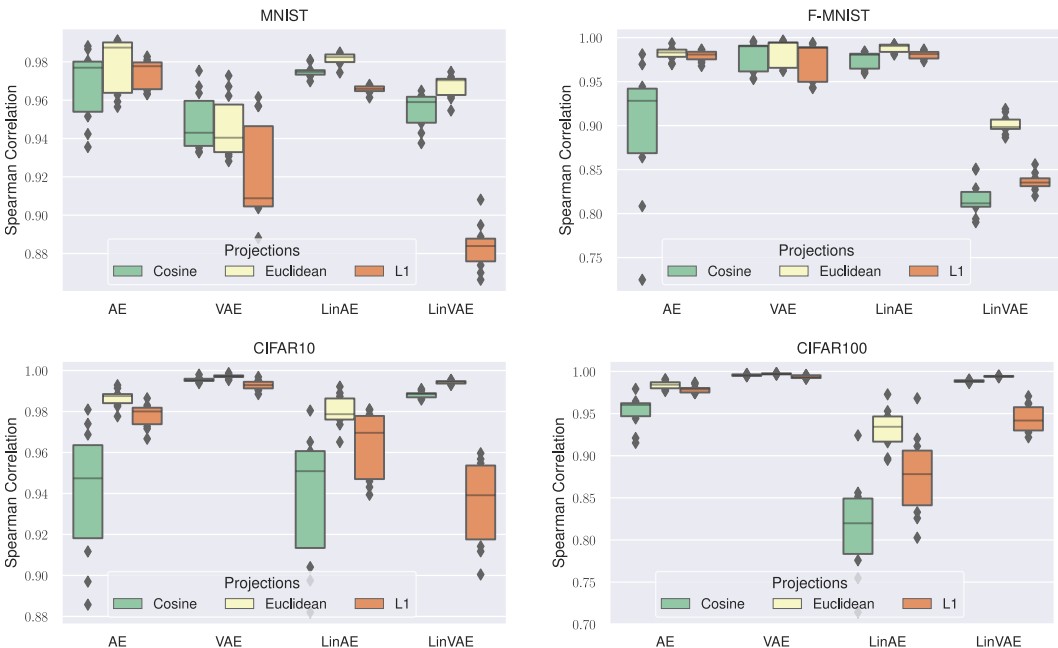

Figure 13: **Latent Spaces Cross-Seed Similarity.** Spearman correlation, measuring the similarity of latent spaces across different seeds for various architectures and datasets.

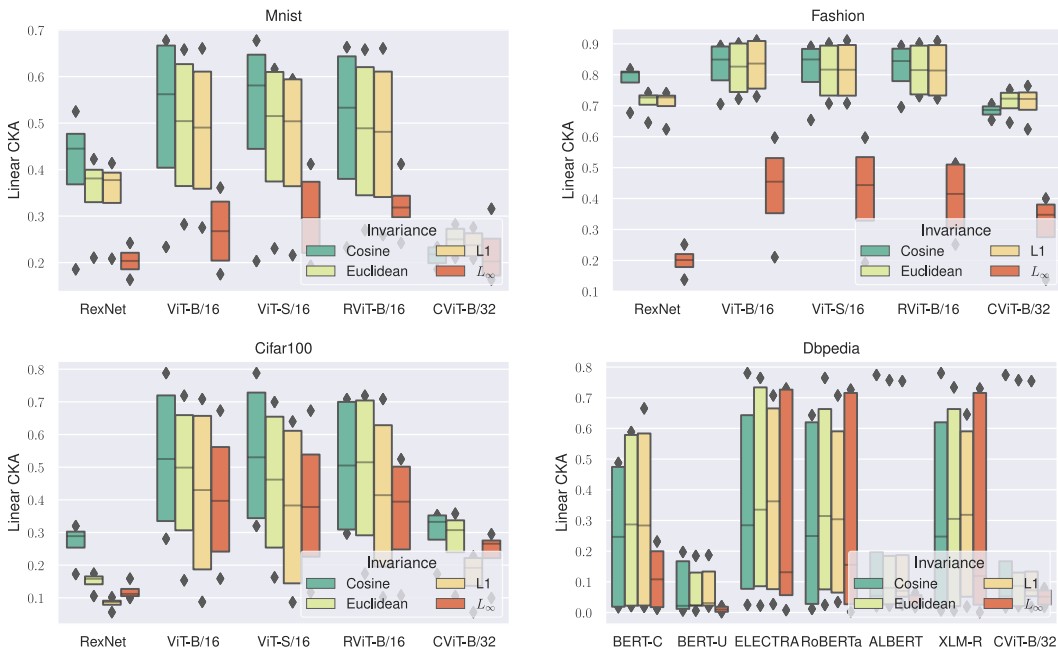

Figure 14: **Latent Spaces Cross-Architecture Similarity.** Linear CKA, measuring the similarity of latent spaces of pretrained models across different architectures and datasets.

Table 10: **Image Stitching Performance Cross-Architecture and Cross-Seed.** Zero-shot accuracy score for image classification task across different pretrained models, seeds, and datasets. The proposed method using the *Aggregation by sum* consistently achieves the highest accuracy score or comparable results, even without prior knowledge of the optimal projection to employ.

| Encoder | Projection | Accuracy ↑ | | | |
|---|---|---|---|---|---|
| | | CIFAR-100 | CIFAR-10 | MNIST | Fashion MNIST |
| CViT-B/32 | Cosine | $0.52 \pm 0.03$ | $\mathbf{0.87} \pm 0.02$ | $0.61 \pm 0.06$ | $0.68 \pm 0.02$ |
| | Euclidean | $0.53 \pm 0.02$ | $\mathbf{0.87} \pm 0.02$ | $\mathbf{0.66} \pm 0.05$ | $\mathbf{0.70} \pm 0.03$ |
| | $L_1$ | $\mathbf{0.53} \pm 0.04$ | $\mathbf{0.87} \pm 0.02$ | $\mathbf{0.66} \pm 0.05$ | $\mathbf{0.70} \pm 0.03$ |
| | $L_\infty$ | $0.27 \pm 0.04$ | $0.52 \pm 0.04$ | $0.57 \pm 0.03$ | $0.55 \pm 0.01$ |
| | Cosine, Euclidean, $L_1$, $L_\infty$ | $\mathbf{0.58} \pm 0.03$ | $\mathbf{0.88} \pm 0.02$ | $\mathbf{0.68} \pm 0.05$ | $\mathbf{0.70} \pm 0.01$ |
| RViT-B/16 | Cosine | $\mathbf{0.79} \pm 0.03$ | $0.94 \pm 0.01$ | $0.69 \pm 0.04$ | $0.76 \pm 0.03$ |
| | Euclidean | $\mathbf{0.79} \pm 0.03$ | $0.94 \pm 0.01$ | $\mathbf{0.71} \pm 0.04$ | $0.77 \pm 0.03$ |
| | $L_1$ | $0.77 \pm 0.04$ | $\mathbf{0.95} \pm 0.01$ | $\mathbf{0.71} \pm 0.04$ | $\mathbf{0.79} \pm 0.03$ |
| | $L_\infty$ | $0.31 \pm 0.03$ | $0.75 \pm 0.04$ | $0.61 \pm 0.05$ | $0.60 \pm 0.03$ |
| | Cosine, Euclidean, $L_1$, $L_\infty$ | $\mathbf{0.81} \pm 0.04$ | $\mathbf{0.95} \pm 0.01$ | $\mathbf{0.72} \pm 0.04$ | $0.76 \pm 0.04$ |
| RexNet | Cosine | $\mathbf{0.50} \pm 0.02$ | $\mathbf{0.79} \pm 0.01$ | $\mathbf{0.71} \pm 0.02$ | $0.74 \pm 0.01$ |
| | Euclidean | $0.43 \pm 0.06$ | $0.72 \pm 0.02$ | $0.69 \pm 0.04$ | $\mathbf{0.76} \pm 0.01$ |
| | $L_1$ | $0.33 \pm 0.06$ | $0.70 \pm 0.01$ | $0.69 \pm 0.04$ | $\mathbf{0.76} \pm 0.02$ |
| | $L_\infty$ | $0.19 \pm 0.03$ | $0.48 \pm 0.03$ | $0.48 \pm 0.02$ | $0.60 \pm 0.05$ |
| | Cosine, Euclidean, $L_1$, $L_\infty$ | $\mathbf{0.52} \pm 0.05$ | $0.75 \pm 0.01$ | $0.70 \pm 0.03$ | $0.75 \pm 0.02$ |
| ViT-B/16 | Cosine | $0.75 \pm 0.05$ | $\mathbf{0.96} \pm 0.01$ | $0.59 \pm 0.05$ | $0.79 \pm 0.03$ |
| | Euclidean | $\mathbf{0.76} \pm 0.05$ | $\mathbf{0.96} \pm 0.01$ | $0.65 \pm 0.06$ | $\mathbf{0.81} \pm 0.02$ |
| | $L_1$ | $\mathbf{0.76} \pm 0.06$ | $\mathbf{0.96} \pm 0.01$ | $\mathbf{0.66} \pm 0.07$ | $\mathbf{0.81} \pm 0.02$ |
| | $L_\infty$ | $0.42 \pm 0.02$ | $0.70 \pm 0.05$ | $0.42 \pm 0.05$ | $0.52 \pm 0.04$ |
| | Cosine, Euclidean, $L_1$, $L_\infty$ | $\mathbf{0.81} \pm 0.05$ | $\mathbf{0.96} \pm 0.01$ | $\mathbf{0.66} \pm 0.04$ | $0.80 \pm 0.04$ |
| ViT-S/32 | Cosine | $\mathbf{0.73} \pm 0.04$ | $\mathbf{0.93} \pm 0.01$ | $0.68 \pm 0.04$ | $0.77 \pm 0.02$ |
| | Euclidean | $0.64 \pm 0.03$ | $\mathbf{0.93} \pm 0.01$ | $\mathbf{0.70} \pm 0.02$ | $0.77 \pm 0.02$ |
| | $L_1$ | $0.58 \pm 0.09$ | $\mathbf{0.93} \pm 0.01$ | $\mathbf{0.70} \pm 0.03$ | $\mathbf{0.78} \pm 0.02$ |
| | $L_\infty$ | $0.33 \pm 0.04$ | $0.69 \pm 0.03$ | $0.48 \pm 0.02$ | $0.53 \pm 0.02$ |
| | Cosine, Euclidean, $L_1$, $L_\infty$ | $\mathbf{0.73} \pm 0.05$ | $\mathbf{0.93} \pm 0.01$ | $0.69 \pm 0.02$ | $0.76 \pm 0.02$ |

Table 11: **Text Stitching Performance Cross-Architecture and Cross-Seed.** Zero-shot accuracy score for text classification task across different pretrained models, seeds, and datasets. The aggregation function is the *Aggregation by sum* and the results are obtained using a linear classifier as a decoder, instead of a MLP as in Moschella et al. (2023).

| Encoder | Projection | Accuracy ↑ | | |
|---|---|---|---|---|
| | | DBpedia | TREC | N24news(Text) |
| ALBERT | Cosine | $0.48 \pm 0.05$ | $0.49 \pm 0.08$ | $\mathbf{0.09} \pm 0.02$ |
| | Euclidean | $0.49 \pm 0.05$ | $0.54 \pm 0.06$ | $\mathbf{0.09} \pm 0.03$ |
| | $L_1$ | $\mathbf{0.51} \pm 0.04$ | $\mathbf{0.59} \pm 0.06$ | $\mathbf{0.09} \pm 0.03$ |
| | $L_\infty$ | $0.17 \pm 0.02$ | $0.32 \pm 0.07$ | $0.07 \pm 0.01$ |
| | Cosine, Euclidean, $L_1$, $L_\infty$ | $0.50 \pm 0.06$ | $0.55 \pm 0.06$ | $\mathbf{0.09} \pm 0.03$ |
| BERT-C | Cosine | $0.46 \pm 0.07$ | $0.46 \pm 0.11$ | $0.21 \pm 0.09$ |
| | Euclidean | $0.48 \pm 0.09$ | $0.57 \pm 0.05$ | $\mathbf{0.22} \pm 0.09$ |
| | $L_1$ | $\mathbf{0.50} \pm 0.10$ | $\mathbf{0.59} \pm 0.06$ | $0.21 \pm 0.09$ |
| | $L_\infty$ | $0.13 \pm 0.04$ | $0.21 \pm 0.04$ | $0.11 \pm 0.04$ |
| | Cosine, Euclidean, $L_1$, $L_\infty$ | $\mathbf{0.50} \pm 0.13$ | $0.54 \pm 0.07$ | $0.21 \pm 0.09$ |
| BERT-U | Cosine | $\mathbf{0.51} \pm 0.05$ | $0.46 \pm 0.12$ | $0.15 \pm 0.06$ |
| | Euclidean | $0.36 \pm 0.05$ | $0.56 \pm 0.07$ | $\mathbf{0.17} \pm 0.06$ |
| | $L_1$ | $0.36 \pm 0.06$ | $\mathbf{0.58} \pm 0.09$ | $0.16 \pm 0.06$ |
| | $L_\infty$ | $0.12 \pm 0.02$ | $0.28 \pm 0.07$ | $0.06 \pm 0.01$ |
| | Cosine, Euclidean, $L_1$, $L_\infty$ | $0.43 \pm 0.07$ | $0.51 \pm 0.10$ | $\mathbf{0.17} \pm 0.07$ |
| CViT-B/32 | Cosine | $0.20 \pm 0.02$ | $0.50 \pm 0.04$ | $0.08 \pm 0.03$ |
| | Euclidean | $\mathbf{0.22} \pm 0.02$ | $0.48 \pm 0.10$ | $\mathbf{0.12} \pm 0.05$ |
| | $L_1$ | $\mathbf{0.22} \pm 0.02$ | $\mathbf{0.57} \pm 0.07$ | $0.11 \pm 0.03$ |
| | $L_\infty$ | $0.12 \pm 0.02$ | $0.25 \pm 0.09$ | $0.07 \pm 0.02$ |
| | Cosine, Euclidean, $L_1$, $L_\infty$ | $\mathbf{0.22} \pm 0.01$ | $0.50 \pm 0.09$ | $0.10 \pm 0.03$ |
| ELECTRA | Cosine | $0.27 \pm 0.05$ | $0.39 \pm 0.11$ | $0.04 \pm 0.01$ |
| | Euclidean | $0.25 \pm 0.05$ | $\mathbf{0.53} \pm 0.05$ | $0.04 \pm 0.01$ |
| | $L_1$ | $\mathbf{0.33} \pm 0.06$ | $0.50 \pm 0.10$ | $\mathbf{0.05} \pm 0.01$ |
| | $L_\infty$ | $0.11 \pm 0.02$ | $0.26 \pm 0.11$ | $\mathbf{0.05} \pm 0.01$ |
| | Cosine, Euclidean, $L_1$, $L_\infty$ | $0.32 \pm 0.07$ | $0.52 \pm 0.14$ | $0.04 \pm 0.01$ |
| RoBERTa | Cosine | $\mathbf{0.31} \pm 0.07$ | $0.43 \pm 0.05$ | $0.41 \pm 0.18$ |
| | Euclidean | $0.26 \pm 0.05$ | $0.52 \pm 0.11$ | $0.39 \pm 0.18$ |
| | $L_1$ | $0.30 \pm 0.06$ | $\mathbf{0.56} \pm 0.07$ | $\mathbf{0.45} \pm 0.20$ |
| | $L_\infty$ | $0.13 \pm 0.01$ | $0.20 \pm 0.07$ | $0.08 \pm 0.02$ |
| | Cosine, Euclidean, $L_1$, $L_\infty$ | $\mathbf{0.34} \pm 0.08$ | $0.53 \pm 0.09$ | $0.43 \pm 0.20$ |
| XLM-R | Cosine | $0.13 \pm 0.03$ | $0.39 \pm 0.10$ | $0.20 \pm 0.10$ |
| | Euclidean | $0.16 \pm 0.03$ | $0.53 \pm 0.06$ | $0.26 \pm 0.11$ |
| | $L_1$ | $\mathbf{0.32} \pm 0.06$ | $\mathbf{0.61} \pm 0.10$ | $\mathbf{0.32} \pm 0.14$ |
| | $L_\infty$ | $0.08 \pm 0.01$ | $0.30 \pm 0.08$ | $0.07 \pm 0.01$ |
| | Cosine, Euclidean, $L_1$, $L_\infty$ | $0.21 \pm 0.03$ | $0.55 \pm 0.07$ | $0.27 \pm 0.14$ |

Table 12: **Image Classification Ablation on the Aggregation Functions.** Zero-shot accuracy score across different architectures, seeds, and datasets. The proposed aggregation function obtains the best accuracy score. It is essential to highlight that the Concat aggregation method is not directly comparable to the others because it increases the dimensionality of the space linearly with the number of subspaces. The standard deviation is high because it accounts for the average across all possible image datasets.

| Encoder | Aggregation Function | Accuracy ↑ |
|---|---|---|
| CViT-B/32 | Concat* | $0.68 \pm 0.13$ |
| | MLP+SelfAttention | $0.66 \pm 0.13$ |
| | MLP+Sum | $\mathbf{0.71} \pm 0.11$ |
| | SelfAttention | $0.54 \pm 0.18$ |
| RViT-B/16 | Concat* | $0.72 \pm 0.18$ |
| | MLP+SelfAttention | $0.72 \pm 0.18$ |
| | MLP+Sum | $\mathbf{0.74} \pm 0.18$ |
| | SelfAttention | $0.61 \pm 0.24$ |
| RexNet | Concat* | $0.58 \pm 0.20$ |
| | MLP+SelfAttention | $0.54 \pm 0.21$ |
| | MLP+Sum | $\mathbf{0.61} \pm 0.20$ |
| | SelfAttention | $0.40 \pm 0.22$ |
| ViT-B/16 | Concat* | $0.72 \pm 0.19$ |
| | MLP+SelfAttention | $0.70 \pm 0.20$ |
| | MLP+Sum | $\mathbf{0.74} \pm 0.19$ |
| | SelfAttention | $0.60 \pm 0.24$ |
| ViT-S/16 | Concat* | $0.69 \pm 0.18$ |
| | MLP+SelfAttention | $0.67 \pm 0.19$ |
| | MLP+Sum | $\mathbf{0.71} \pm 0.18$ |
| | SelfAttention | $0.57 \pm 0.22$ |

Table 13: **Text Classification Ablation on the Aggregation Functions.** Zero-shot accuracy score across different architectures, seeds, and datasets. The proposed aggregation function obtains the best accuracy score. It is essential to highlight that the Concat aggregation method is not directly comparable to the others because it increases the dimensionality of the space linearly with the number of subspaces. The standard deviation is high because it accounts for the average across all possible text datasets.

| Encoder | Aggregation Function | Accuracy ↑ |
|---|---|---|
| ALBERT | Concat* | **0.38** $\pm$ 0.22 |
| | MLP+SelfAttention | 0.35 $\pm$ 0.21 |
| | MLP+Sum | **0.38** $\pm$ 0.21 |
| | SelfAttention | 0.21 $\pm$ 0.17 |
| BERT-C | Concat* | 0.41 $\pm$ 0.17 |
| | MLP+SelfAttention | 0.36 $\pm$ 0.17 |
| | MLP+Sum | **0.42** $\pm$ 0.18 |
| | SelfAttention | 0.20 $\pm$ 0.16 |
| BERT-U | Concat* | 0.36 $\pm$ 0.17 |
| | MLP+SelfAttention | 0.31 $\pm$ 0.15 |
| | MLP+Sum | **0.37** $\pm$ 0.17 |
| | SelfAttention | 0.19 $\pm$ 0.15 |
| CViT-B/32 | Concat* | 0.23 $\pm$ 0.18 |
| | MLP+SelfAttention | 0.22 $\pm$ 0.18 |
| | MLP+Sum | **0.26** $\pm$ 0.21 |
| | SelfAttention | 0.15 $\pm$ 0.13 |
| ELECTRA | Concat* | **0.30** $\pm$ 0.18 |
| | MLP+SelfAttention | 0.27 $\pm$ 0.16 |
| | MLP+Sum | **0.30** $\pm$ 0.18 |
| | SelfAttention | 0.14 $\pm$ 0.11 |
| RoBERTa | Concat* | 0.41 $\pm$ 0.15 |
| | MLP+SelfAttention | 0.36 $\pm$ 0.14 |
| | MLP+Sum | **0.44** $\pm$ 0.15 |
| | SelfAttention | 0.23 $\pm$ 0.16 |
| XLM-R | Concat* | **0.35** $\pm$ 0.17 |
| | MLP+SelfAttention | 0.29 $\pm$ 0.17 |
| | MLP+Sum | **0.35** $\pm$ 0.17 |
| | SelfAttention | 0.18 $\pm$ 0.16 |

Table 14: **Graph Classification Ablation Study.** Zero-shot accuracy score across different architectures and seeds. We analyzed the effect of selecting only specific spaces and including all available spaces. Furthermore, we assessed all the possible aggregation functions and calculated the stitching index to validate stitching performance. Our proposed aggregation function achieved the highest accuracy score. It is important to note that the Concat aggregation method cannot be directly compared to others as it linearly increases the dimensionality of the space with the number of spaces.

| Aggregation | Projection | Accuracy ↑ | Stitching index ↑ |
|---|---|---|---|
| - | Absolute | $0.14 \pm 0.04$ | 0.18 |
| | Cosine | $\mathbf{0.53} \pm 0.06$ | 0.71 |
| | Euclidean | $0.27 \pm 0.06$ | 0.58 |
| | $L_1$ | $0.26 \pm 0.06$ | 0.58 |
| | $L_\infty$ | $0.12 \pm 0.03$ | **1.00** |
| Concat* | Cosine,Euclidean | $0.69 \pm 0.04$ | 0.90 |
| | Cosine, $L_1$ | $0.69 \pm 0.04$ | 0.90 |
| | Cosine,$L_\infty$ | $0.65 \pm 0.05$ | 0.87 |
| | Euclidean,$L_1$ | $0.40 \pm 0.07$ | 0.72 |
| | Euclidean,$L_\infty$ | $0.30 \pm 0.09$ | 0.65 |
| | $L_1,L_\infty$ | $0.32 \pm 0.09$ | 0.67 |
| | Cosine, Euclidean, $L_1, L_\infty$ | $\mathbf{0.75} \pm 0.02$ | **0.97** |
| SelfAttention | Cosine,Euclidean | $0.72 \pm 0.04$ | 0.96 |
| | Cosine,$L_1$ | $0.72 \pm 0.04$ | 0.96 |
| | Cosine,$L_\infty$ | $\mathbf{0.76} \pm 0.02$ | **1.00** |
| | Euclidean,$L_1$ | $0.68 \pm 0.03$ | 0.92 |
| | Euclidean,$L_\infty$ | $0.69 \pm 0.04$ | 0.93 |
| | $L_1,L_\infty$ | $0.68 \pm 0.04$ | 0.91 |
| | Cosine, Euclidean, $L_1, L_\infty$ | $0.63 \pm 0.13$ | 0.85 |
| MLP+SelfAttention | Cosine,Euclidean | $\mathbf{0.76} \pm 0.01$ | **1.00** |
| | Cosine,$L_1$ | $\mathbf{0.76} \pm 0.01$ | **1.00** |
| | Cosine,$L_\infty$ | $\mathbf{0.76} \pm 0.01$ | **1.00** |
| | Euclidean,$L_1$ | $0.68 \pm 0.06$ | 0.92 |
| | Euclidean,$L_\infty$ | $0.71 \pm 0.03$ | 0.95 |
| | $L_1,L_\infty$ | $0.69 \pm 0.03$ | 0.93 |
| | Cosine, Euclidean, $L_1, L_\infty$ | $\mathbf{0.76} \pm 0.02$ | 0.99 |
| MLP+Sum | Cosine,Euclidean | $\mathbf{0.78} \pm 0.02$ | **1.00** |
| | Cosine,$L_1$ | $\mathbf{0.78} \pm 0.01$ | **1.00** |
| | Cosine,$L_\infty$ | $\mathbf{0.77} \pm 0.01$ | **1.00** |
| | Euclidean,$L_1$ | $0.72 \pm 0.03$ | 0.97 |
| | Euclidean,$L_\infty$ | $0.68 \pm 0.04$ | 0.94 |
| | $L_1,L_\infty$ | $0.67 \pm 0.01$ | 0.94 |
| | Cosine, Euclidean, $L_1, L_\infty$ | $\mathbf{0.77} \pm 0.01$ | 1.00 |

Table 15: **Zero-Shot Stitching Results on** `CIFAR-10`. The table shows the mean and standard deviation of the test accuracy for the different projection methods sorted by maximum difference between projections, reported in the first column.

| Encoder | Decoder | MaxDiff | Cosine | Euclidean | $L_1$ | $L_\infty$ |
|---------|---------|---------|--------|-----------|-------|-----------|
| RexNet | RViT-B/16 | $0.11 \pm 0.01$ | $0.80 \pm 0.01$ | $0.74 \pm 0.02$ | $0.69 \pm 0.02$ | $0.46 \pm 0.02$ |
| | ViT-B/16 | $0.11 \pm 0.00$ | $0.79 \pm 0.01$ | $0.69 \pm 0.02$ | $0.69 \pm 0.01$ | $0.51 \pm 0.03$ |
| | CViT-B/32 | $0.09 \pm 0.01$ | $0.79 \pm 0.01$ | $0.73 \pm 0.01$ | $0.70 \pm 0.01$ | $0.48 \pm 0.03$ |
| | ViT-S/16 | $0.09 \pm 0.02$ | $0.79 \pm 0.01$ | $0.72 \pm 0.02$ | $0.70 \pm 0.02$ | $0.45 \pm 0.02$ |
| ViT-B/16 | RexNet | $0.02 \pm 0.01$ | $0.94 \pm 0.01$ | $0.95 \pm 0.00$ | $0.96 \pm 0.00$ | $0.63 \pm 0.01$ |
| CViT-B/32 | ViT-B/16 | $0.02 \pm 0.01$ | $0.87 \pm 0.01$ | $0.87 \pm 0.01$ | $0.87 \pm 0.01$ | $0.56 \pm 0.00$ |
| | RViT-B/16 | $0.02 \pm 0.00$ | $0.84 \pm 0.01$ | $0.84 \pm 0.01$ | $0.84 \pm 0.01$ | $0.54 \pm 0.02$ |
| RViT-B/16 | RexNet | $0.01 \pm 0.00$ | $0.93 \pm 0.00$ | $0.94 \pm 0.00$ | $0.94 \pm 0.00$ | $0.70 \pm 0.02$ |
| CViT-B/32 | RexNet | $0.01 \pm 0.00$ | $0.87 \pm 0.00$ | $0.88 \pm 0.00$ | $0.87 \pm 0.00$ | $0.47 \pm 0.01$ |
| | ViT-S/16 | $0.01 \pm 0.01$ | $0.89 \pm 0.01$ | $0.89 \pm 0.01$ | $0.89 \pm 0.01$ | $0.52 \pm 0.01$ |
| ViT-S/16 | RViT-B/16 | $0.01 \pm 0.00$ | $0.93 \pm 0.00$ | $0.93 \pm 0.00$ | $0.92 \pm 0.00$ | $0.73 \pm 0.00$ |
| RViT-B/16 | ViT-B/16 | $0.01 \pm 0.00$ | $0.95 \pm 0.00$ | $0.95 \pm 0.00$ | $0.95 \pm 0.00$ | $0.74 \pm 0.01$ |
| ViT-S/16 | ViT-B/16 | $0.01 \pm 0.00$ | $0.94 \pm 0.00$ | $0.93 \pm 0.00$ | $0.94 \pm 0.00$ | $0.69 \pm 0.01$ |
| RViT-B/16 | CViT-B/32 | $0.01 \pm 0.01$ | $0.93 \pm 0.00$ | $0.94 \pm 0.00$ | $0.94 \pm 0.00$ | $0.78 \pm 0.03$ |
| ViT-S/16 | CViT-B/32 | $0.01 \pm 0.00$ | $0.93 \pm 0.00$ | $0.93 \pm 0.00$ | $0.92 \pm 0.00$ | $0.66 \pm 0.03$ |
| | RexNet | $0.01 \pm 0.00$ | $0.92 \pm 0.00$ | $0.91 \pm 0.00$ | $0.92 \pm 0.00$ | $0.66 \pm 0.02$ |
| ViT-B/16 | RViT-B/16 | $0.01 \pm 0.00$ | $0.96 \pm 0.00$ | $0.96 \pm 0.00$ | $0.97 \pm 0.00$ | $0.76 \pm 0.00$ |
| | CViT-B/32 | $0.01 \pm 0.00$ | $0.95 \pm 0.00$ | $0.96 \pm 0.00$ | $0.96 \pm 0.00$ | $0.67 \pm 0.02$ |
| RViT-B/16 | ViT-S/16 | $0.00 \pm 0.00$ | $0.95 \pm 0.00$ | $0.95 \pm 0.00$ | $0.95 \pm 0.00$ | $0.79 \pm 0.02$ |
| ViT-B/16 | ViT-S/16 | $0.00 \pm 0.00$ | $0.97 \pm 0.00$ | $0.97 \pm 0.00$ | $0.97 \pm 0.00$ | $0.73 \pm 0.03$ |

Table 16: **Zero-Shot Stitching Results on** `CIFAR-100`. The table shows the mean and standard deviation of the test accuracy for the different projection methods sorted by maximum difference between projections, reported in the first column.

| Encoder | Decoder | MaxDiff | Cosine | Euclidean | $L_1$ | $L_\infty$ |
|---------|---------|---------|--------|-----------|-------|-----------|
| RexNet | ViT-B/16 | $0.24 \pm 0.01$ | $0.50 \pm 0.01$ | $0.40 \pm 0.02$ | $0.25 \pm 0.02$ | $0.21 \pm 0.02$ |
| ViT-S/16 | RexNet | $0.23 \pm 0.01$ | $0.73 \pm 0.00$ | $0.63 \pm 0.01$ | $0.50 \pm 0.01$ | $0.31 \pm 0.00$ |
| RexNet | RViT-B/16 | $0.20 \pm 0.02$ | $0.52 \pm 0.01$ | $0.42 \pm 0.02$ | $0.31 \pm 0.02$ | $0.22 \pm 0.01$ |
| ViT-S/16 | CViT-B/32 | $0.17 \pm 0.03$ | $0.68 \pm 0.00$ | $0.61 \pm 0.01$ | $0.51 \pm 0.03$ | $0.30 \pm 0.00$ |
| | ViT-B/16 | $0.14 \pm 0.00$ | $0.77 \pm 0.00$ | $0.64 \pm 0.00$ | $0.71 \pm 0.01$ | $0.33 \pm 0.01$ |
| | RViT-B/16 | $0.13 \pm 0.01$ | $0.75 \pm 0.00$ | $0.67 \pm 0.01$ | $0.62 \pm 0.01$ | $0.39 \pm 0.00$ |
| RexNet | ViT-S/16 | $0.12 \pm 0.01$ | $0.47 \pm 0.01$ | $0.40 \pm 0.02$ | $0.34 \pm 0.01$ | $0.17 \pm 0.00$ |
| | CViT-B/32 | $0.11 \pm 0.01$ | $0.50 \pm 0.01$ | $0.52 \pm 0.02$ | $0.41 \pm 0.02$ | $0.17 \pm 0.02$ |
| CViT-B/32 | RViT-B/16 | $0.07 \pm 0.03$ | $0.48 \pm 0.04$ | $0.52 \pm 0.02$ | $0.49 \pm 0.03$ | $0.32 \pm 0.01$ |
| RViT-B/16 | RexNet | $0.06 \pm 0.00$ | $0.79 \pm 0.00$ | $0.77 \pm 0.00$ | $0.72 \pm 0.00$ | $0.33 \pm 0.00$ |
| CViT-B/32 | ViT-S/16 | $0.05 \pm 0.01$ | $0.53 \pm 0.02$ | $0.54 \pm 0.02$ | $0.57 \pm 0.02$ | $0.22 \pm 0.01$ |
| | ViT-B/16 | $0.05 \pm 0.02$ | $0.53 \pm 0.02$ | $0.52 \pm 0.02$ | $0.56 \pm 0.01$ | $0.26 \pm 0.01$ |
| ViT-B/16 | RexNet | $0.04 \pm 0.01$ | $0.72 \pm 0.01$ | $0.72 \pm 0.01$ | $0.68 \pm 0.00$ | $0.40 \pm 0.02$ |
| | CViT-B/32 | $0.04 \pm 0.01$ | $0.70 \pm 0.01$ | $0.71 \pm 0.01$ | $0.73 \pm 0.01$ | $0.39 \pm 0.01$ |
| | ViT-S/16 | $0.03 \pm 0.00$ | $0.80 \pm 0.00$ | $0.83 \pm 0.00$ | $0.84 \pm 0.00$ | $0.44 \pm 0.01$ |
| CViT-B/32 | RexNet | $0.03 \pm 0.02$ | $0.52 \pm 0.01$ | $0.55 \pm 0.02$ | $0.53 \pm 0.00$ | $0.27 \pm 0.01$ |
| RViT-B/16 | ViT-S/16 | $0.03 \pm 0.01$ | $0.79 \pm 0.01$ | $0.82 \pm 0.00$ | $0.81 \pm 0.01$ | $0.34 \pm 0.00$ |
| ViT-B/16 | RViT-B/16 | $0.02 \pm 0.01$ | $0.78 \pm 0.01$ | $0.80 \pm 0.01$ | $0.79 \pm 0.00$ | $0.44 \pm 0.01$ |
| RViT-B/16 | ViT-B/16 | $0.01 \pm 0.00$ | $0.82 \pm 0.01$ | $0.81 \pm 0.00$ | $0.82 \pm 0.00$ | $0.26 \pm 0.01$ |
| | CViT-B/32 | $0.01 \pm 0.01$ | $0.75 \pm 0.00$ | $0.75 \pm 0.01$ | $0.75 \pm 0.00$ | $0.30 \pm 0.01$ |

Table 17: **Zero-Shot Stitching Results on** `Fashion MNIST`. The table shows the mean and standard deviation of the test accuracy for the different projection methods sorted by maximum difference between projections, reported in the first column.

| Encoder | Decoder | MaxDiff | Cosine | Euclidean | $L_1$ | $L_\infty$ |
|---------|---------|---------|--------|-----------|-------|------------|
| CViT-B/32 | RexNet | $0.05 \pm 0.00$ | $0.67 \pm 0.00$ | $0.72 \pm 0.01$ | $0.72 \pm 0.00$ | $0.54 \pm 0.00$ |
| RViT-B/16 | RexNet | $0.04 \pm 0.01$ | $0.76 \pm 0.01$ | $0.79 \pm 0.01$ | $0.80 \pm 0.01$ | $0.59 \pm 0.01$ |
| ViT-B/16 | CViT-B/32 | $0.03 \pm 0.00$ | $0.76 \pm 0.01$ | $0.78 \pm 0.00$ | $0.79 \pm 0.00$ | $0.50 \pm 0.01$ |
| CViT-B/32 | RViT-B/16 | $0.03 \pm 0.02$ | $0.68 \pm 0.01$ | $0.72 \pm 0.01$ | $0.71 \pm 0.01$ | $0.56 \pm 0.00$ |
| RViT-B/16 | ViT-B/16 | $0.03 \pm 0.01$ | $0.78 \pm 0.01$ | $0.79 \pm 0.01$ | $0.81 \pm 0.00$ | $0.61 \pm 0.01$ |
| ViT-S/16 | RViT-B/16 | $0.03 \pm 0.00$ | $0.76 \pm 0.00$ | $0.77 \pm 0.00$ | $0.78 \pm 0.00$ | $0.52 \pm 0.01$ |
| RViT-B/16 | CViT-B/32 | $0.03 \pm 0.01$ | $0.72 \pm 0.01$ | $0.72 \pm 0.01$ | $0.74 \pm 0.01$ | $0.57 \pm 0.01$ |
| ViT-B/16 | RexNet | $0.03 \pm 0.01$ | $0.79 \pm 0.00$ | $0.81 \pm 0.00$ | $0.81 \pm 0.00$ | $0.48 \pm 0.01$ |
| RexNet | RViT-B/16 | $0.03 \pm 0.01$ | $0.74 \pm 0.01$ | $0.77 \pm 0.00$ | $0.76 \pm 0.00$ | $0.53 \pm 0.02$ |
| CViT-B/32 | ViT-S/16 | $0.03 \pm 0.01$ | $0.66 \pm 0.02$ | $0.65 \pm 0.01$ | $0.65 \pm 0.01$ | $0.56 \pm 0.00$ |
| RexNet | ViT-S/16 | $0.02 \pm 0.01$ | $0.74 \pm 0.00$ | $0.77 \pm 0.00$ | $0.76 \pm 0.01$ | $0.64 \pm 0.01$ |
| ViT-S/16 | RexNet | $0.02 \pm 0.01$ | $0.78 \pm 0.01$ | $0.78 \pm 0.01$ | $0.78 \pm 0.01$ | $0.51 \pm 0.01$ |
| ViT-B/16 | ViT-S/16 | $0.02 \pm 0.01$ | $0.79 \pm 0.01$ | $0.80 \pm 0.01$ | $0.80 \pm 0.01$ | $0.57 \pm 0.01$ |
| CViT-B/32 | ViT-B/16 | $0.02 \pm 0.01$ | $0.70 \pm 0.01$ | $0.71 \pm 0.01$ | $0.72 \pm 0.00$ | $0.55 \pm 0.01$ |
| RexNet | CViT-B/32 | $0.01 \pm 0.01$ | $0.72 \pm 0.01$ | $0.74 \pm 0.01$ | $0.73 \pm 0.01$ | $0.63 \pm 0.00$ |
| RViT-B/16 | ViT-S/16 | $0.01 \pm 0.01$ | $0.79 \pm 0.00$ | $0.79 \pm 0.00$ | $0.80 \pm 0.01$ | $0.63 \pm 0.00$ |
| ViT-S/16 | ViT-B/16 | $0.01 \pm 0.00$ | $0.79 \pm 0.00$ | $0.80 \pm 0.00$ | $0.80 \pm 0.01$ | $0.54 \pm 0.01$ |
| RexNet | ViT-B/16 | $0.01 \pm 0.01$ | $0.76 \pm 0.00$ | $0.77 \pm 0.01$ | $0.77 \pm 0.00$ | $0.60 \pm 0.02$ |
| ViT-S/16 | CViT-B/32 | $0.01 \pm 0.00$ | $0.75 \pm 0.00$ | $0.75 \pm 0.00$ | $0.75 \pm 0.01$ | $0.55 \pm 0.02$ |
| ViT-B/16 | RViT-B/16 | $0.01 \pm 0.00$ | $0.84 \pm 0.00$ | $0.84 \pm 0.00$ | $0.84 \pm 0.00$ | $0.52 \pm 0.00$ |

Table 18: **Zero-Shot Stitching Results on** `MNIST`. The table shows the mean and standard deviation of the test accuracy for the different projection methods sorted by maximum difference between projections, reported in the first column.

| Encoder | Decoder | MaxDiff | Cosine | Euclidean | $L_1$ | $L_\infty$ |
|---|---|---|---|---|---|---|
| CViT-B/32 | ViT-B/16 | $0.09 \pm 0.01$ | $0.57 \pm 0.00$ | $0.65 \pm 0.01$ | $0.66 \pm 0.01$ | $0.55 \pm 0.01$ |
| ViT-S/16 | RexNet | $0.08 \pm 0.01$ | $0.62 \pm 0.01$ | $0.69 \pm 0.01$ | $0.71 \pm 0.00$ | $0.48 \pm 0.01$ |
| ViT-B/16 | RexNet | $0.08 \pm 0.01$ | $0.56 \pm 0.00$ | $0.62 \pm 0.01$ | $0.64 \pm 0.00$ | $0.43 \pm 0.01$ |
| | CViT-B/32 | $0.07 \pm 0.02$ | $0.62 \pm 0.02$ | $0.69 \pm 0.01$ | $0.69 \pm 0.01$ | $0.47 \pm 0.00$ |
| RexNet | ViT-S/16 | $0.07 \pm 0.01$ | $0.71 \pm 0.00$ | $0.65 \pm 0.01$ | $0.64 \pm 0.01$ | $0.51 \pm 0.00$ |
| ViT-B/16 | ViT-S/16 | $0.07 \pm 0.01$ | $0.66 \pm 0.01$ | $0.71 \pm 0.00$ | $0.73 \pm 0.00$ | $0.42 \pm 0.00$ |
| CViT-B/32 | RViT-B/16 | $0.05 \pm 0.03$ | $0.57 \pm 0.01$ | $0.62 \pm 0.02$ | $0.60 \pm 0.02$ | $0.54 \pm 0.01$ |
| ViT-B/16 | RViT-B/16 | $0.05 \pm 0.03$ | $0.53 \pm 0.02$ | $0.58 \pm 0.02$ | $0.56 \pm 0.02$ | $0.33 \pm 0.00$ |
| ViT-S/16 | RViT-B/16 | $0.05 \pm 0.02$ | $0.69 \pm 0.01$ | $0.73 \pm 0.01$ | $0.74 \pm 0.01$ | $0.49 \pm 0.02$ |
| CViT-B/32 | ViT-S/16 | $0.04 \pm 0.01$ | $0.60 \pm 0.01$ | $0.65 \pm 0.01$ | $0.65 \pm 0.01$ | $0.57 \pm 0.01$ |
| ViT-S/16 | CViT-B/32 | $0.04 \pm 0.01$ | $0.71 \pm 0.01$ | $0.69 \pm 0.01$ | $0.67 \pm 0.00$ | $0.49 \pm 0.01$ |
| RViT-B/16 | RexNet | $0.04 \pm 0.02$ | $0.63 \pm 0.00$ | $0.66 \pm 0.01$ | $0.67 \pm 0.01$ | $0.65 \pm 0.01$ |
| RexNet | CViT-B/32 | $0.04 \pm 0.01$ | $0.71 \pm 0.01$ | $0.74 \pm 0.01$ | $0.74 \pm 0.00$ | $0.48 \pm 0.00$ |
| RViT-B/16 | ViT-S/16 | $0.03 \pm 0.01$ | $0.74 \pm 0.01$ | $0.77 \pm 0.00$ | $0.77 \pm 0.00$ | $0.63 \pm 0.01$ |
| | ViT-B/16 | $0.03 \pm 0.01$ | $0.69 \pm 0.01$ | $0.71 \pm 0.00$ | $0.72 \pm 0.00$ | $0.53 \pm 0.00$ |
| CViT-B/32 | RexNet | $0.02 \pm 0.01$ | $0.71 \pm 0.00$ | $0.73 \pm 0.01$ | $0.73 \pm 0.02$ | $0.61 \pm 0.00$ |
| RViT-B/16 | CViT-B/32 | $0.02 \pm 0.01$ | $0.69 \pm 0.01$ | $0.68 \pm 0.01$ | $0.68 \pm 0.01$ | $0.61 \pm 0.01$ |
| RexNet | RViT-B/16 | $0.02 \pm 0.01$ | $0.67 \pm 0.01$ | $0.67 \pm 0.01$ | $0.66 \pm 0.01$ | $0.47 \pm 0.01$ |
| ViT-S/16 | ViT-B/16 | $0.01 \pm 0.01$ | $0.68 \pm 0.01$ | $0.69 \pm 0.00$ | $0.68 \pm 0.00$ | $0.46 \pm 0.02$ |
| RexNet | ViT-B/16 | $0.01 \pm 0.01$ | $0.73 \pm 0.01$ | $0.72 \pm 0.01$ | $0.72 \pm 0.00$ | $0.46 \pm 0.01$ |

Table 19: **Zero-Shot Stitching Results on** DBpedia. The table shows the mean and standard deviation of the test accuracy for the different projection methods sorted by maximum difference between projections, reported in the first column.

| Encoder | Decoder | MaxDiff | Cosine | Euclidean | $L_1$ | $L_\infty$ |
|---|---|---|---|---|---|---|
| XLM-R | BERT-C | $0.25 \pm 0.01$ | $0.14 \pm 0.00$ | $0.19 \pm 0.01$ | $0.39 \pm 0.01$ | $0.09 \pm 0.01$ |
| | ALBERT | $0.22 \pm 0.03$ | $0.15 \pm 0.03$ | $0.17 \pm 0.02$ | $0.36 \pm 0.02$ | $0.10 \pm 0.00$ |
| | ELECTRA | $0.22 \pm 0.02$ | $0.08 \pm 0.00$ | $0.13 \pm 0.01$ | $0.30 \pm 0.02$ | $0.06 \pm 0.01$ |
| BERT-U | RoBERTa | $0.20 \pm 0.01$ | $0.53 \pm 0.01$ | $0.36 \pm 0.01$ | $0.33 \pm 0.01$ | $0.14 \pm 0.01$ |
| XLM-R | RoBERTa | $0.19 \pm 0.02$ | $0.16 \pm 0.02$ | $0.19 \pm 0.01$ | $0.35 \pm 0.01$ | $0.08 \pm 0.01$ |
| BERT-U | ELECTRA | $0.19 \pm 0.02$ | $0.50 \pm 0.01$ | $0.31 \pm 0.01$ | $0.35 \pm 0.00$ | $0.14 \pm 0.01$ |
| XLM-R | BERT-U | $0.16 \pm 0.01$ | $0.12 \pm 0.00$ | $0.14 \pm 0.01$ | $0.27 \pm 0.01$ | $0.10 \pm 0.00$ |
| BERT-U | BERT-C | $0.15 \pm 0.01$ | $0.57 \pm 0.00$ | $0.43 \pm 0.00$ | $0.42 \pm 0.01$ | $0.13 \pm 0.01$ |
| | CViT-B/32 | $0.15 \pm 0.03$ | $0.41 \pm 0.02$ | $0.30 \pm 0.01$ | $0.26 \pm 0.01$ | $0.11 \pm 0.00$ |
| | XLM-R | $0.15 \pm 0.01$ | $0.50 \pm 0.02$ | $0.35 \pm 0.01$ | $0.36 \pm 0.01$ | $0.08 \pm 0.01$ |
| RoBERTa | BERT-C | $0.12 \pm 0.02$ | $0.40 \pm 0.03$ | $0.30 \pm 0.02$ | $0.28 \pm 0.01$ | $0.13 \pm 0.01$ |
| BERT-C | ELECTRA | $0.12 \pm 0.02$ | $0.49 \pm 0.02$ | $0.46 \pm 0.01$ | $0.58 \pm 0.02$ | $0.11 \pm 0.01$ |
| XLM-R | CViT-B/32 | $0.12 \pm 0.02$ | $0.12 \pm 0.01$ | $0.12 \pm 0.02$ | $0.23 \pm 0.00$ | $0.08 \pm 0.00$ |
| RoBERTa | BERT-U | $0.11 \pm 0.05$ | $0.31 \pm 0.06$ | $0.27 \pm 0.06$ | $0.31 \pm 0.06$ | $0.15 \pm 0.01$ |
| ELECTRA | ALBERT | $0.11 \pm 0.01$ | $0.25 \pm 0.01$ | $0.27 \pm 0.02$ | $0.36 \pm 0.02$ | $0.11 \pm 0.00$ |
| BERT-U | ALBERT | $0.11 \pm 0.04$ | $0.53 \pm 0.01$ | $0.42 \pm 0.03$ | $0.43 \pm 0.02$ | $0.11 \pm 0.00$ |
| ELECTRA | BERT-C | $0.11 \pm 0.01$ | $0.31 \pm 0.01$ | $0.27 \pm 0.01$ | $0.38 \pm 0.03$ | $0.10 \pm 0.00$ |
| ALBERT | XLM-R | $0.09 \pm 0.01$ | $0.40 \pm 0.00$ | $0.42 \pm 0.00$ | $0.49 \pm 0.01$ | $0.14 \pm 0.00$ |
| RoBERTa | ELECTRA | $0.09 \pm 0.02$ | $0.36 \pm 0.03$ | $0.29 \pm 0.01$ | $0.37 \pm 0.02$ | $0.13 \pm 0.01$ |
| BERT-C | XLM-R | $0.09 \pm 0.05$ | $0.49 \pm 0.03$ | $0.57 \pm 0.04$ | $0.51 \pm 0.03$ | $0.12 \pm 0.01$ |
| | RoBERTa | $0.08 \pm 0.05$ | $0.52 \pm 0.03$ | $0.56 \pm 0.02$ | $0.60 \pm 0.02$ | $0.14 \pm 0.00$ |
| ELECTRA | BERT-U | $0.08 \pm 0.03$ | $0.24 \pm 0.01$ | $0.21 \pm 0.01$ | $0.29 \pm 0.02$ | $0.14 \pm 0.00$ |
| | RoBERTa | $0.08 \pm 0.01$ | $0.35 \pm 0.02$ | $0.34 \pm 0.01$ | $0.41 \pm 0.01$ | $0.10 \pm 0.01$ |
| RoBERTa | CViT-B/32 | $0.07 \pm 0.02$ | $0.22 \pm 0.02$ | $0.21 \pm 0.01$ | $0.24 \pm 0.06$ | $0.11 \pm 0.01$ |
| BERT-C | ALBERT | $0.07 \pm 0.03$ | $0.43 \pm 0.04$ | $0.43 \pm 0.05$ | $0.42 \pm 0.05$ | $0.13 \pm 0.00$ |
| | BERT-U | $0.06 \pm 0.06$ | $0.49 \pm 0.04$ | $0.53 \pm 0.04$ | $0.54 \pm 0.04$ | $0.21 \pm 0.01$ |
| ELECTRA | CViT-B/32 | $0.06 \pm 0.01$ | $0.22 \pm 0.01$ | $0.19 \pm 0.01$ | $0.25 \pm 0.01$ | $0.10 \pm 0.01$ |
| BERT-C | CViT-B/32 | $0.06 \pm 0.05$ | $0.33 \pm 0.02$ | $0.34 \pm 0.03$ | $0.35 \pm 0.07$ | $0.09 \pm 0.01$ |
| RoBERTa | ALBERT | $0.06 \pm 0.00$ | $0.32 \pm 0.04$ | $0.31 \pm 0.01$ | $0.36 \pm 0.01$ | $0.14 \pm 0.00$ |
| ELECTRA | XLM-R | $0.05 \pm 0.01$ | $0.25 \pm 0.00$ | $0.25 \pm 0.00$ | $0.30 \pm 0.01$ | $0.13 \pm 0.01$ |
| RoBERTa | XLM-R | $0.05 \pm 0.01$ | $0.24 \pm 0.02$ | $0.21 \pm 0.01$ | $0.26 \pm 0.00$ | $0.12 \pm 0.01$ |
| ALBERT | ELECTRA | $0.04 \pm 0.01$ | $0.50 \pm 0.00$ | $0.50 \pm 0.01$ | $0.53 \pm 0.01$ | $0.17 \pm 0.01$ |
| CViT-B/32 | ELECTRA | $0.04 \pm 0.01$ | $0.20 \pm 0.00$ | $0.23 \pm 0.00$ | $0.21 \pm 0.00$ | $0.11 \pm 0.01$ |
| | RoBERTa | $0.03 \pm 0.00$ | $0.21 \pm 0.00$ | $0.24 \pm 0.00$ | $0.24 \pm 0.01$ | $0.15 \pm 0.01$ |
| | BERT-U | $0.03 \pm 0.01$ | $0.19 \pm 0.01$ | $0.22 \pm 0.01$ | $0.21 \pm 0.00$ | $0.12 \pm 0.00$ |
| ALBERT | BERT-U | $0.03 \pm 0.02$ | $0.54 \pm 0.01$ | $0.55 \pm 0.00$ | $0.57 \pm 0.01$ | $0.18 \pm 0.00$ |
| CViT-B/32 | ALBERT | $0.03 \pm 0.01$ | $0.17 \pm 0.01$ | $0.19 \pm 0.00$ | $0.20 \pm 0.01$ | $0.09 \pm 0.01$ |
| ALBERT | RoBERTa | $0.02 \pm 0.02$ | $0.50 \pm 0.02$ | $0.50 \pm 0.00$ | $0.52 \pm 0.01$ | $0.18 \pm 0.02$ |
| | BERT-C | $0.02 \pm 0.02$ | $0.51 \pm 0.01$ | $0.53 \pm 0.01$ | $0.53 \pm 0.01$ | $0.18 \pm 0.00$ |
| | CViT-B/32 | $0.02 \pm 0.00$ | $0.43 \pm 0.00$ | $0.45 \pm 0.01$ | $0.43 \pm 0.01$ | $0.15 \pm 0.00$ |
| CViT-B/32 | BERT-C | $0.02 \pm 0.01$ | $0.20 \pm 0.00$ | $0.21 \pm 0.01$ | $0.22 \pm 0.01$ | $0.14 \pm 0.02$ |
| | XLM-R | $0.02 \pm 0.00$ | $0.22 \pm 0.00$ | $0.23 \pm 0.00$ | $0.24 \pm 0.00$ | $0.13 \pm 0.00$ |

Table 20: **Zero-Shot Stitching Results on** `TREC`. The table shows the mean and standard deviation of the test accuracy for the different projection methods sorted by maximum difference between projections, reported in the first column.

| Encoder | Decoder | MaxDiff | Cosine | Euclidean | $L_1$ | $L_\infty$ |
|---|---|---|---|---|---|---|
| XLM-R | RoBERTa | $0.32 \pm 0.11$ | $0.40 \pm 0.10$ | $0.46 \pm 0.04$ | $0.69 \pm 0.06$ | $0.33 \pm 0.09$ |
| | BERT-U | $0.30 \pm 0.13$ | $0.36 \pm 0.18$ | $0.53 \pm 0.04$ | $0.64 \pm 0.01$ | $0.31 \pm 0.04$ |
| BERT-C | XLM-R | $0.30 \pm 0.14$ | $0.34 \pm 0.08$ | $0.55 \pm 0.03$ | $0.64 \pm 0.06$ | $0.17 \pm 0.03$ |
| XLM-R | BERT-C | $0.26 \pm 0.08$ | $0.42 \pm 0.09$ | $0.56 \pm 0.02$ | $0.68 \pm 0.02$ | $0.26 \pm 0.11$ |
| CViT-B/32 | BERT-C | $0.25 \pm 0.16$ | $0.24 \pm 0.12$ | $0.45 \pm 0.03$ | $0.45 \pm 0.12$ | $0.33 \pm 0.01$ |
| RoBERTa | XLM-R | $0.25 \pm 0.03$ | $0.37 \pm 0.03$ | $0.36 \pm 0.01$ | $0.60 \pm 0.02$ | $0.27 \pm 0.08$ |
| XLM-R | CViT-B/32 | $0.25 \pm 0.12$ | $0.30 \pm 0.06$ | $0.47 \pm 0.04$ | $0.55 \pm 0.07$ | $0.34 \pm 0.03$ |
| ALBERT | XLM-R | $0.24 \pm 0.12$ | $0.37 \pm 0.09$ | $0.50 \pm 0.02$ | $0.61 \pm 0.03$ | $0.28 \pm 0.04$ |
| BERT-U | XLM-R | $0.24 \pm 0.08$ | $0.34 \pm 0.08$ | $0.53 \pm 0.03$ | $0.58 \pm 0.00$ | $0.27 \pm 0.10$ |
| | ELECTRA | $0.23 \pm 0.09$ | $0.35 \pm 0.06$ | $0.57 \pm 0.06$ | $0.43 \pm 0.07$ | $0.32 \pm 0.01$ |
| RoBERTa | ALBERT | $0.23 \pm 0.04$ | $0.42 \pm 0.01$ | $0.65 \pm 0.03$ | $0.62 \pm 0.03$ | $0.14 \pm 0.04$ |
| ELECTRA | ALBERT | $0.21 \pm 0.11$ | $0.51 \pm 0.04$ | $0.48 \pm 0.11$ | $0.65 \pm 0.04$ | $0.16 \pm 0.04$ |
| CViT-B/32 | RoBERTa | $0.20 \pm 0.02$ | $0.45 \pm 0.02$ | $0.54 \pm 0.01$ | $0.65 \pm 0.01$ | $0.32 \pm 0.09$ |
| | ELECTRA | $0.20 \pm 0.07$ | $0.33 \pm 0.07$ | $0.52 \pm 0.01$ | $0.39 \pm 0.01$ | $0.31 \pm 0.01$ |
| BERT-C | ELECTRA | $0.20 \pm 0.06$ | $0.40 \pm 0.04$ | $0.60 \pm 0.02$ | $0.53 \pm 0.03$ | $0.20 \pm 0.02$ |
| ELECTRA | RoBERTa | $0.20 \pm 0.00$ | $0.52 \pm 0.02$ | $0.33 \pm 0.01$ | $0.45 \pm 0.05$ | $0.32 \pm 0.07$ |
| XLM-R | ALBERT | $0.19 \pm 0.12$ | $0.46 \pm 0.08$ | $0.58 \pm 0.07$ | $0.64 \pm 0.07$ | $0.21 \pm 0.01$ |
| RoBERTa | BERT-U | $0.19 \pm 0.02$ | $0.44 \pm 0.02$ | $0.60 \pm 0.02$ | $0.62 \pm 0.03$ | $0.21 \pm 0.01$ |
| XLM-R | ELECTRA | $0.19 \pm 0.07$ | $0.37 \pm 0.06$ | $0.56 \pm 0.02$ | $0.45 \pm 0.02$ | $0.38 \pm 0.03$ |
| CViT-B/32 | ALBERT | $0.18 \pm 0.08$ | $0.45 \pm 0.10$ | $0.60 \pm 0.04$ | $0.48 \pm 0.03$ | $0.11 \pm 0.01$ |
| | XLM-R | $0.17 \pm 0.06$ | $0.36 \pm 0.06$ | $0.51 \pm 0.04$ | $0.53 \pm 0.03$ | $0.34 \pm 0.01$ |
| ELECTRA | BERT-U | $0.16 \pm 0.05$ | $0.52 \pm 0.02$ | $0.44 \pm 0.10$ | $0.59 \pm 0.05$ | $0.20 \pm 0.04$ |
| BERT-U | RoBERTa | $0.14 \pm 0.02$ | $0.56 \pm 0.00$ | $0.64 \pm 0.03$ | $0.70 \pm 0.02$ | $0.29 \pm 0.08$ |
| | CViT-B/32 | $0.14 \pm 0.06$ | $0.46 \pm 0.03$ | $0.52 \pm 0.01$ | $0.60 \pm 0.04$ | $0.25 \pm 0.08$ |
| BERT-C | BERT-U | $0.14 \pm 0.18$ | $0.53 \pm 0.13$ | $0.61 \pm 0.03$ | $0.65 \pm 0.06$ | $0.23 \pm 0.04$ |
| RoBERTa | CViT-B/32 | $0.13 \pm 0.05$ | $0.42 \pm 0.04$ | $0.51 \pm 0.07$ | $0.54 \pm 0.03$ | $0.27 \pm 0.10$ |
| ALBERT | CViT-B/32 | $0.12 \pm 0.04$ | $0.51 \pm 0.08$ | $0.47 \pm 0.02$ | $0.56 \pm 0.02$ | $0.30 \pm 0.02$ |
| BERT-C | ALBERT | $0.12 \pm 0.10$ | $0.53 \pm 0.10$ | $0.54 \pm 0.06$ | $0.59 \pm 0.08$ | $0.23 \pm 0.03$ |
| BERT-U | BERT-C | $0.11 \pm 0.02$ | $0.41 \pm 0.00$ | $0.48 \pm 0.01$ | $0.53 \pm 0.02$ | $0.35 \pm 0.02$ |
| BERT-C | CViT-B/32 | $0.11 \pm 0.09$ | $0.43 \pm 0.07$ | $0.53 \pm 0.01$ | $0.55 \pm 0.03$ | $0.18 \pm 0.06$ |
| ALBERT | RoBERTa | $0.11 \pm 0.06$ | $0.54 \pm 0.03$ | $0.60 \pm 0.03$ | $0.65 \pm 0.02$ | $0.29 \pm 0.06$ |
| ELECTRA | CViT-B/32 | $0.10 \pm 0.08$ | $0.48 \pm 0.05$ | $0.50 \pm 0.04$ | $0.57 \pm 0.05$ | $0.33 \pm 0.05$ |
| | XLM-R | $0.10 \pm 0.08$ | $0.48 \pm 0.08$ | $0.55 \pm 0.02$ | $0.57 \pm 0.02$ | $0.30 \pm 0.07$ |
| BERT-C | RoBERTa | $0.10 \pm 0.09$ | $0.56 \pm 0.11$ | $0.59 \pm 0.04$ | $0.60 \pm 0.02$ | $0.24 \pm 0.01$ |
| BERT-U | ALBERT | $0.09 \pm 0.05$ | $0.63 \pm 0.07$ | $0.64 \pm 0.05$ | $0.64 \pm 0.05$ | $0.22 \pm 0.06$ |
| ALBERT | BERT-U | $0.08 \pm 0.05$ | $0.54 \pm 0.04$ | $0.57 \pm 0.03$ | $0.60 \pm 0.04$ | $0.47 \pm 0.01$ |
| CViT-B/32 | BERT-U | $0.08 \pm 0.01$ | $0.51 \pm 0.00$ | $0.58 \pm 0.02$ | $0.53 \pm 0.05$ | $0.12 \pm 0.00$ |
| RoBERTa | BERT-C | $0.07 \pm 0.03$ | $0.52 \pm 0.03$ | $0.59 \pm 0.00$ | $0.57 \pm 0.01$ | $0.18 \pm 0.03$ |
| ELECTRA | BERT-C | $0.07 \pm 0.02$ | $0.49 \pm 0.02$ | $0.56 \pm 0.02$ | $0.56 \pm 0.01$ | $0.17 \pm 0.05$ |
| ALBERT | BERT-C | $0.07 \pm 0.01$ | $0.43 \pm 0.00$ | $0.49 \pm 0.02$ | $0.49 \pm 0.01$ | $0.32 \pm 0.01$ |
| | ELECTRA | $0.06 \pm 0.01$ | $0.57 \pm 0.02$ | $0.61 \pm 0.02$ | $0.62 \pm 0.01$ | $0.29 \pm 0.01$ |
| RoBERTa | ELECTRA | $0.02 \pm 0.01$ | $0.41 \pm 0.00$ | $0.40 \pm 0.01$ | $0.42 \pm 0.01$ | $0.16 \pm 0.01$ |

Table 21: **Zero-Shot Stitching Results on** `N24news(Text)`. The table shows the mean and standard deviation of the test accuracy for the different projection methods sorted by maximum difference between projections, reported in the first column.

| Encoder | Decoder | MaxDiff | Cosine | Euclidean | $L_1$ | $L_\infty$ |
|---------|---------|---------|--------|-----------|-------|------------|
| XLM-R | ALBERT | $0.18 \pm 0.01$ | $0.16 \pm 0.01$ | $0.29 \pm 0.02$ | $0.34 \pm 0.01$ | $0.07 \pm 0.00$ |
| | BERT-C | $0.17 \pm 0.01$ | $0.22 \pm 0.01$ | $0.33 \pm 0.02$ | $0.39 \pm 0.02$ | $0.07 \pm 0.00$ |
| | BERT-U | $0.15 \pm 0.00$ | $0.21 \pm 0.01$ | $0.29 \pm 0.01$ | $0.36 \pm 0.01$ | $0.08 \pm 0.00$ |
| | ELECTRA | $0.15 \pm 0.01$ | $0.18 \pm 0.01$ | $0.26 \pm 0.01$ | $0.33 \pm 0.01$ | $0.06 \pm 0.00$ |
| RoBERTa | XLM-R | $0.11 \pm 0.00$ | $0.53 \pm 0.00$ | $0.55 \pm 0.00$ | $0.64 \pm 0.00$ | $0.08 \pm 0.00$ |
| XLM-R | RoBERTa | $0.09 \pm 0.02$ | $0.37 \pm 0.01$ | $0.37 \pm 0.02$ | $0.45 \pm 0.02$ | $0.06 \pm 0.01$ |
| RoBERTa | ELECTRA | $0.09 \pm 0.01$ | $0.38 \pm 0.01$ | $0.35 \pm 0.00$ | $0.44 \pm 0.00$ | $0.11 \pm 0.01$ |
| | ALBERT | $0.08 \pm 0.01$ | $0.38 \pm 0.02$ | $0.39 \pm 0.01$ | $0.46 \pm 0.01$ | $0.08 \pm 0.00$ |
| | BERT-U | $0.08 \pm 0.01$ | $0.50 \pm 0.01$ | $0.44 \pm 0.02$ | $0.51 \pm 0.01$ | $0.08 \pm 0.01$ |
| ELECTRA | RoBERTa | $0.08 \pm 0.01$ | $0.11 \pm 0.01$ | $0.19 \pm 0.01$ | $0.13 \pm 0.01$ | $0.06 \pm 0.01$ |
| | BERT-U | $0.07 \pm 0.04$ | $0.05 \pm 0.03$ | $0.10 \pm 0.02$ | $0.11 \pm 0.01$ | $0.09 \pm 0.01$ |
| | BERT-C | $0.06 \pm 0.02$ | $0.07 \pm 0.01$ | $0.13 \pm 0.01$ | $0.11 \pm 0.01$ | $0.09 \pm 0.01$ |
| BERT-C | BERT-U | $0.04 \pm 0.04$ | $0.21 \pm 0.02$ | $0.23 \pm 0.04$ | $0.23 \pm 0.04$ | $0.13 \pm 0.01$ |
| ELECTRA | XLM-R | $0.04 \pm 0.01$ | $0.09 \pm 0.00$ | $0.12 \pm 0.01$ | $0.11 \pm 0.01$ | $0.07 \pm 0.00$ |
| BERT-C | ELECTRA | $0.03 \pm 0.01$ | $0.18 \pm 0.01$ | $0.20 \pm 0.01$ | $0.21 \pm 0.01$ | $0.10 \pm 0.01$ |
| RoBERTa | BERT-C | $0.03 \pm 0.00$ | $0.59 \pm 0.01$ | $0.57 \pm 0.00$ | $0.60 \pm 0.01$ | $0.08 \pm 0.00$ |
| BERT-C | RoBERTa | $0.03 \pm 0.01$ | $0.26 \pm 0.00$ | $0.25 \pm 0.01$ | $0.23 \pm 0.01$ | $0.14 \pm 0.01$ |
| BERT-U | ELECTRA | $0.03 \pm 0.02$ | $0.14 \pm 0.02$ | $0.16 \pm 0.01$ | $0.17 \pm 0.01$ | $0.07 \pm 0.01$ |
| | ALBERT | $0.03 \pm 0.01$ | $0.15 \pm 0.01$ | $0.18 \pm 0.00$ | $0.18 \pm 0.01$ | $0.07 \pm 0.00$ |
| BERT-C | CViT-B/32 | $0.03 \pm 0.01$ | $0.05 \pm 0.02$ | $0.04 \pm 0.02$ | $0.04 \pm 0.02$ | $0.04 \pm 0.01$ |
| CViT-B/32 | RoBERTa | $0.03 \pm 0.01$ | $0.04 \pm 0.01$ | $0.04 \pm 0.02$ | $0.05 \pm 0.00$ | $0.05 \pm 0.00$ |
| RoBERTa | CViT-B/32 | $0.03 \pm 0.03$ | $0.06 \pm 0.02$ | $0.04 \pm 0.01$ | $0.04 \pm 0.01$ | $0.04 \pm 0.01$ |
| ELECTRA | CViT-B/32 | $0.02 \pm 0.02$ | $0.05 \pm 0.02$ | $0.04 \pm 0.02$ | $0.04 \pm 0.02$ | $0.05 \pm 0.01$ |
| BERT-U | XLM-R | $0.02 \pm 0.01$ | $0.18 \pm 0.01$ | $0.19 \pm 0.01$ | $0.18 \pm 0.02$ | $0.07 \pm 0.00$ |
| ALBERT | CViT-B/32 | $0.02 \pm 0.00$ | $0.05 \pm 0.02$ | $0.04 \pm 0.01$ | $0.04 \pm 0.01$ | $0.05 \pm 0.01$ |
| BERT-U | RoBERTa | $0.02 \pm 0.00$ | $0.18 \pm 0.01$ | $0.20 \pm 0.01$ | $0.17 \pm 0.01$ | $0.07 \pm 0.01$ |
| | BERT-C | $0.02 \pm 0.02$ | $0.20 \pm 0.01$ | $0.22 \pm 0.01$ | $0.22 \pm 0.01$ | $0.06 \pm 0.00$ |
| XLM-R | CViT-B/32 | $0.02 \pm 0.01$ | $0.06 \pm 0.01$ | $0.05 \pm 0.01$ | $0.04 \pm 0.01$ | $0.05 \pm 0.01$ |
| ELECTRA | ALBERT | $0.02 \pm 0.02$ | $0.11 \pm 0.02$ | $0.13 \pm 0.01$ | $0.13 \pm 0.01$ | $0.06 \pm 0.00$ |
| BERT-U | CViT-B/32 | $0.01 \pm 0.01$ | $0.04 \pm 0.01$ | $0.05 \pm 0.01$ | $0.05 \pm 0.01$ | $0.04 \pm 0.00$ |
| ALBERT | RoBERTa | $0.01 \pm 0.00$ | $0.08 \pm 0.00$ | $0.09 \pm 0.00$ | $0.07 \pm 0.01$ | $0.08 \pm 0.00$ |
| BERT-C | XLM-R | $0.01 \pm 0.01$ | $0.29 \pm 0.00$ | $0.29 \pm 0.00$ | $0.28 \pm 0.01$ | $0.14 \pm 0.00$ |
| | ALBERT | $0.01 \pm 0.01$ | $0.28 \pm 0.01$ | $0.29 \pm 0.00$ | $0.29 \pm 0.01$ | $0.10 \pm 0.01$ |
| ALBERT | BERT-C | $0.01 \pm 0.01$ | $0.08 \pm 0.00$ | $0.09 \pm 0.01$ | $0.09 \pm 0.01$ | $0.06 \pm 0.00$ |
| | ELECTRA | $0.01 \pm 0.01$ | $0.11 \pm 0.00$ | $0.12 \pm 0.01$ | $0.12 \pm 0.01$ | $0.08 \pm 0.01$ |
| | BERT-U | $0.01 \pm 0.01$ | $0.11 \pm 0.01$ | $0.11 \pm 0.00$ | $0.11 \pm 0.00$ | $0.07 \pm 0.00$ |
| CViT-B/32 | XLM-R | $0.01 \pm 0.01$ | $0.04 \pm 0.01$ | $0.05 \pm 0.00$ | $0.05 \pm 0.00$ | $0.05 \pm 0.00$ |
| ALBERT | XLM-R | $0.01 \pm 0.01$ | $0.09 \pm 0.00$ | $0.10 \pm 0.00$ | $0.09 \pm 0.01$ | $0.07 \pm 0.00$ |
| CViT-B/32 | ELECTRA | $0.00 \pm 0.00$ | $0.05 \pm 0.00$ | $0.05 \pm 0.00$ | $0.05 \pm 0.00$ | $0.05 \pm 0.00$ |
| | BERT-U | $0.00 \pm 0.00$ | $0.05 \pm 0.00$ | $0.05 \pm 0.00$ | $0.05 \pm 0.00$ | $0.05 \pm 0.00$ |
| | BERT-C | $0.00 \pm 0.00$ | $0.03 \pm 0.00$ | $0.03 \pm 0.00$ | $0.03 \pm 0.00$ | $0.03 \pm 0.00$ |
| | ALBERT | $0.00 \pm 0.00$ | $0.05 \pm 0.00$ | $0.05 \pm 0.00$ | $0.05 \pm 0.00$ | $0.05 \pm 0.00$ |

Table 22: **Zero-Shot Stitching Results on** `ImageNet1k`. Zero-shot accuracy score across different architectures and seeds. The standard deviation is provided for each accuracy score. The proposed methodology utilizing the *Aggregation by sum* consistently outperforms the other results.

| Encoder | Aggregation Function | Accuracy ↑ |
|---------|---------------------|------------|
| RViT-B/16 | Cosine | $0.50 \pm 0.01$ |
| | Euclidean | $\mathbf{0.52} \pm 0.01$ |
| | $L_1$ | $0.45 \pm 0.01$ |
| | $L_\infty$ | $0.01 \pm 0.00$ |
| | Cosine, Euclidean, $L_1$, $L_\infty$ | $\mathbf{0.67} \pm 0.01$ |
| ViT-B/16 | Cosine | $0.27 \pm 0.03$ |
| | Euclidean | $0.25 \pm 0.02$ |
| | $L_1$ | $\mathbf{0.43} \pm 0.02$ |
| | $L_\infty$ | $0.02 \pm 0.01$ |
| | Cosine, Euclidean, $L_1$, $L_\infty$ | $\mathbf{0.56} \pm 0.02$ |
| ViT-S/16 | Cosine | $0.34 \pm 0.02$ |
| | Euclidean | $0.33 \pm 0.03$ |
| | $L_1$ | $\mathbf{0.35} \pm 0.02$ |
| | $L_\infty$ | $0.02 \pm 0.01$ |
| | Cosine, Euclidean, $L_1$, $L_\infty$ | $\mathbf{0.51} \pm 0.02$ |

Table 23: **Zero-Shot Stitching Reconstruction Performance.** Zero-shot $L_1$ and MSE score for the reconstruction task across different seeds on `CIFAR-100` using AEs. The proposed methodology utilizing the *Aggregation by sum* consistently matches or outperforms the other results.

| Aggregation | Projection | $L_1$ | MSE |
|-------------|-----------|-------|-----|
| MLP+Sum | Cosine, Euclidean, $L_1$, $L_\infty$ | $\mathbf{0.01} \pm 0.00$ | $\mathbf{0.06} \pm 0.00$ |
| MLP+Sum | Cosine,Euclidean,$L_1$ | $0.02 \pm 0.00$ | $0.10 \pm 0.01$ |
| - | Cosine | $0.06 \pm 0.02$ | $0.20 \pm 0.03$ |
| - | Euclidean | $\mathbf{0.01} \pm 0.00$ | $\mathbf{0.06} \pm 0.00$ |
| - | $L_1$ | $\mathbf{0.01} \pm 0.00$ | $0.09 \pm 0.00$ |
| - | $L_\infty$ | $0.03 \pm 0.01$ | $0.14 \pm 0.01$ |
| - | Absolute | $0.26 \pm 0.06$ | $0.43 \pm 0.06$ |

Table 24: **End-to-end Performance Results on** `CIFAR-10`. The classifier head is a simple Linear layer.

| Model | Aggregation | Projection | |
|---|---|---|---|
| `CViT-B/32` | - | Absolute | $0.92 \pm 0.05$ |
| | | Cosine | $0.90 \pm 0.06$ |
| | | Euclidean | $0.90 \pm 0.06$ |
| | | $L_1$ | $0.90 \pm 0.06$ |
| | | $L_\infty$ | $0.79 \pm 0.13$ |
| | MLP+Sum | Cosine, Euclidean, $L_1$, $L_\infty$ | $0.91 \pm 0.05$ |
| | SelfAttention | Cosine, Euclidean, $L_1$, $L_\infty$ | $0.90 \pm 0.06$ |
| `RViT-B/16` | - | Absolute | $0.85 \pm 0.19$ |
| | | Cosine | $0.83 \pm 0.18$ |
| | | Euclidean | $0.83 \pm 0.18$ |
| | | $L_1$ | $0.83 \pm 0.19$ |
| | | $L_\infty$ | $0.79 \pm 0.20$ |
| | MLP+Sum | Cosine, Euclidean, $L_1$, $L_\infty$ | $0.84 \pm 0.19$ |
| | SelfAttention | Cosine, Euclidean, $L_1$, $L_\infty$ | $0.82 \pm 0.20$ |
| `RexNet` | - | Absolute | $0.80 \pm 0.21$ |
| | | Cosine | $0.76 \pm 0.20$ |
| | | Euclidean | $0.76 \pm 0.20$ |
| | | $L_1$ | $0.77 \pm 0.20$ |
| | | $L_\infty$ | $0.72 \pm 0.20$ |
| | MLP+Sum | Cosine, Euclidean, $L_1$, $L_\infty$ | $0.79 \pm 0.21$ |
| | SelfAttention | Cosine, Euclidean, $L_1$, $L_\infty$ | $0.76 \pm 0.21$ |
| `ViT-B/16` | - | Absolute | $0.86 \pm 0.18$ |
| | | Cosine | $0.84 \pm 0.18$ |
| | | Euclidean | $0.84 \pm 0.18$ |
| | | $L_1$ | $0.85 \pm 0.18$ |
| | | $L_\infty$ | $0.73 \pm 0.19$ |
| | MLP+Sum | Cosine, Euclidean, $L_1$, $L_\infty$ | $0.86 \pm 0.18$ |
| | SelfAttention | Cosine, Euclidean, $L_1$, $L_\infty$ | $0.84 \pm 0.19$ |
| `ViT-S/16` | - | Absolute | $0.83 \pm 0.19$ |
| | | Cosine | $0.82 \pm 0.19$ |
| | | Euclidean | $0.82 \pm 0.18$ |
| | | $L_1$ | $0.82 \pm 0.19$ |
| | | $L_\infty$ | $0.74 \pm 0.19$ |
| | MLP+Sum | Cosine, Euclidean, $L_1$, $L_\infty$ | $0.83 \pm 0.18$ |
| | SelfAttention | Cosine, Euclidean, $L_1$, $L_\infty$ | $0.82 \pm 0.20$ |

Table 25: **End-to-end Performance Results on** `CIFAR-100`. The classifier head is a simple Linear layer.

| Model | Aggregation | Projection | |
|---|---|---|---|
| `CViT-B/32` | - | Absolute | $0.92 \pm 0.05$ |
| | | Cosine | $0.90 \pm 0.06$ |
| | | Euclidean | $0.90 \pm 0.06$ |
| | | $L_1$ | $0.90 \pm 0.06$ |
| | | $L_\infty$ | $0.79 \pm 0.13$ |
| | MLP+Sum | Cosine, Euclidean, $L_1$, $L_\infty$ | $0.91 \pm 0.05$ |
| | SelfAttention | Cosine, Euclidean, $L_1$, $L_\infty$ | $0.90 \pm 0.06$ |
| `RViT-B/16` | - | Absolute | $0.85 \pm 0.19$ |
| | | Cosine | $0.83 \pm 0.18$ |
| | | Euclidean | $0.83 \pm 0.18$ |
| | | $L_1$ | $0.83 \pm 0.19$ |
| | | $L_\infty$ | $0.79 \pm 0.20$ |
| | MLP+Sum | Cosine, Euclidean, $L_1$, $L_\infty$ | $0.84 \pm 0.19$ |
| | SelfAttention | Cosine, Euclidean, $L_1$, $L_\infty$ | $0.82 \pm 0.20$ |
| `RexNet` | - | Absolute | $0.80 \pm 0.21$ |
| | | Cosine | $0.76 \pm 0.20$ |
| | | Euclidean | $0.76 \pm 0.20$ |
| | | $L_1$ | $0.77 \pm 0.20$ |
| | | $L_\infty$ | $0.72 \pm 0.20$ |
| | MLP+Sum | Cosine, Euclidean, $L_1$, $L_\infty$ | $0.79 \pm 0.21$ |
| | SelfAttention | Cosine, Euclidean, $L_1$, $L_\infty$ | $0.76 \pm 0.21$ |
| `ViT-B/16` | - | Absolute | $0.86 \pm 0.18$ |
| | | Cosine | $0.84 \pm 0.18$ |
| | | Euclidean | $0.84 \pm 0.18$ |
| | | $L_1$ | $0.85 \pm 0.18$ |
| | | $L_\infty$ | $0.73 \pm 0.19$ |
| | MLP+Sum | Cosine, Euclidean, $L_1$, $L_\infty$ | $0.86 \pm 0.18$ |
| | SelfAttention | Cosine, Euclidean, $L_1$, $L_\infty$ | $0.84 \pm 0.19$ |
| `ViT-S/16` | - | Absolute | $0.83 \pm 0.19$ |
| | | Cosine | $0.82 \pm 0.19$ |
| | | Euclidean | $0.82 \pm 0.18$ |
| | | $L_1$ | $0.82 \pm 0.19$ |
| | | $L_\infty$ | $0.74 \pm 0.19$ |
| | MLP+Sum | Cosine, Euclidean, $L_1$, $L_\infty$ | $0.83 \pm 0.18$ |
| | SelfAttention | Cosine, Euclidean, $L_1$, $L_\infty$ | $0.82 \pm 0.20$ |

Table 26: **End-to-end Performance Results on** `Fashion MNIST`. The classifier head is a simple Linear layer.

| Model | Aggregation | Projection | |
|---|---|---|---|
| `CViT-B/32` | - | Absolute | $0.92 \pm 0.05$ |
| | | Cosine | $0.90 \pm 0.06$ |
| | | Euclidean | $0.90 \pm 0.06$ |
| | | $L_1$ | $0.90 \pm 0.06$ |
| | | $L_\infty$ | $0.79 \pm 0.13$ |
| | MLP+Sum | Cosine, Euclidean, $L_1$, $L_\infty$ | $0.91 \pm 0.05$ |
| | SelfAttention | Cosine, Euclidean, $L_1$, $L_\infty$ | $0.90 \pm 0.06$ |
| `RViT-B/16` | - | Absolute | $0.85 \pm 0.19$ |
| | | Cosine | $0.83 \pm 0.18$ |
| | | Euclidean | $0.83 \pm 0.18$ |
| | | $L_1$ | $0.83 \pm 0.19$ |
| | | $L_\infty$ | $0.79 \pm 0.20$ |
| | MLP+Sum | Cosine, Euclidean, $L_1$, $L_\infty$ | $0.84 \pm 0.19$ |
| | SelfAttention | Cosine, Euclidean, $L_1$, $L_\infty$ | $0.82 \pm 0.20$ |
| `RexNet` | - | Absolute | $0.80 \pm 0.21$ |
| | | Cosine | $0.76 \pm 0.20$ |
| | | Euclidean | $0.76 \pm 0.20$ |
| | | $L_1$ | $0.77 \pm 0.20$ |
| | | $L_\infty$ | $0.72 \pm 0.20$ |
| | MLP+Sum | Cosine, Euclidean, $L_1$, $L_\infty$ | $0.79 \pm 0.21$ |
| | SelfAttention | Cosine, Euclidean, $L_1$, $L_\infty$ | $0.76 \pm 0.21$ |
| `ViT-B/16` | - | Absolute | $0.86 \pm 0.18$ |
| | | Cosine | $0.84 \pm 0.18$ |
| | | Euclidean | $0.84 \pm 0.18$ |
| | | $L_1$ | $0.85 \pm 0.18$ |
| | | $L_\infty$ | $0.73 \pm 0.19$ |
| | MLP+Sum | Cosine, Euclidean, $L_1$, $L_\infty$ | $0.86 \pm 0.18$ |
| | SelfAttention | Cosine, Euclidean, $L_1$, $L_\infty$ | $0.84 \pm 0.19$ |
| `ViT-S/16` | - | Absolute | $0.83 \pm 0.19$ |
| | | Cosine | $0.82 \pm 0.19$ |
| | | Euclidean | $0.82 \pm 0.18$ |
| | | $L_1$ | $0.82 \pm 0.19$ |
| | | $L_\infty$ | $0.74 \pm 0.19$ |
| | MLP+Sum | Cosine, Euclidean, $L_1$, $L_\infty$ | $0.83 \pm 0.18$ |
| | SelfAttention | Cosine, Euclidean, $L_1$, $L_\infty$ | $0.82 \pm 0.20$ |

Table 27: **End-to-end Performance Results on** `MNIST`. The classifier head is a simple Linear layer.

| Model | Aggregation | Projection | |
|---|---|---|---|
| `CViT-B/32` | - | Absolute | $0.92 \pm 0.05$ |
| | | Cosine | $0.90 \pm 0.06$ |
| | | Euclidean | $0.90 \pm 0.06$ |
| | | $L_1$ | $0.90 \pm 0.06$ |
| | | $L_\infty$ | $0.79 \pm 0.13$ |
| | MLP+Sum | Cosine, Euclidean, $L_1$, $L_\infty$ | $0.91 \pm 0.05$ |
| | SelfAttention | Cosine, Euclidean, $L_1$, $L_\infty$ | $0.90 \pm 0.06$ |
| `RViT-B/16` | - | Absolute | $0.85 \pm 0.19$ |
| | | Cosine | $0.83 \pm 0.18$ |
| | | Euclidean | $0.83 \pm 0.18$ |
| | | $L_1$ | $0.83 \pm 0.19$ |
| | | $L_\infty$ | $0.79 \pm 0.20$ |
| | MLP+Sum | Cosine, Euclidean, $L_1$, $L_\infty$ | $0.84 \pm 0.19$ |
| | SelfAttention | Cosine, Euclidean, $L_1$, $L_\infty$ | $0.82 \pm 0.20$ |
| `RexNet` | - | Absolute | $0.80 \pm 0.21$ |
| | | Cosine | $0.76 \pm 0.20$ |
| | | Euclidean | $0.76 \pm 0.20$ |
| | | $L_1$ | $0.77 \pm 0.20$ |
| | | $L_\infty$ | $0.72 \pm 0.20$ |
| | MLP+Sum | Cosine, Euclidean, $L_1$, $L_\infty$ | $0.79 \pm 0.21$ |
| | SelfAttention | Cosine, Euclidean, $L_1$, $L_\infty$ | $0.76 \pm 0.21$ |
| `ViT-B/16` | - | Absolute | $0.86 \pm 0.18$ |
| | | Cosine | $0.84 \pm 0.18$ |
| | | Euclidean | $0.84 \pm 0.18$ |
| | | $L_1$ | $0.85 \pm 0.18$ |
| | | $L_\infty$ | $0.73 \pm 0.19$ |
| | MLP+Sum | Cosine, Euclidean, $L_1$, $L_\infty$ | $0.86 \pm 0.18$ |
| | SelfAttention | Cosine, Euclidean, $L_1$, $L_\infty$ | $0.84 \pm 0.19$ |
| `ViT-S/16` | - | Absolute | $0.83 \pm 0.19$ |
| | | Cosine | $0.82 \pm 0.19$ |
| | | Euclidean | $0.82 \pm 0.18$ |
| | | $L_1$ | $0.82 \pm 0.19$ |
| | | $L_\infty$ | $0.74 \pm 0.19$ |
| | MLP+Sum | Cosine, Euclidean, $L_1$, $L_\infty$ | $0.83 \pm 0.18$ |
| | SelfAttention | Cosine, Euclidean, $L_1$, $L_\infty$ | $0.82 \pm 0.20$ |

Table 28: **End-to-end Performance Results on** `DBpedia`. The classifier head is a simple Linear layer.

| Model | Aggregation | Projection | |
|---|---|---|---|
| `ALBERT` | - | Absolute | $0.71 \pm 0.21$ |
| | | Cosine | $0.61 \pm 0.23$ |
| | | Euclidean | $0.62 \pm 0.22$ |
| | | $L_1$ | $0.63 \pm 0.23$ |
| | | $L_\infty$ | $0.49 \pm 0.19$ |
| | MLP+Sum | Cosine, Euclidean, $L_1, L_\infty$ | $0.63 \pm 0.22$ |
| | SelfAttention | Cosine, Euclidean, $L_1, L_\infty$ | $0.61 \pm 0.23$ |
| `BERT-C` | - | Absolute | $0.84 \pm 0.11$ |
| | | Cosine | $0.78 \pm 0.13$ |
| | | Euclidean | $0.80 \pm 0.13$ |
| | | $L_1$ | $0.81 \pm 0.13$ |
| | | $L_\infty$ | $0.56 \pm 0.12$ |
| | MLP+Sum | Cosine, Euclidean, $L_1, L_\infty$ | $0.81 \pm 0.11$ |
| | SelfAttention | Cosine, Euclidean, $L_1, L_\infty$ | $0.77 \pm 0.13$ |
| `BERT-U` | - | Absolute | $0.77 \pm 0.17$ |
| | | Cosine | $0.73 \pm 0.17$ |
| | | Euclidean | $0.74 \pm 0.17$ |
| | | $L_1$ | $0.74 \pm 0.18$ |
| | | $L_\infty$ | $0.46 \pm 0.08$ |
| | MLP+Sum | Cosine, Euclidean, $L_1, L_\infty$ | $0.74 \pm 0.16$ |
| | SelfAttention | Cosine, Euclidean, $L_1, L_\infty$ | $0.66 \pm 0.18$ |
| `CViT-B/32` | - | Absolute | $0.33 \pm 0.24$ |
| | | Cosine | $0.31 \pm 0.23$ |
| | | Euclidean | $0.33 \pm 0.26$ |
| | | $L_1$ | $0.33 \pm 0.26$ |
| | | $L_\infty$ | $0.23 \pm 0.14$ |
| | MLP+Sum | Cosine, Euclidean, $L_1, L_\infty$ | $0.33 \pm 0.25$ |
| | SelfAttention | Cosine, Euclidean, $L_1, L_\infty$ | $0.20 \pm 0.16$ |
| `ELECTRA` | - | Absolute | $0.76 \pm 0.14$ |
| | | Cosine | $0.58 \pm 0.15$ |
| | | Euclidean | $0.59 \pm 0.09$ |
| | | $L_1$ | $0.64 \pm 0.13$ |
| | | $L_\infty$ | $0.32 \pm 0.07$ |
| | MLP+Sum | Cosine, Euclidean, $L_1, L_\infty$ | $0.65 \pm 0.10$ |
| | SelfAttention | Cosine, Euclidean, $L_1, L_\infty$ | $0.61 \pm 0.14$ |
| `RoBERTa` | - | Absolute | $0.81 \pm 0.10$ |
| | | Cosine | $0.75 \pm 0.04$ |
| | | Euclidean | $0.77 \pm 0.01$ |
| | | $L_1$ | $0.81 \pm 0.02$ |
| | | $L_\infty$ | $0.36 \pm 0.07$ |
| | MLP+Sum | Cosine, Euclidean, $L_1, L_\infty$ | $0.80 \pm 0.04$ |
| | SelfAttention | Cosine, Euclidean, $L_1, L_\infty$ | $0.61 \pm 0.27$ |
| `XLM-R` | - | Absolute | $0.74 \pm 0.12$ |
| | | Cosine | $0.58 \pm 0.07$ |
| | | Euclidean | $0.64 \pm 0.07$ |
| | | $L_1$ | $0.78 \pm 0.05$ |
| | | $L_\infty$ | $0.26 \pm 0.09$ |
| | MLP+Sum | Cosine, Euclidean, $L_1, L_\infty$ | $0.78 \pm 0.06$ |
| | SelfAttention | Cosine, Euclidean, $L_1, L_\infty$ | $0.71 \pm 0.22$ |

Table 29: **End-to-end Performance Results on** TREC. The classifier head is a simple Linear layer.

| Model | Aggregation | Projection | |
|---|---|---|---|
| ALBERT | - | Absolute | $0.71 \pm 0.21$ |
| | | Cosine | $0.61 \pm 0.23$ |
| | | Euclidean | $0.62 \pm 0.22$ |
| | | $L_1$ | $0.63 \pm 0.23$ |
| | | $L_\infty$ | $0.49 \pm 0.19$ |
| | MLP+Sum | Cosine, Euclidean, $L_1, L_\infty$ | $0.63 \pm 0.22$ |
| | SelfAttention | Cosine, Euclidean, $L_1, L_\infty$ | $0.61 \pm 0.23$ |
| BERT-C | - | Absolute | $0.84 \pm 0.11$ |
| | | Cosine | $0.78 \pm 0.13$ |
| | | Euclidean | $0.80 \pm 0.13$ |
| | | $L_1$ | $0.81 \pm 0.13$ |
| | | $L_\infty$ | $0.56 \pm 0.12$ |
| | MLP+Sum | Cosine, Euclidean, $L_1, L_\infty$ | $0.81 \pm 0.11$ |
| | SelfAttention | Cosine, Euclidean, $L_1, L_\infty$ | $0.77 \pm 0.13$ |
| BERT-U | - | Absolute | $0.77 \pm 0.17$ |
| | | Cosine | $0.73 \pm 0.17$ |
| | | Euclidean | $0.74 \pm 0.17$ |
| | | $L_1$ | $0.74 \pm 0.18$ |
| | | $L_\infty$ | $0.46 \pm 0.08$ |
| | MLP+Sum | Cosine, Euclidean, $L_1, L_\infty$ | $0.74 \pm 0.16$ |
| | SelfAttention | Cosine, Euclidean, $L_1, L_\infty$ | $0.66 \pm 0.18$ |
| CViT-B/32 | - | Absolute | $0.33 \pm 0.24$ |
| | | Cosine | $0.31 \pm 0.23$ |
| | | Euclidean | $0.33 \pm 0.26$ |
| | | $L_1$ | $0.33 \pm 0.26$ |
| | | $L_\infty$ | $0.23 \pm 0.14$ |
| | MLP+Sum | Cosine, Euclidean, $L_1, L_\infty$ | $0.33 \pm 0.25$ |
| | SelfAttention | Cosine, Euclidean, $L_1, L_\infty$ | $0.20 \pm 0.16$ |
| ELECTRA | - | Absolute | $0.76 \pm 0.14$ |
| | | Cosine | $0.58 \pm 0.15$ |
| | | Euclidean | $0.59 \pm 0.09$ |
| | | $L_1$ | $0.64 \pm 0.13$ |
| | | $L_\infty$ | $0.32 \pm 0.07$ |
| | MLP+Sum | Cosine, Euclidean, $L_1, L_\infty$ | $0.65 \pm 0.10$ |
| | SelfAttention | Cosine, Euclidean, $L_1, L_\infty$ | $0.61 \pm 0.14$ |
| RoBERTa | - | Absolute | $0.81 \pm 0.10$ |
| | | Cosine | $0.75 \pm 0.04$ |
| | | Euclidean | $0.77 \pm 0.01$ |
| | | $L_1$ | $0.81 \pm 0.02$ |
| | | $L_\infty$ | $0.36 \pm 0.07$ |
| | MLP+Sum | Cosine, Euclidean, $L_1, L_\infty$ | $0.80 \pm 0.04$ |
| | SelfAttention | Cosine, Euclidean, $L_1, L_\infty$ | $0.61 \pm 0.27$ |
| XLM-R | - | Absolute | $0.74 \pm 0.12$ |
| | | Cosine | $0.58 \pm 0.07$ |
| | | Euclidean | $0.64 \pm 0.07$ |
| | | $L_1$ | $0.78 \pm 0.05$ |
| | | $L_\infty$ | $0.26 \pm 0.09$ |
| | MLP+Sum | Cosine, Euclidean, $L_1, L_\infty$ | $0.78 \pm 0.06$ |
| | SelfAttention | Cosine, Euclidean, $L_1, L_\infty$ | $0.71 \pm 0.22$ |

Table 30: **End-to-end Performance Results on** `N24news(Text)`. The classifier head is a simple Linear layer.

| Model | Aggregation | Projection | |
|---|---|---|---|
| ALBERT | - | Absolute | $0.71 \pm 0.21$ |
| | | Cosine | $0.61 \pm 0.23$ |
| | | Euclidean | $0.62 \pm 0.22$ |
| | | $L_1$ | $0.63 \pm 0.23$ |
| | | $L_\infty$ | $0.49 \pm 0.19$ |
| | MLP+Sum | Cosine, Euclidean, $L_1$, $L_\infty$ | $0.63 \pm 0.22$ |
| | SelfAttention | Cosine, Euclidean, $L_1$, $L_\infty$ | $0.61 \pm 0.23$ |
| BERT-C | - | Absolute | $0.84 \pm 0.11$ |
| | | Cosine | $0.78 \pm 0.13$ |
| | | Euclidean | $0.80 \pm 0.13$ |
| | | $L_1$ | $0.81 \pm 0.13$ |
| | | $L_\infty$ | $0.56 \pm 0.12$ |
| | MLP+Sum | Cosine, Euclidean, $L_1$, $L_\infty$ | $0.81 \pm 0.11$ |
| | SelfAttention | Cosine, Euclidean, $L_1$, $L_\infty$ | $0.77 \pm 0.13$ |
| BERT-U | - | Absolute | $0.77 \pm 0.17$ |
| | | Cosine | $0.73 \pm 0.17$ |
| | | Euclidean | $0.74 \pm 0.17$ |
| | | $L_1$ | $0.74 \pm 0.18$ |
| | | $L_\infty$ | $0.46 \pm 0.08$ |
| | MLP+Sum | Cosine, Euclidean, $L_1$, $L_\infty$ | $0.74 \pm 0.16$ |
| | SelfAttention | Cosine, Euclidean, $L_1$, $L_\infty$ | $0.66 \pm 0.18$ |
| CViT-B/32 | - | Absolute | $0.33 \pm 0.24$ |
| | | Cosine | $0.31 \pm 0.23$ |
| | | Euclidean | $0.33 \pm 0.26$ |
| | | $L_1$ | $0.33 \pm 0.26$ |
| | | $L_\infty$ | $0.23 \pm 0.14$ |
| | MLP+Sum | Cosine, Euclidean, $L_1$, $L_\infty$ | $0.33 \pm 0.25$ |
| | SelfAttention | Cosine, Euclidean, $L_1$, $L_\infty$ | $0.20 \pm 0.16$ |
| ELECTRA | - | Absolute | $0.76 \pm 0.14$ |
| | | Cosine | $0.58 \pm 0.15$ |
| | | Euclidean | $0.59 \pm 0.09$ |
| | | $L_1$ | $0.64 \pm 0.13$ |
| | | $L_\infty$ | $0.32 \pm 0.07$ |
| | MLP+Sum | Cosine, Euclidean, $L_1$, $L_\infty$ | $0.65 \pm 0.10$ |
| | SelfAttention | Cosine, Euclidean, $L_1$, $L_\infty$ | $0.61 \pm 0.14$ |
| RoBERTa | - | Absolute | $0.81 \pm 0.10$ |
| | | Cosine | $0.75 \pm 0.04$ |
| | | Euclidean | $0.77 \pm 0.01$ |
| | | $L_1$ | $0.81 \pm 0.02$ |
| | | $L_\infty$ | $0.36 \pm 0.07$ |
| | MLP+Sum | Cosine, Euclidean, $L_1$, $L_\infty$ | $0.80 \pm 0.04$ |
| | SelfAttention | Cosine, Euclidean, $L_1$, $L_\infty$ | $0.61 \pm 0.27$ |
| XLM-R | - | Absolute | $0.74 \pm 0.12$ |
| | | Cosine | $0.58 \pm 0.07$ |
| | | Euclidean | $0.64 \pm 0.07$ |
| | | $L_1$ | $0.78 \pm 0.05$ |
| | | $L_\infty$ | $0.26 \pm 0.09$ |
| | MLP+Sum | Cosine, Euclidean, $L_1$, $L_\infty$ | $0.78 \pm 0.06$ |
| | SelfAttention | Cosine, Euclidean, $L_1$, $L_\infty$ | $0.71 \pm 0.22$ |

Table 31: **Graph End-to-End Classification Score.** Accuracy score across different architectures and seeds. The aggregation function is the *Aggregation by sum*.

| Aggregation | Projection | Accuracy ↑ |
|---|---|---|
| - | Absolute | **0.79** $\pm$ 0.01 |
| | Cosine | 0.74 $\pm$ 0.01 |
| | Euclidean | 0.46 $\pm$ 0.06 |
| | $L_1$ | 0.44 $\pm$ 0.06 |
| | $L_\infty$ | 0.12 $\pm$ 0.03 |
| Concat* | Cosine,Euclidean | 0.76 $\pm$ 0.01 |
| | Cosine, $L_1$ | **0.77** $\pm$ 0.01 |
| | Cosine,$L_\infty$ | 0.75 $\pm$ 0.01 |
| | Euclidean,$L_1$ | 0.55 $\pm$ 0.06 |
| | Euclidean,$L_\infty$ | 0.46 $\pm$ 0.11 |
| | $L_1,L_\infty$ | 0.48 $\pm$ 0.11 |
| | Cosine, Euclidean, $L_1, L_\infty$ | **0.77** $\pm$ 0.00 |
| SelfAttention | Cosine,Euclidean | 0.75 $\pm$ 0.01 |
| | Cosine,$L_1$ | 0.75 $\pm$ 0.01 |
| | Cosine,$L_\infty$ | **0.76** $\pm$ 0.02 |
| | Euclidean,$L_1$ | 0.74 $\pm$ 0.02 |
| | Euclidean,$L_\infty$ | 0.75 $\pm$ 0.02 |
| | $L_1,L_\infty$ | 0.75 $\pm$ 0.02 |
| | Cosine, Euclidean, $L_1, L_\infty$ | 0.74 $\pm$ 0.02 |
| MLP+SelfAttention | Cosine,Euclidean | **0.76** $\pm$ 0.02 |
| | Cosine,$L_1$ | **0.76** $\pm$ 0.02 |
| | Cosine,$L_\infty$ | **0.76** $\pm$ 0.02 |
| | Euclidean,$L_1$ | 0.73 $\pm$ 0.01 |
| | Euclidean,$L_\infty$ | 0.74 $\pm$ 0.01 |
| | $L_1,L_\infty$ | 0.74 $\pm$ 0.01 |
| | Cosine, Euclidean, $L_1, L_\infty$ | **0.77** $\pm$ 0.01 |
| MLP+Sum | Cosine,Euclidean | **0.77** $\pm$ 0.01 |
| | Cosine,$L_1$ | **0.77** $\pm$ 0.01 |
| | Cosine,$L_\infty$ | **0.77** $\pm$ 0.02 |
| | Euclidean,$L_1$ | 0.75 $\pm$ 0.02 |
| | Euclidean,$L_\infty$ | 0.72 $\pm$ 0.02 |
| | $L_1,L_\infty$ | 0.71 $\pm$ 0.03 |
| | Cosine, Euclidean, $L_1, L_\infty$ | 0.76 $\pm$ 0.01 |

