# OpenReview forum: "From Bricks to Bridges: Product of Invariances to Enhance Latent Space Communication"
_ICLR.cc/2024/Conference — ICLR 2024 spotlight_

### Official Review · Reviewer_pGUF · 2023-10-29

**Soundness:** 3 good
**Presentation:** 3 good
**Contribution:** 3 good
**Rating:** 6
**Confidence:** 3

**Summary:**

The paper is about enhancing the relative representation. Relative representation is determined with dissimilarity measure between target data and anchor that is invariant to angle transformation. The former work of Moschella et al. (2022) uses cosine angle as this dissimilarity, but in this paper, it aggregates other dissmilarity to enhance latent communication. The results of this aggregation is assessed by accuracy of zero-shot classification using stiching models.

**Strengths:**

The paper gives evidences that why the relative representation only using cosine angle can be inappropriate.

**Weaknesses:**

1. The definition of the RR framework is strange. It is stated that RR is concatenation of $d(z, a_i)$, but $z$ and $a_i$ should be in different domain $(\mathcal{Z},$ and $\mathcal{X})$. I am assuming that the anchors are also encoded with $E_\theta$ so that $a_i$'s in the latent space $\mathcal{Z}$

2. The experiments setting in the section 4 is unclear. I am having trouble figuring out what is a stiching model for this downstream task and how it is trained. I am assuming it is the same definition as the stiching model defined in Moschella et al. (2022), but I am having trouble how the decoder for this down-stream task is (pre-)tained.

3. The enhancement of relative representation through aggregating is not convincing for me. In Table 2, the aggreagated accuracies closely matches with using $L_1$ encoder. Using MLP or SelfAttention in aggreagation does not seems to be fair in that it requires an addtional training to get the additional parameters for these layers (correct me if I am wrong.)

**Questions:**

1. In experiment in Section 4.3 ~ 4.4, is the MLP and self-attention aggregation trained (fine-tuned) in end-to-end fashion?

2. Does the downstream task with relative representation presented also enhance the performance of other tasks? (e.g. generation)

---

> ### Author Response · Authors · 2023-11-19
>
> We express our appreciation to the reviewer for their valuable comments and inquiries. We would like to provide clarifications on the following points:
>
> &nbsp;
>
> Weakness 1 (**RR definition**): We sincerely thank the reviewer for highlighting that the way we formulated RR contains an abuse of notation. The corrected version has been incorporated into the PDF revision.
>
> &nbsp;
>
> Weakness 2 (**Model-stitching definition**): We expanded the description of the stitching procedure, dedicating a specific paragraph after the experimental setting (Section 4.2) and Figure 10 describing the stitching procedure. The revised version is accessible in the PDF revision.
>
> &nbsp;
>
> Weakness 3:
> - **Paper aim**: Our paper aims to explore and understand the transformations that relate different latent spaces, introducing a method to achieve latent communication without relying on prior knowledge of the optimal invariance to incorporate. As further clarified in the [general response](https://openreview.net/forum?id=vngVydDWft&noteId=44LjVL0qyv), the purpose is not to enhance relative representation through an aggregating mechanism.
> - **Additional training**:  Our methodology does not require additional training in the stitching procedure. We have added a paragraph in Section 4.2 and Figure 10 to clarify this important aspect of our method. For further details, please refer to the response to Question 1 and the [General Response](https://openreview.net/forum?id=vngVydDWft&noteId=44LjVL0qyv).
>
> &nbsp;
>
> We would also like to reply to the following questions:
>
> Question 1 (**Fine-tuned vs end-to-end**): All the downstream experiments are performed in a zero-shot way (including Section 4.3), the learnable parameters are exclusively trained during the initial, end-to-end network training phase and are then employed in their trained frozen state in the stitched model (as part of the decoder model). The only exception to this setting is the one in Section 4.4, “Subspace selection”, where we want to assess the possible gains in integrating additional information into the subspace selection process. For further details, please refer to the [general response](https://openreview.net/forum?id=vngVydDWft&noteId=44LjVL0qyv), Section 4.2, and Figure 10 of the revision.
>
> &nbsp;
>
> Question 2 (**Other downstream tasks**): To assess the applicability of our method to other downstream tasks, we conducted the **zero-shot generation task on CIFAR100**. Qualitative (Figure 14) and quantitative results (Table 24) demonstrate that our approach, incorporating all invariances (*Cos, Euc, $L_1$, $L_\infty$*) with the MLP+Sum aggregation (same setting of all the other experiments), consistently equals or exceeds the performance of the best single-invariance model. We would like to remark that this achievement is accomplished without prior knowledge of the most effective invariance and without any training or fine-tuning involved after assembling the pre-trained models into the stitched model. Results can be found in the PDF revision.
>
> &nbsp;
>
> ---
>
> We hope to have clarified the reviewer's concerns and remain available to discuss any further concerns or questions.

---

> > ### Author Response · Authors · 2023-11-22
> >
> > We thank the reviewer once again for providing valuable feedback and suggesting actionable modifications to improve the paper further.
> >
> > Please let us know if all concerns were correctly addressed in our revision or if there are any further questions or concerns to clarify during the remaining discussion period.

---

> ### Comment · Reviewer_pGUF · 2023-11-22
>
> I thank the authors for their efforts on the rebuttal and kind responds. My questions are all cleared up and I adjusted the score.

---

### Official Review · Reviewer_8YvH · 2023-10-31

**Soundness:** 3 good
**Presentation:** 3 good
**Contribution:** 3 good
**Rating:** 6
**Confidence:** 4

**Summary:**

This paper presents a product projection mechanism to generalize the framework of relative representation. In particular, the authors incorporate a set of invariances into the representation by constructing a production space of invariances. The findings are intuitive that multiple projections behave differently across different choices on initialization, model architecture, etc. Experimental results proved the effectiveness of the proposed method.

**Strengths:**

(1) The motivation is well presented of infusing multiple invariances into relative representation.

(2) The explanations and illustrations are mostly clear and intuitive of the manifold assumption and the product projection mechanism.

**Weaknesses:**

(1) On Page 5, the result analysis presents the discovery challenges the assumption in Moschella et al. (2022). More explanations are required to make this point clear. Besides, wondering if the experimental results are just a normal fluctuation due to different runs.

(2) On Page 6, the authors used 1280 randomly selected but fixed anchors. This is also a kind of randomness that is not explained away. In fact, for different choice of anchors, the sensitivity is different of the projection and measure function.

(3) On the experiments, the employed datasets and models are in small-scale and probably prone to overfitting issues. Do the analysis conclusions hold for large-scale models such as Stable Diffusion and GPT? There should be large-scale results to support the findings.

**Questions:**

No.

---

> ### Author Response · Authors · 2023-11-19
> **Response [1/2]**
>
> We thank the reviewer for providing valuable feedback, and we would like to offer further clarification on the following points:
>
> &nbsp;
>
> Weakness 1 (**Runs fluctuation**): To ensure the robustness of our experiments, we deliberately employed a minimum of three different random seeds. We selected this limit not only to validate the reliability of our experiments but also to assess the applicability and generalizability of our method across a broader range of experiments.
>
> &nbsp;
>
> Weakness 2 (**Anchor selection**): When referring to "randomly but fixed anchors," we mean that the anchors are randomly sampled uniformly from the training set but with a fixed seed value. In this regard, we **conducted an ablation study on the number of randomly selected anchors** for the stitching task using CIFAR100 across three different anchor seeds. The results reveal that varying the number of anchors leads to different transformations, indicating that a single projection function cannot incorporate the desired invariance. In contrast, our method enables the attainment of the highest score regardless of the number of anchors or the seed value employed for the uniform random sampling of anchors. We have introduced a new section (A.7) in the PDF revision for additional details, including Table 10 and Figure 9.
>
> &nbsp;
>
> Weakness 3 (**Large scale datasets and models**): In our work, we intentionally aligned our choice of models with those used in Moschella et al. (2022), which are widely recognized in the field for their embedding capabilities [1,2,3,4,5,6,7,8,9,10,11]. While these models do not match the large-scale scope of systems like Stable Diffusion or GPT, they are nonetheless significant for their demonstrative value in capturing emerging behaviors. This decision is further substantiated by existing literature, which suggests that even super large-scale models exhibit similar phenomena of emergent similarities [12, 13, 14, 15].
> Additionally, in this revision, we measure the performance of the proposed model on the ImageNet1k dataset, which is larger than the other employed datasets. Results in Table 23 show that the proposed methodology utilizing the *Aggregation by sum* consistently matches or outperforms the others.
>
> &nbsp;
>
> ---
>
> We hope that our responses address all the concerns raised by the reviewer. Additionally, we have included supplementary experiments on ImageNet and reconstruction tasks, detailed in the [General Response](https://openreview.net/forum?id=vngVydDWft&noteId=44LjVL0qyv). If further clarification is needed or if there are additional concerns, we are more than willing to engage in further discussion.

---

> > ### Author Response · Authors · 2023-11-19
> > **Response [2/2]**
> >
> > [1] Imperial, J.M. (2021). BERT Embeddings for Automatic Readability Assessment. Recent Advances in Natural Language Processing.
> >
> > [2] S. Gogineni and A. Pimpalshende, "Predicting IMDB Movie Rating Using Deep Learning," 2020 5th International Conference on Communication and Electronics Systems (ICCES)
> >
> > [3] Shijie Wu and Mark Dredze. 2019. Beto, Bentz, Becas: The Surprising Cross-Lingual Effectiveness of BERT. In Proceedings of the 2019 Conference on Empirical Methods in Natural Language Processing and the 9th International Joint Conference on Natural Language Processing (EMNLP-IJCNLP)
> >
> > [4] Alexis Conneau, Shijie Wu, Haoran Li, Luke Zettlemoyer, and Veselin Stoyanov. 2020. Emerging Cross-lingual Structure in Pretrained Language Models. In Proceedings of the 58th Annual Meeting of the Association for Computational Linguistics.
> >
> > [5] Yang Liu and Mirella Lapata. 2019. Text Summarization with Pretrained Encoders. In Proceedings of the 2019 Conference on Empirical Methods in Natural Language Processing and the 9th International Joint Conference on Natural Language Processing (EMNLP-IJCNLP).
> >
> > [6] Yajie Wu, Weihan Ren, and Zhihui Yang. 2023. What Does Pre-Train Bring to Vision Transformer. In Proceedings of the 2023 7th International Conference on Big Data and Internet of Things (BDIOT '23).
> >
> > [7] Yao, J., Wang, X., Yang, S., & Wang, B. (2023). ViTMatte: Boosting image matting with pre-trained plain vision transformers. Information Fusion.
> >
> > [8] Cha, J., Lee, K., Park, S., Chun, S. (2022). Domain Generalization by Mutual-Information Regularization with Pre-trained Models. In: Avidan, S., Brostow, G., Cissé, M., Farinella, G.M., Hassner, T. (eds) Computer Vision – ECCV 2022. ECCV 2022.
> >
> > [9] Minghao Zhu, Youzhe Song, Ge Jin, and Keyuan Jiang. 2020. Identifying Personal Experience Tweets of Medication Effects Using Pre-trained RoBERTa Language Model and Its Updating. In Proceedings of the 11th International Workshop on Health Text Mining and Information Analysis.
> >
> > [10] Amir, S., Gandelsman, Y., Bagon, S., Dekel, T. (2023). On the Effectiveness of ViT Features as Local Semantic Descriptors. In: Karlinsky, L., Michaeli, T., Nishino, K. (eds) Computer Vision – ECCV 2022 Workshops. ECCV 2022.
> >
> > [11] Xiaohua Zhai, Xiao Wang, Basil Mustafa, Andreas Steiner, Daniel Keysers, Alexander Kolesnikov, Lucas Beyer. LiT: Zero-Shot Transfer With Locked-Image Text TuningProceedings of the IEEE/CVF Conference on Computer Vision and Pattern Recognition (CVPR), 2022.
> >
> > [12] Norelli, Antonio et al. “ASIF: Coupled Data Turns Unimodal Models to Multimodal Without; Thirty-seventh Conference on Neural Information Processing Systems (NeurIPS), 2023.
> >
> > [13] Kadkhodaie, Z., Guth, F., Simoncelli, E. P., & Mallat, S. (2023). Generalization in diffusion models arises from geometry-adaptive harmonic representation. arXiv preprint arXiv:2310.02557.
> >
> > [14] Wu, C. H., & De la Torre, F. (2022). Unifying diffusion models' latent space, with applications to CycleDiffusion and guidance. arXiv preprint arXiv:2210.05559.
> >
> > [15] Klabunde, M., Schumacher, T., Strohmaier, M., and Lemmerich, F., “Similarity of Neural Network Models: A Survey of Functional and Representational Measures”, arXiv e-prints, 10.48550/arXiv.2305.06329.

---

> > > ### Author Response · Authors · 2023-11-22
> > >
> > > We thank the reviewer once again for providing valuable feedback and suggesting actionable modifications to improve the paper further.
> > >
> > > Please let us know if all concerns were correctly addressed in our revision or if there are any further questions or concerns to clarify during the remaining discussion period.

---

> ### Comment · Reviewer_8YvH · 2023-11-23
>
> Thanks for the authors's response and the concerns are addressed clearly, thus I will keep the score.

---

### Official Review · Reviewer_NofE · 2023-11-01

**Soundness:** 3 good
**Presentation:** 2 fair
**Contribution:** 3 good
**Rating:** 6
**Confidence:** 3

**Summary:**

This paper extends the work on Relative Representation by ensembling multiple relative representations obtained by different distances. The combination of four distances cos, Euc, L1, $L_\infty$ and three ensemble methods concat, sum, and attention are explored in the text. Extensive experiments across text, graph, and vision domains demonstrated that the ensembled version can improve the performance of zero-shot stitching.

**Strengths:**

1. Ensembling multiple relative representations is a reasonable idea and it enhances the power of the original cosine relative representation.
2. The experiments are extensive. There are 28 tables including the appendix.
3. The writing style is formal.

**Weaknesses:**

1. The selection of distances seems arbitrary.
    - (a) While the Euclidean distance is invariant under the Euclidean isometries and is a reasonable candidate beyond the Cosine distance. What is the rationale of the rest of the distances? Any geometric intuitions?
    - (b) The Euclidean isometry is a special case of conformal (angle-preserving) map. For experiments that show better performance on Euclidean distances than on Cosine distances, what can we say about the underlying symmetries of the neural representation? Does it mean that that latent space contain less invarinace? I am asking this question because I want to see what extra understanding on neural representations we can get from this new formulation.
    - (c) Page 2 "... which, combined, can capture arbitrary complex transformations of the latent space". It seems an overstatement to claim that the four chosen distances can capture "arbitrary complex transformations".
2. The Assumption in page 3 does not read smoothly.
    - (a) The equivalence class of encoders is defined as the set of E such that $ \pi_\mathcal{M}TE=\pi_M E, \forall T\in\mathcal{T}$. This definition is confusing. I fail to see why it is an equivalence class. For example, say, $\mathcal{T}_1$ is scalings, and $\mathcal{T}_2$ is rotations, and $E$ is a constant mapping to the origin. Does $\mathcal{T}_1$ and  $\mathcal{T}_2$ induces two different equivalence classes of transformations? But clearly $E$ belongs to both classes of transformations.
    - (b) Suppose $\mathcal{M}$ is a single point. Then $\pi_{\mathcal{M}}TE=\pi_{\mathcal{M}}E$ for all $E$ and all $T$. This definition does not contain any useful info then.
3. Page 5 "it is not possible to connect latent spaces of models with different initializations ..." It seems that the Pearson correlation for Cosine is higher than 0.94 in the left subfigure of Fig. 3 and higher than 0.8 in the right subfigure. What is the criterion for the statement of "no connection"? Any reference for the choice of criterion? I do not see this as "challenges the assumption in Moschella et al.".
4. Please clarify the aggregation used in Tab. 1 to Tab. 3, since there are multiple possibilities.
5. Sec. 4.4 leads confusion. What is the difference between SelfAttention and SelfAttention + MLP opt? Isn't the SelfAttention trained (finetuned)? If not, what is the exact computation formula for the SelfAttention aggregation? Where is the initial values of the attention weights come from? Also, the numbers in Tab. 5 does not match that of in Tab. 15.

**Questions:**

See weakness section.

typo:

1. page 4, invriances
2. page 9, fourth row -> fifth row

---

> ### Author Response · Authors · 2023-11-19
> **Response [1/2]**
>
> We thank the reviewer for the positive feedback, highlighting the extensive experiments and the formal writing style. We corrected the suggested typos, and we welcome the opportunity to address the raised concerns and offer clarifications on specific points:
>
> &nbsp;
>
> Weakness 1 (**Arbitrary complex**):
> - (a) **Arbitrary distances and geometric intuition**: The distances function chosen enforces invariances to distinct fundamental groups of transformations (e.g., orthogonal, conformal, and isometry groups with respect to different metrics: see Table 6 in the appendix for a complete picture). The aim is to combine the different sets of invariances as building blocks to capture more complex unknown transformations acting on real data. Each distance function will have its own specific properties: for instance, the choice of $L_{\infty}$ provides invariances w.r.t.  the Chebyshev transformations, resulting in robust to bounded perturbations of the points. In general, our approach could employ arbitrary similarity functions: characterizing from a theoretical perspective which classes of transformations can be captured using the employed invariances as building blocks is a promising future direction.
> - (b) **Conformal map**: The angle-preserving transformation we consider corresponds to the conformal orthogonal group:  its elements are compositions of global orthogonal transformations and local rescalings, which depend on each coordinate. The cosine distance is invariant to this group of transformations. The Euclidean isometry group is not properly a subset of this group, as it contains translations, while the former isn’t (similarly, cosine distance is invariant to local scaling, while euclidean distance isn’t). Therefore we attribute the difference in performance to the fact that these two distance functions retain different invariances. To avoid confusion, we incorporated a precise definition of orthogonal conformal transformations in the manuscript.
> - (c) **Statement**: We will re-modulate the statement in the paper and make it less bold. Still, our framework can capture transformations that the single latent spaces can’t capture (see qualitative example reported in Figure 14) where the compositions of all spaces lead to the best qualitative reconstruction), hinting at a major expressiveness in combining single invariances. In general, our framework could easily handle additional invariance other than the one considered in the paper, and as highlighted in the previous answer, it’s a promising direction to characterize theoretically which classes of transformations can be captured by combining different invariances as building blocks.
>
> &nbsp;
>
> Weakness 2 (**Assumption**): We will incorporate in the assumption specifics that avoids degenerated case, as the one presented by the Reviewer.
> - (a) The definition of the equivalence class of encoders depends entirely on the projection $\pi_\mathcal{M}$. In the example provided by the reviewer, if the encoder maps everything to the origin, still the projection will be the identity in this case, and the equivalence class will be with respect to the class of transformations $\mathcal{T}_1 \cup \mathcal{T}_2$. This is a degenerate case since the encoder will map everything to a constant; however, our definition still applies.  To the best of our understanding, this should address the reviewer's concern. If we missed something, we remain available for further clarification.
> - (b) The case where $\mathcal{M}$ corresponds to a single point is degenerate. Our definition still applies, but we agree with the reviewer that treating this specific case is not of great interest. $\pi_\mathcal{M}$ should ideally be a map invariant to all transformations $T \in \mathcal{T}$, but still retaining enough information in the latent spaces in order to solve the task at hand.
>
> &nbsp;
>
> Weakness 3 (**RR challenges**):  We have revised the sentence to clarify our statement. We recognize that invariances differ, and consistently depending on the cosine measure, might not be the optimal choice, even though it is sufficient.
>
> &nbsp;
>
> Weakness 4 (**Employed aggregation**): We have added a statement in the "Aggregation functions" section specifying the aggregation function utilized in the experiment, namely the SumAggregation (MLP+Sum).  Additionally, details have been included in the table's caption for clarity. The revised version is accessible in the PDF revision.

---

> ### Author Response · Authors · 2023-11-19
> **Response [2/2]**
>
> Weakness 5 (**Subspace selection Sec. 4.4**): We have incorporated additional details into the section, and we hope the following clarification provides a better understanding of our experiments:
> - **Experimental details**: Section 4.4 focuses on utilizing the Self-attention aggregation described in the method section under "Aggregation functions," which involves a single self-attention layer. The distinction lies in using "SelfAttention" to denote our classic method, where a self-attention layer aggregates different projection functions. On the other hand, when referring to "SelfAttention+... opt" we indicate that there is a fine-tuning procedure at stitching time, which is unique to this particular experiment. In particular, "SelfAttention+MLP opt" indicates the fine-tuning is performed on the whole classifier head; meanwhile, "SelfAttention+QKV opt" indicates only the Query, Key, and Value are fine-tuned. This is an analysis to understand if we can select an optimal subspace (i.e., perform subspace selection) at a reasonable cost at stitching time. Please refer to the [general response](https://openreview.net/forum?id=vngVydDWft&noteId=44LjVL0qyv) for further details on the zero-shot stitching procedure.
> - **Table results**: The results do not match because we utilized a more expressive classifier in this experiment. We chose this approach to establish a less favorable scenario and illustrate that our method still performs better than using a single projection function. In our opinion, this is a less favorable scenario since we are fine-tuning more parameters in the "SelfAttention+MLP opt" setting.
> Following the reviewer's suggestion, we have moved the deep classifier results to Appendix A.6 for a more accurate comparison. In the main paper, we now showcase the results obtained with a simpler classifier, aligning with the data in the full appendix table and with the classifier adopted in all the other experiments. The updates are already present in the PDF revision.
>
> &nbsp;
>
> ---
>
> We hope that our responses have addressed the concerns raised by the reviewer and that the modifications made to the paper have resolved any doubts. Additionally, we have included supplementary experiments on ImageNet and reconstruction tasks, detailed in the [General Response](https://openreview.net/forum?id=vngVydDWft&noteId=44LjVL0qyv). We are open to further discussions and available to address any additional concerns or questions.

---

> > ### Comment · Reviewer_NofE · 2023-11-22
> > **Response to rebuttal**
> >
> > Thanks for the response. It addresses most of the questions, except for the definition of "equivalent class". My understanding of the equivalent class is per the definition in https://en.wikipedia.org/wiki/Equivalence_class, that the equivalence classes form a partition of all the encoders. In the paper's definition, the equivalent class is parametrized by manifold M and transformation T. I do not think different choice of M or T would results in non-overlapping subsets for the set of all encoders.

---

> ### Author Response · Authors · 2023-11-22
>
> We express our gratitude to the reviewer for their insightful comments and are pleased to note that their concerns have been adequately addressed.
>
> Regarding the definition of equivalence class:
>
> We concur with the reviewer's observation that varying $\mathcal{M}$ and $\mathcal{T}$ may identify overlapping sets of encoders (for instance, considering $\mathcal{T}_1$ as a subset of $\mathcal{T}_2$, such as $\mathcal{T}_1$ being rotations and $\mathcal{T}_2$ being orthogonal transformations).
>
> However, our intention was to define the equivalence class as identified by the projection $\pi_\mathcal{M}$, i.e., fixing $\mathcal{M}$ and $\mathcal{T}$. In this context, for a specific $\pi_\mathcal{M}$, the set of possible encoders are partitioned in those invariant to $\mathcal{T}$ on $\mathcal{M}$ and those not. These two sets are disjoint by construction.
>
>
> Different selections of $\mathcal{M}$ and $\mathcal{T}$ will lead to identifying distinct equivalence classes. We will clarify this point and refine the definition accordingly in our manuscript.
>
> We remain available for further clarifications and kindly ask the reviewer to reconsider their evaluation in light of their belief that their concerns have been correctly addressed.

---

> > ### Comment · Reviewer_NofE · 2023-11-22
> > **Follow up discussion**
> >
> > Yeah makes sense.

---

### Official Review · Reviewer_ay1h · 2023-11-01

**Soundness:** 4 excellent
**Presentation:** 4 excellent
**Contribution:** 2 fair
**Rating:** 6
**Confidence:** 3

**Summary:**

This paper expands on Relative Representations by allowing several distances to be combined, which allows incorporating additional invariances in the resulting representation.

They first present evidence that single distances aren’t sufficient as they are data/model dependent (the original work used Cosine distance), they then explore adding 3 new distances (Euclidean, L1, L_\infty) in Text, Image and Graph domains.

This is a rather simple yet very clear and well-executed paper, which brought back Relative Representations to my attention, a nice idea which got washed up in the recent wave of LLM excitement. It might have limited scope, but currently I lean favorably.

**Strengths:**

1. The paper has a clear focus, presents the problem well, and is overall extremely well executed.
2. It was easy to follow and the extensions to the math were very well brought up.
3. It explores appropriate choices of distances and aggregations. Good details and interpretations were provided for what one should expect from them (e.g. Table 6 in the Appendix was exactly what I was looking for)
4. Results are clear and do improve in predictable fashion over baselines.

**Weaknesses:**

1. I feel like too much of the paper is spent on presenting evidence for the sub-optimality of single distance relative representations. I did not really understand why that point was made so repeatedly (Figure 1, Figure 3, Figure 4, Appendix Figure 8 and 9), instead of spending more time presenting different *combinations* of distances and their benefits/implications. In effect Table 1 is the first time a clear combination of distances is shown, and it is clearly better than the rest, so I would have wanted more of that.
2. Equally, as a result, less emphasis and space was spent explaining the results in 4.2, 4.3 and 4.4. I had to go back to the original paper to remember/understand what “zero-shot stitching” meant and how it was implemented.
3. Details were lacking in a few places, for example which aggregation function was used for most of the results. I assume MLP+sum given it performed the best in Section 4.3, but this isn’t spelt out?
4. Section 4.4 is also lacking in details and could benefit from some improvements, see below.
5. It is potentially of limited scope, but I would defer to the majority vote to see if that is a blocker or not.

**Questions:**

1. Do you really need to spend that much space and energy on presenting the failures of single distance Relative Representations?
   1. Figure 1, 3 and 4 are all making a similar point, and Section 4.1 does not feel as crucial as its length suggests it.
   2. I would probably recommend re-balancing this down and using the extra space to expand on the other Results sections.
   3. I would recommend keeping either 3 or 4 in the main text but not both.
   4. I am not sure that Figure 1 is the best framing figure to open the paper with, I might prefer to start with Figure 2.
2. The aggregations functions are presented well in Section 3, but it would have been useful to present implications for the choices of Sum and SelfAttention, in a similar manner to Concat (“giving to M the structure of a cartesian product space”).
   1. The Sum aggregation is actually a DeepSet by implementation. I would have liked having this spelt out explicitly and discussed?
3. The choice of Anchor points A_X and their implications on the invariances or properties of the relative representations are not discussed.
   1. Section 4.2 mentions using 1280 randomly selected fixed anchors. Did you try changing it? Does it affect distances differently?
4. I could not find which aggregation function was used for results in Table 1, 2 and 3. This should be specified clearly.
5. It feels like showing other combinations of distances (instead of “single” vs “all”) would have been helpful, especially if different domains require different distances.
   1. Section 4.4 tries to go in that direction, but the Transformer aggregation is not the best one and combined with my issue 4, I wasn’t sure what you used, so it muddles the results.
6. Section 4.4 would benefit from being extended, I do not think it contains enough details currently.
   1. The experimental setup needs more details, there is no description of the transformer aggregation anywhere I could find.
   2. Table 5 should contain the value for the best other aggregation (e.g. MLP+sum?), as currently it makes it harder to see if QKV opt is sufficiently accurate or not.
   3. It is unfortunate that the Transformer aggregation performs poorly. It would be good to bring the MLP+Transformer one to the main text, or at least present more clearly what model is used. It is not my expectation that a DeepSet should outperform a Transformer if it has enough layers?

---

> ### Author Response · Authors · 2023-11-19
>
> We thank the reviewer for the positive feedback, highlighting the clear focus, presentation, and extremely good execution. We welcome the opportunity to address the raised concerns and offer clarifications on specific points:
>
> &nbsp;
>
> Weaknesses 1, 5 & Question 1 (**Work scope and space for presenting RR failures**): It is crucial to emphasize that our paper's primary goal is to delve into and comprehend the transformations that connect different and independent latent spaces. Building upon the observations that emerged with the analysis, we aim to achieve latent communication without prior knowledge of the optimal invariance to incorporate. The extensive discussion in the paper was undertaken as a first step to study the emerging similarities between latent spaces empirically. Please refer to the [General Response](https://openreview.net/forum?id=vngVydDWft&noteId=44LjVL0qyv) for further discussion on our work scope.
>
> &nbsp;
>
> Weakness 2 (**Model-stitching definition**):  We expanded the description of the stitching procedure, dedicating a specific paragraph after the experimental setting (Section 4.2) and Figure 13 describing the stitching procedure. The revised version is accessible in the PDF revision.
>
> &nbsp;
>
> Weakness 3 & Question 4 (**Employed aggregation**): We have incorporated a statement in the "Aggregation functions" section specifying the aggregation function used in the experiments. Additionally, details have been included in each caption for clarity. The revised version is accessible in the PDF revision.
>
> &nbsp;
>
> Weakness 4 & Question 6 (**Subspace selection details Sec. 4.4**): We updated the section with additional details:
> - **Self-attention aggregation**: Section 4.4 focuses on using the Self-attention aggregation described in the method section under “Aggregation functions”, which is a single self-attention layer. We want to remark that this is the only case in which we perform stitching-time fine-tuning since we fine-tune the Q, K, and V parameters responsible for blending the subspaces. The stitching is done zero-shot in all the other experiments, as detailed in sec 4.2. All the details were included in the PDF revision.
> - **Self-attention aggregation**: We have included the results using the MLP+Sum (DeepSet) aggregation mechanism in Table 5. We believe this addition enhances the comprehensiveness of our study.
> - **Self-attention results**: We acknowledge your doubt about DeepSet outperforming transformer architectures. However, in our case, it is only a single attention layer rather than a full transformer, and we believe this distinction is a crucial factor contributing to the performance.
>
> &nbsp;
>
> Question 2 (**DeepSet**): We agree on the similarity with DeepSet and added the appropriate citation in the "Aggregation functions" section. The revised version is accessible in the PDF revision.
>
> &nbsp;
>
> Question 3 (**Anchors selection**): In response to the reviewer's insightful question, which enabled us to further enhance the strength of the proposed method, we conducted an ablation study on the number of randomly selected anchors for the stitching task using CIFAR100 across three different anchor seeds. **The results reveal that varying the number of anchors leads to different transformations, indicating that a single projection function cannot incorporate the desired invariance.** In contrast, our method enables the attainment of the highest score regardless of the number of anchors or the seed value employed for the uniform random sampling of anchors. We have introduced a new section (Section A.7) in the PDF revision for additional details, including Table 10 and Figure 9.  Given the importance of this finding, we also mentioned it in the main manuscript.
>
> &nbsp;
>
> Question 5 (**Aggregation ablation**): We sincerely hope the experiments presented in the appendix effectively address the question and provide valuable insights. Tables 13 and 14 showcase the performances using image and text data, conducting an ablation study across all the aggregation methodologies described in Section 3. Additionally, Table 15 presents a comprehensive ablation on various numbers of distances, combining only one, two, or all four projection functions. While acknowledging resource and time constraints prevented a combinatorial exploration of all possible settings, we trust that the results offer valuable insights to the discussion. Your understanding and consideration are greatly appreciated.
>
> &nbsp;
>
> ---
>
> We hope that our responses address the primary concerns raised by the reviewer, particularly the weakness regarding the perceived limited scope of the paper. Additionally, we have included supplementary experiments on ImageNet and reconstruction tasks, detailed in the [General Response](https://openreview.net/forum?id=vngVydDWft&noteId=44LjVL0qyv). We are more than willing to engage in further discussions if additional clarification is needed.

---

> > ### Author Response · Authors · 2023-11-22
> >
> > We thank the reviewer once again for providing valuable feedback and suggesting actionable modifications to improve the paper further.
> >
> > Please let us know if all concerns were correctly addressed in our revision or if there are any further questions or concerns to clarify during the remaining discussion period.

---

### Author Response · Authors · 2023-11-19
**General response [1/2]**

We express our sincere gratitude for the valuable feedback provided by the reviewers.
We very much appreciate they found *“the motivation well presented”* (`8YvH`), that *“the paper has a clear focus, presents the problem well, and is overall extremely well executed”* (`ay1h`), and commending that  *“the experiments are extensive”* (`NofE`), as well as that *“the explanations and illustrations are mostly clear and intuitive”* (`8YvH`).
We also thank the reviewers for recognizing that our work  *“explores appropriate choices of distances and aggregations”*  (`ay1h`), highlighting it is *“easy to follow, and the extensions to the math were very well brought up”* (`ay1h`)  and that *“is a reasonable idea”* (`NofE`).  Lastly, we are grateful for the acknowledgment that our *“results are clear and do improve in predictable fashion over baselines”*  (`ay1h`).

&nbsp;

### Relationship with Relative Representation (RR)

Regarding the relationship between our work and the concepts presented in Moschella et al. 2022, it is essential to clarify that our research follows a distinct path. Our work is not a direct extension of theirs.

Our main objective is to **characterize the transformations relating different independent latent spaces**. Given the challenges of tackling this problem from a theoretical perspective, we have approached the problem empirically to gain an initial understanding of the phenomenon. Our approach involves studying these transformations by incorporating different invariances into latent representations and analyzing the resulting spaces' correlation. Indeed, we leverage the RR framework proposed by Moschella et al. to enforce these different invariances into the latent spaces. However, we want to highlight this **choice is arbitrary, and other methods to infuse invariances could be used**.

Conversely, Moschella et al. (2022) demonstrated that when different latent spaces share semantics, a simple invariance to angle-preserving transformations can facilitate latent communication between them.

&nbsp;

### The stitching procedure is zero-shot

We would like to emphasize that **our zero-shot stitched model is never subject to further training or fine-tuning**. It is important to note that while some components of our module have learnable parameters (e.g., using the  MLP+Sum aggregation methodology), these parameters are exclusively trained during the initial, end-to-end network training phase and are then employed in their trained, frozen state in the stitched model.
The only exception to this setting is the one in Section 4.4, “Subspace selection”, where we want to assess the possible gains in integrating additional information into the subspace selection process. Therefore, we compare a tuning of only that part of the module to the classical tuning of the downstream classifier.

We have dedicated a specific paragraph right after the experimental setting (Section 4.2) and Figure 10 that describes the stitching procedure to clarify the method.

&nbsp;

### Additional analysis and results on ImageNet and reconstruction
Following the reviewers' feedback, we introduced three major new experiments:
- Qualitative and quantitative experiments for the zero-shot stitching on a **new task**: **reconstruction on CIFAR100 with AEs**. This experiment also offers a visual depiction of the performance of our method. Please refer to Figure 14 and Table 24 for the obtained results.
- Performance of the proposed model on the **ImageNet1k** dataset, which is larger than the other employed datasets. Results in Table 23 show that the proposed methodology utilizing the *Aggregation by sum* consistently matches or outperforms the others.
- Analysis **on the number of anchors** and **on different anchors selections** in Section A.7, Table 10, and Figure 9. These new experiments show the strong robustness of our method to changes in the anchor sampling.


&nbsp;

In summary, we appreciate the opportunity to clarify these points and are thankful for the chance to enhance our work through this valuable exchange with the reviewers!

---

> ### Author Response · Authors · 2023-11-19
> **General response [2/2]**
>
> ###  Changelist
> We sincerely thank all the reviewers for their constructive feedback on our work, suggesting actionable modifications to improve the paper further.
>
> Following their advice, we list here the main changes we adopted:
> + Add qualitative (Figure 14) and quantitative (Table 24) performance comparison on zero-shot **CIFAR100 reconstruction with AEs** (`pGUF`).
> + Added zero-shot classification **results on Imagenet1k** (Table 23) in Appendix (A.9) (`8YvH`).
> + Added analysis **on the number of anchors** and **on different anchors selections** in Section A.7 (`ay1h`, `8YvH`).
> + Added **experiment on QKV optimization with a linear classifier**  in Section 4.4 and moved the older in Section A.6 (`NofE`).
> + Clarified the zero-shot stitching procedure in Section 4.2 and added Figure 10 to **describe the method visually** (`ay1h`, `pGUF`).
> + Improve clarity and presentation of Section 4.4 (`ay1h`, `NofE`).
> + Included MLP+Sum aggregation in Table 5 (`ay1h`).
> + Added clarification in Section 3 that, unless specified otherwise, the results employ the Aggregation by sum (MLP+Sum), which is now also reported in the captions of the tables (`ay1h`, `NofE`).
> + Included conformal transformation definition (`NofE`).
> + Re-written statement on arbitrary complex transformation (`NofE`).
> + Revised the sentence on RR challenges to clarify our statement (`NofE`).
> + Re-formulated RR definition (`pGUF`).
> + Fixed minor typos (`NofE`).
>
> We provided details and specific replies as answers to each review.

---

### Meta-Review · Area_Chair_UMHe · 2023-12-18

**Metareview:**

This paper builds on the relative representation works, which project independent latent spaces into a shared relative space, and questions the use of a single distance measure when doing so. Instead, the paper proposes to combine four distances via various ensembling methods, empirically determining the best combination through thorough experimentation. Results are shown across a range of settings and ablations, notably in a zero-shot stitching setting used in prior works.

  Overall, the reviewers appreciated the clear focus and demonstration of the downsides of using a single distance, the formal description and exposition, and thorough results and experiments. Some weaknesses were raised including imbalanced emphasis on the motivation versus the technical method for stitching, lack of comparison of additional aggregation methods, lack of detailed explanation motivating the specific distances used and the corresponding invariances produced or random selection of anchors, and scale of experimentation. The authors provided a thorough rebuttal and discussion with the reviewers, including clarification of the method (e.g. that fine-tuning is not involved during stitching) and other aspects, and importantly results on ImageNet1k which has thus far been lacking in prior works as well.

  After considering the paper, reviews, rebuttal, and discussion, I believe the paper is a strong addition to this sub-field and provides thorough experimentation and interesting findings regarding the use of multiple distances, as well as larger-scale experiments that enhance its impact. Therefore, I recommend acceptance.

**Justification For Why Not Higher Score:**

Overall, the paper is positively regarded by the reviewers after the rebuttal. However, its impact on the larger field (rather than the smaller sub-field it is in) might be limited, and so may not be of interest to those outside of it. As a result, I opted not to select it as oral.

**Justification For Why Not Lower Score:**

The paper could be a poster as well; however, since it does have some interesting things to say about representation learning and in particular stitching along with larger-scale experiments, and the fact that the reviewers positively rated it after rebuttals, I would opt for a spotlight. However, I would not object if it is reduced to a poster.

---

### Decision · Program_Chairs · 2024-01-16

Accept (spotlight)